# A lightweight durable full-body electrical stimulation suit for haptic feedback and therapeutic applications

Jin Hee Hwang [1,11], Sun Hong Kim [2,11], Ju-Hwan Kim [3,4,11], Jae-Young Yoo [5,11], Jeongmin Seo[6], Geonoh Choe[1], Jae Min Lee[1], Byungkeun Choi [1], Sungjun Park [7], Joohoon Kang [8], Sang Min Won [9], Jeonghee Kim[1,6] ✉, Dong-Wook Park[3,4] ✉ & Yei Hwan Jung [1,6,10] ✉

Electrical stimulation represents a promising approach for the integrated delivery of haptic feedback and therapeutic interventions. Devices engineered for electrical stimulation interactions can simultaneously support a range of biomedical applications, including tissue regeneration, wound healing, pain management, and cosmetic procedures. When implemented at a full-body scale, such systems can provide precise spatiotemporal haptic feedback within extended reality (XR) environments, while simultaneously serving as therapeutic platforms for conditions amenable to electrical treatment. We introduce a full-body electrical stimulation suit which comprises a textile-based compression garment embedded with soft electrical stimulators designed for universal applicability across diverse user populations. An integrated calibration system continuously monitors contact pressure at each stimulation site across the body and dynamically adjusts electrical stimulation parameters based on site-specific impedance, enabling precise and consistent electrotactile feedback. The suit's lightweight, breathable, and durable design ensures comfort and wearability comparable to that of standard undergarments, thereby addressing the usability and discomfort limitations commonly associated with existing full-body systems. By incorporating soft and flexible materials, the suit supports prolonged usage without compromising the effectiveness of electrical stimulation. This technology holds significant potential for enhancing immersive experiences in XR applications and therapeutic interventions by seamlessly integrating high-fidelity haptic feedback with optimized wearability.

Electrical stimulation offers significant potential across diverse applications, including haptic feedback in extended reality (XR) systems and therapeutic interventions. When integrated with virtual or augmented reality, XR-assisted therapy combines immersive environments with targeted electrical stimulation to enhance rehabilitation, pain management, and neuromuscular recovery[1–3]. By utilizing the immersive and interactive features of XR, this approach actively engages patients both cognitively and physically, while electrical stimulation is employed to activate muscles, modulate pain perception, and support neural recovery. Within XR environments, electrotactile stimulation functions as a high-fidelity haptic feedback modality by delivering precisely modulated electrical signals to the skin with rapid

response times[4–6]. This technique offers high spatial resolution[6–8] and adaptable configurations[9,10], enabling the simulation of diverse tactile sensations through controlled variations in intensity and frequency[7,11–14]. As such, electrical stimulation provides multi-functionality, supporting the simultaneous delivery of therapeutic and sensory stimuli through a unified platform. An optimal system would thus integrate haptic feedback and therapeutic electrical stimulation within a single wearable interface—such as a full-body suit—allowing users to engage in immersive XR experiences while concurrently receiving clinical interventions. This dual-functionality approach has the potential to significantly enhance results in areas such as functional training, cognitive therapy, pain management, and neuromuscular rehabilitation, thereby offering a cohesive and efficient solution for both interactive and therapeutic applications.

However, achieving uniform electrical stimulation across large body surfaces remains a critical challenge[11,15,16], primarily due to the absence of an integrated mechanism for coordinating multiple stimulation patches within a single wearable platform, as well as inconsistencies in stimulation delivery caused by interindividual variability in body morphology. Inconsistencies in contact pressure at the skin-stimulator interface further complicate stimulation delivery by altering electrode contact area and spacing, which can compromise the reliability of the stimulation. Such discrepancies may also result in discomfort, including sensations of pricking or skin irritation due to leakage currents[17,18]. Consequently, most conventional electrotactile devices are limited to glove-like configurations designed for localized haptic feedback on the hands, particularly the palms and fingers, which significantly constrains their scalability for full-body applications.

Furthermore, therapeutic electrical stimulation is generally less effective on the palms and fingers[12], as it is optimized for broader skin regions such as the arms, legs, and chest. Consequently, glove-based designs are suboptimal for systems aiming to combine haptic feedback with therapeutic stimulation. In addition, many haptic systems employ non-breathable materials such as silicone rubber at the skin-device interface[19–21]. These materials tend to trap moisture, leading to sweat accumulation that can degrade signal quality and increase the risk of skin irritation or lesions. Breathability is thus a critical requirement for haptic interfaces, particularly in XR applications, as it promotes user comfort without compromising the fidelity of tactile feedback. Moreover, for extended use, breathable materials help prevent current leakage caused by moisture accumulation on the skin, thereby ensuring stable and reliable sensory performance.

In this study, we introduce a textile-based, lightweight, durable full-body electrical stimulation suit (TESS) integrated into a compression garment. The suit is engineered to form a soft, comfortable, and scalable interface with the skin, suitable for large-area applications (Fig. 1a). To maximize user comfort, the garment is constructed from a lightweight, breathable, and stretchable polyurethane (PU) textile that provides an ergonomic and adaptable fit. For the stimulation interface, the suit incorporates a dehydration-mitigating conductive hydrogel (DMCH), which retains its electrical performance over prolonged use. All functional components are seamlessly integrated using stretchable, textile-printable conductive ink, ensuring robust electrical interconnectivity without compromising the mechanical flexibility of the garment.

A critical feature of the electrical stimulation suit is its capability to deliver precise electrotactile stimulation by compensating for ergonomical variability of individuals through an integrated calibration software system. The system dynamically monitors pressure across the entire body that is applied from the pressure garment-based suit and adjusts stimulation parameters in real time to ensure consistent tactile feedback at each stimulation site. The core software architecture of the suit (Fig. 1b) follows a systematic workflow, which starts with the measurement of pressure at each corresponding body location using

embedded sensors, enabling quantification of the impedance at the electrode–skin interface. Based on such calibrated feedback, the system generates customized electrotactile stimuli that maintain consistency across varying pressure conditions (Fig. 1c). Specifically, detection of low pressure (i.e., high impedance) at a local site increases stimulation voltage, while high pressure decreases it, maintaining consistent current delivery and ensuring stable electrical and tactile feedback. This functionality represents a significant advancement in the delivery and understanding of full-body electrotactile perception. Our suit is capable of producing four distinct tactile sensations—touch, tickle, roughness, and pressure—by modulating the amplitude and frequency of electrical pulses. The real-time control enabled by this architecture greatly enhances tactile precision, which is essential for delivering reliable and high-fidelity electrical stimulation.

## Results

### Design and performance of the stimulation electrodes

Conventional electrical stimulation electrodes[22] typically rely on strong adhesion and low impedance to maintain a stable skin–electrode interface. However, their high modulus and tacky surfaces often cause skin irritation, making them unsuitable for extended use. In contrast, textile-based electrodes[23] offer improved softness and comfort, but their poor adhesion to skin limits attachment robustness, necessitating alternative integration methods such as pressure-based fixation. Figure 2a illustrates the layout of an electrical stimulation system fabricated on a soft, highly breathable textile substrate that combines high conductivity with mechanical stretchability. The interconnect network is formed using a composite of silver (Ag) flakes and PU conductive ink (Ag-PU), which is coated directly onto the textile. This composite forms a stretchable conductive layer embedded within the fabric, enabling reliable electrical connections between individual electrodes and a microcontroller-controlled stimulation circuit. The Ag-PU interconnects are pre-patterned and printed onto the textile, ensuring the integration of multiple electrodes into a large-area array. One such array, covering a full side of the compression garment ($4 \times 4$ configuration over an area of 3700 cm²), is shown in Supplementary Fig. 1a, b.

During the coating process, the ink's capillary action promotes deep penetration into the PU textile substrate, resulting in a durable bonding layer[23] Cross-sectional scanning electron microscopy (SEM) analysis confirmed that the Ag-PU conductors infiltrate the textile to a depth of approximately 160 μm, effectively anchoring within the 360 μm-thick PU matrix to form mechanically robust interfaces (Supplementary Fig. 2).

To further enhance adhesion and conductivity, the system incorporates HydroMed—an ether-based hydrophilic PU that absorbs approximately 50% of its weight in water and undergoes linear expansion. HydroMed also possesses strong adhesive properties[24], and its thermoplastic nature facilitates the stable integration of Ag particles without causing physical deformation. This allows the formation of highly conductive pathways without requiring conventional sintering, thereby preserving material integrity and flexibility.

Supplementary Fig. 4a–d detail the properties of a stretchable Ag-PU conductor screen-printed onto a PU-based textile, using inks with varying Ag volume fractions ranging from 40% to 80%. The Ag-PU conductor conforms to the textile's stretchability, maintaining functional integrity during deformation (Supplementary Fig. 4a). Under tensile loading, the conductor with a 40% Ag volume fraction achieved a maximum strain of 620%, exhibiting a relative resistance change ($\Delta R/R_0$) of approximately 75, while the conductor with 80% Ag volume fraction reached a strain of 200% with a relative resistance change of just 3 (Supplementary Fig. 4b). The conductivity of the ink (80% Ag volume fraction) was measured at $5 \times 10^3$ S/cm. These results highlight a fundamental trade-off between conductivity and fracture strain, governed by the Ag volume fraction, as detailed in Supplementary

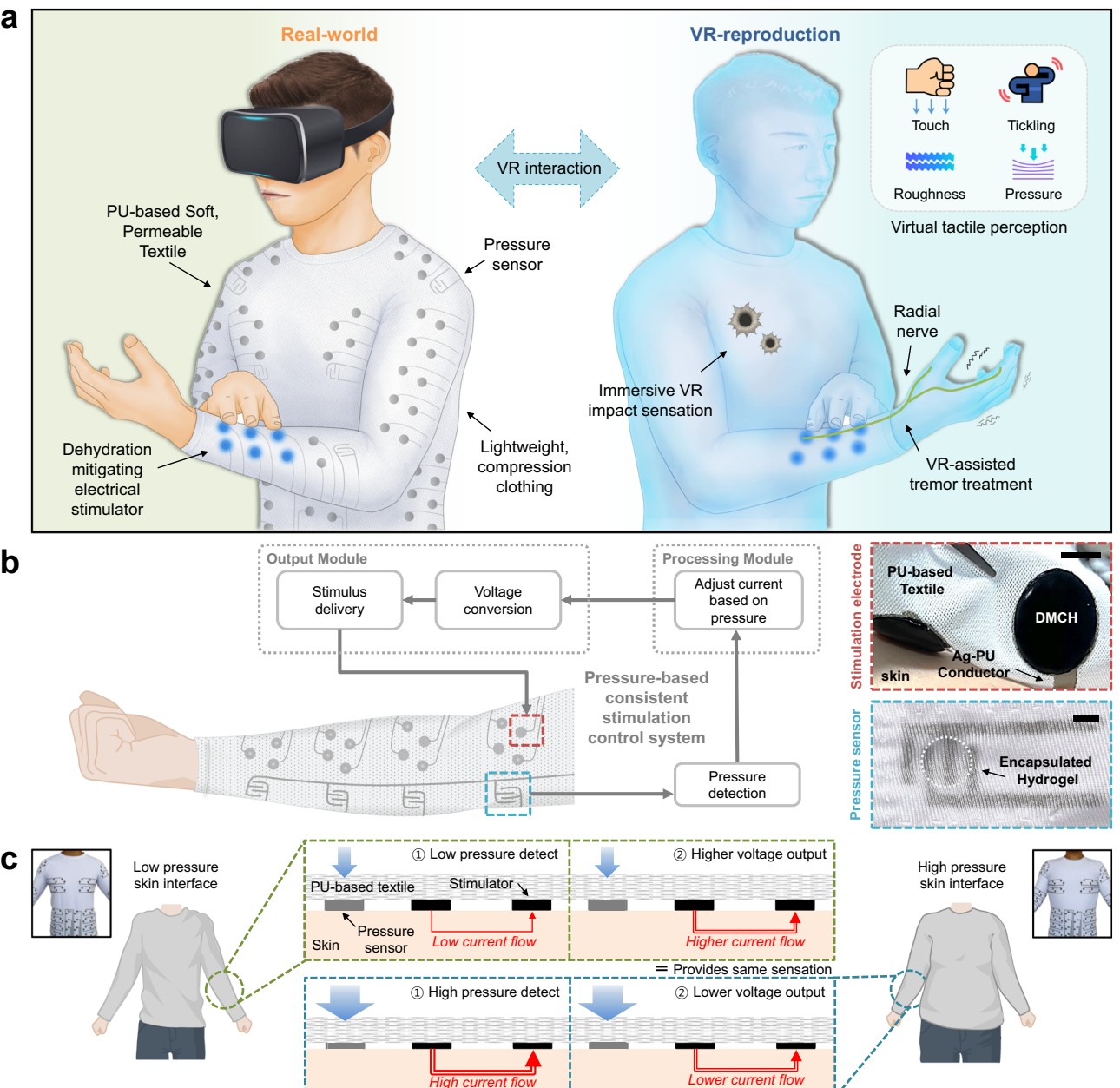

**Fig. 1 | Design of a large-area textile-based electrical stimulation feedback suit.**
**a** Conceptual illustration of the textile-based electrical stimulation feedback suit. The suit is constructed using a lightweight, breathable polyurethane (PU)-based textile, which delivers four distinct electrotactile stimuli to the skin through a dehydration-resistant hydrogel under appropriate compression. The system enables immersive interaction within virtual reality (VR) environments. **b** Flowchart illustrating the real-time haptic feedback mechanism on the forearm. Integrated resistive pressure sensors detect localized pressure variations and adjust stimulation intensity accordingly, providing personalized tactile feedback based on the user's unique sensory response. Scale bars, 5 mm. **c** Illustration and photograph of a participant wearing the feedback suit. Due to variations in body shape, different pressure levels (low on the left and high on the right) result in changes in current delivery. The system dynamically senses clothing pressure and modulates output voltage in real time to maintain consistent tactile stimulation across different body regions.

Fig. 4c. Given that skin stretchability varies across the body—reaching up to 70% in joint regions[25,26]—an Ag-PU conductor with 80% Ag content, offering 200% stretchability, is sufficient to ensure robust and reliable performance in wearable electronic applications. Moreover, cyclic tensile testing at 30% strain confirmed the mechanical and electrical stability of the optimized Ag-PU conductor over 1000 cycles (Supplementary Fig. 4d). Detailed evaluations of the printing accuracy and electrical properties of the Ag-PU conductor are provided in Supplementary Notes 1 and 2 (Supplementary Figs. 3a, b, 5, 6, and Supplementary Table 1).

Conventional screen-printing inks typically rely on metal particles dispersed in insulating or weakly conductive binders and require high-temperature sintering to achieve conductivity, which can damage substrates and increase processing time[27,28]. The low glass transition temperature ($T_g$) and low crosslinking density of the HydroMed-based PU, which creates a free volume that ensures enhanced contact between Ag fillers, allow the Ag-PU conductor to achieve high conductivity without the need for thermal processing, preserving the textile substrate and reducing processing time. Supplementary Fig. 7 presents the resistance stabilization behavior of the Ag-PU conductor

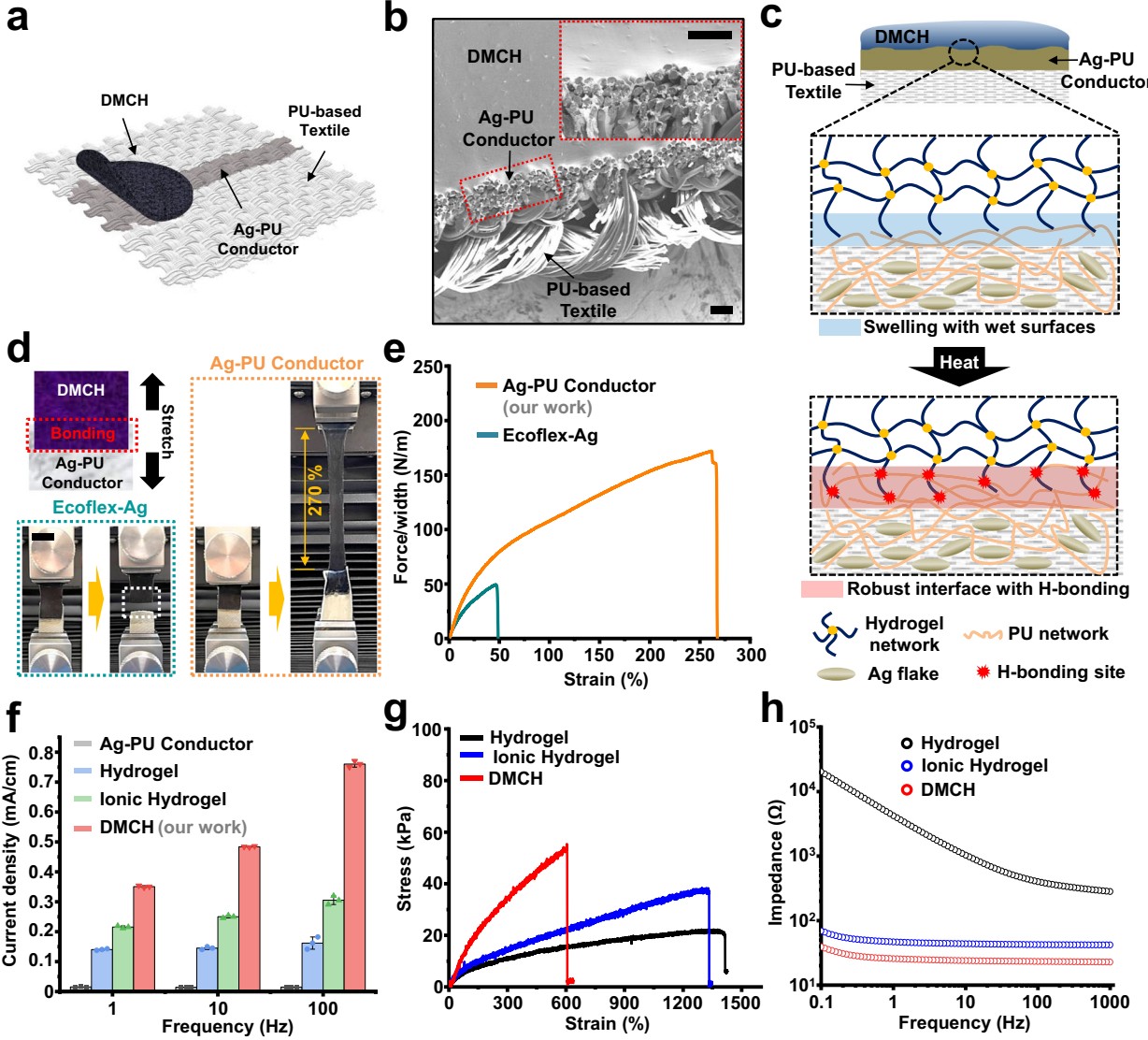

**Fig. 2 | Characterization of the electrical stimulation electrode. a** Schematic diagram of a single electrical stimulator showing the DMCH and the Ag-PU conductor printed on a PU-based textile. **b** Cross-sectional SEM image showing seamless integration of the DMCH on the Ag-PU conductor printed onto a PU-based textile. No interfacial gaps were observed. This experiment was repeated independently three times with similar results. Scale bars, 100 μm. **c** Schematic showing the mechanism of robust adhesion between DMCH and the Ag-PU conductor. **d** Photographs of the peeling test used to compare the peeling force of the DMCH/ Ag-PU conductor with that of the DMCH/silicon elastomer. Scale bars, 20 mm. **e** Peeling forces per width of DMCH attached to the Ag-PU conductor and silicon elastomer. **f** Current density measured on skin as a function of frequency for the Ag-PU conductor (used as the interconnect) and three electrical stimulation electrodes with different hydrogel compositions ($n = 3$ repeated measurements in a single participant; dots represent individual measurements; bars represent mean values; error bars, s.d.). **g** Stress as a function of strain for conductive hydrogel (red), hydrogel (black), and ionic hydrogel (blue). The DMCH is capable of withstanding strains of up to 600%. **h** Impedance spectra of various hydrogel electrodes with different compositions: pure hydrogel (black), ionic hydrogel (blue), and DMCH (red). The DMCH formed by mixing PEDOT and ions exhibited the lowest impedance, owing to the doping effect of the ions on PEDOT.

after screen-printing at room temperature. At lower Ag volume fractions (e.g., 40%), the limited concentration of Ag particles necessitates approximately 60 s for percolation path formation, driven by solvent evaporation, which induces particle rearrangement and network development. In contrast, at 80% Ag content, sufficient interparticle connectivity is established during solvent evaporation, resulting in more rapid stabilization of electrical resistance. Consequently, higher Ag volume fractions lead to improved initial conductivity and faster resistance stabilization. In addition, as shown in Supplementary Fig. 8a, b, when the same Ag-PU conductor was screen-printed onto three textile substrates with distinct woven structures, the resulting traces exhibited similar absolute resistances (1.90 Ω, 1.96 Ω, and 2.08 Ω for

Textiles 1–3, respectively), indicating that the electrical performance in this work is largely insensitive to the underlying textile substrate.

In this study, DMCH was employed as the base electrode. DMCH is synthesized by incorporating the conductive polymer poly(3,4-ethylenedioxythiophene):polystyrene sulfonate (PEDOT:PSS) with hygroscopic aqueous ionic salts, such as lithium chloride (LiCl), resulting in a biocompatible[29] hydrogel with high electrical conductivity and low impedance. Structurally, DMCH consists of a cross-linked polymer network that retains a significant amount of water within its porous matrix[30]. The porous architecture is formed through the polymerization of monomers, crosslinkers, and initiators, yielding a three-dimensional network. Upon hydration, water penetrates the

interstitial regions between polymer chains, promoting the development of the porous structure.

The uniform dispersion of PEDOT:PSS within the hydrogel further refines the microporosity, increasing the physicochemical interface available for ionic and electronic conduction. Cross-sectional SEM images reveal that as the PEDOT:PSS content increases from 8.4 wt% to 21.6 wt%, the microporous structure is preserved while the pore size decreases (Supplementary Fig. 9). These smaller pores enhance contact with the skin[31], leading to reduced impedance[32], and improved ion transport efficiency[33], thereby making DMCH an effective low-impedance material for electrotactile electrodes. Low-impedance electrodes are crucial for delivering tactile sensations with minimal noise and efficient charge injection. Impedance measurements conducted on DMCH electrodes of various diameters (ranging from 5 mm to 40 mm) demonstrated that the material maintains a favorable low impedance across the standard frequency range used in transcutaneous electrical nerve stimulation (TENS), typically between 1–150 Hz (Supplementary Fig. 10)[34,35]. A discussion on the thickness of the DMCH is provided in Supplementary Note 3 (Supplementary Figs. 11, 12).

A circular DMCH-based electrical stimulation electrode was directly laminated onto the Ag-PU conductor, followed by heat treatment at 60 °C, resulting in a strong and durable bond. This robust integration ensures reliable electrical performance and mechanical stability under repeated deformation, maintaining consistent conductivity during use. The strength of the interface between the DMCH electrode and the Ag-PU polymer is primarily attributed to hydrogen bonding. The Ag-PU polymer, composed of ether-based hydrophilic urethanes, exhibits excellent adhesive and cohesive properties along with high water absorption capacity. Upon physical contact, the Ag-PU conductor absorbs moisture from the hydrogel, leading to surface swelling that ensures partial penetration of the hydrophilic polymer chains from DMCH into the Ag-PU substrate. This interpenetration, combined with subsequent crosslinking, produces a seamless, gap-free interface (Fig. 2b). The polymer chains become grafted onto the elastomeric surface of the Ag-PU, forming hydrogen bonds between urethane groups (NH–COO), which further reinforce the interface with heat (Fig. 2c)[36]. A peel test confirmed the mechanical robustness of the laminated interface, demonstrating superior adhesion compared to conventional Ag-ink/silicone rubber (Ecoflex-Ag) composites (Fig. 2d). For instance, the DMCH–Ag-PU laminate withstood stretching up to 2.7 times its original length without delamination, whereas the Ag–silicone interface exhibited weak adhesion and early failure (Fig. 2e and Supplementary Note 6).

To evaluate the electrical performance of the developed electrodes on skin, two electrodes (cathode and anode) were applied to the skin surface, and a voltage was used to induce current. To compare the individual and combined effects of ionic salt and conductive polymer on electrode performance, current densities were measured for four electrode types: a Ag-PU conductor, a pure hydrogel, an ionic hydrogel incorporating LiCl to enhance ionic conductivity, and the DMCH formulation containing both LiCl and PEDOT:PSS (Fig. 2f). Although the Ag-PU conductor is directly coated on the textile for interconnects, it does not produce electrotactile sensations on the skin. When used as a standalone electrode, it delivers substantially lower current than hydrogel-based electrodes because the Ag-PU interface exhibits much higher skin–electrode impedance than wet hydrogel contacts, limiting charge transfer and effective current delivery. As a result, the current density remains below the perceptual threshold and is not expected to elicit a noticeable sensation. In contrast, current density increased progressively from the pure hydrogel to the ionic hydrogel and reached its maximum in DMCH, thus allowing efficient current delivery. Mechanical properties were characterized via stress–strain analysis (Fig. 2g), showing that the inclusion of PEDOT:PSS strengthened the polymer network in DMCH, yielding a relatively high Young's modulus (~55 kPa) and high stretchability (~600%). However, as ultra-high

stretchability is not a requirement for textile-based systems, DMCH was selected primarily for its superior electrical performance. Consequently, the DMCH-based electrode—integrating both PEDOT:PSS and LiCl—demonstrated significantly higher current density compared to the ionic hydrogel and exhibited the lowest impedance across the low-frequency range (1–150 Hz) typical of TENS (Fig. 2h).

Biocompatibility was evaluated using a 24 h skin-contact test in which a participant wore a sleeve containing DMCH and the printed Ag-PU conductor during daily activities (Supplementary Fig. 13). No visible skin irritation was observed after removal, consistent with prior reports supporting the biocompatibility of polyacrylamide-based hydrogels incorporating PEDOT:PSS or LiCl[81–83]. Because the Ag-PU conductor is embedded within the textile architecture, it is less able to maintain sustained, conformal skin contact than the protruding DMCH electrode layer. Prior studies have similarly used Ag–polymer composites for skin-interfacing applications, typically embedding Ag fillers within elastomeric matrices to minimize direct exposure[37].

## Characteristic of textile-based electrical stimulation suit

The experimental results presented in Fig. 3 highlight the key performance characteristics of the electrical stimulation suit. Hydrogels are widely employed in electrical stimulation applications due to their low modulus, mechanical softness, biocompatibility, and ability to form low-impedance interfaces with the skin. However, their susceptibility to dehydration remains a major limitation for long-term use. Dehydration significantly impacts both the mechanical and electrical properties of hydrogels, and must be carefully considered when integrating them into wearable electronic systems. Conventional strategies to reduce dehydration—such as encapsulating hydrogels with thin elastomeric films—often involve complex fabrication steps, including film–hydrogel assembly and ultraviolet (UV) curing processes[38]. These additional procedures increase production time and cost, while UV exposure can compromise the flexibility and stretchability of the hydrogel, further limiting practical application.

In this study, we introduce a DMCH-based electrode that intrinsically resists dehydration without requiring additional encapsulation or processing (Fig. 3a). Hydrogels formulated with lithium chloride (LiCl) demonstrate not only enhanced ionic conductivity (Fig. 2f, g) but also superior moisture-retention capabilities[39]. The hygroscopic nature of aqueous LiCl promotes strong hydrogen bonding with water molecules, thereby reducing evaporation and preserving the hydrogel's mechanical and electrical performance over time[40]. To evaluate dehydration resistance, we conducted a comparative study using DMCH, a pure hydrogel, and a PEDOT:PSS-containing hydrogel of identical size (circular shape, 10 mm diameter; initial weight, 0.1 g), under ambient environmental conditions (21.3 °C, 66.3% relative humidity) for 24 h. These three hydrogels were selected to determine whether the enhanced water retention was primarily due to the presence of LiCl or PEDOT:PSS. Both the pure hydrogel and the PEDOT:PSS hydrogel exhibited substantial water loss (~70 wt% and ~66 wt%, respectively), whereas the DMCH sample retained significantly more moisture, losing only ~36 wt% over the same period (Fig. 3b). The PEDOT:PSS hydrogel demonstrated slightly lower moisture loss compared to the pure hydrogel, likely due to the ability of PEDOT:PSS to partially reduce dehydration. DMCH exhibited a distinct behavior: after an initial moisture loss during the first ~4 h, it reached equilibrium and maintained a stable moisture content throughout the remainder of the 24-h period. To assess long-term environmental stability, we compared the dehydration behavior of DMCH with three other hydrogels (hydrogel, PEDOT:PSS hydrogel, and ionic hydrogel) under room temperature (20 °C, 46% relative humidity), high temperature (60 °C, 46% relative humidity), and high humidity (20 °C, 84% relative humidity) for 156 h (Supplementary Fig. 14). Quantitatively, DMCH maintained an average moisture content of 0.68% and 0.58% at room temperature and high temperature,

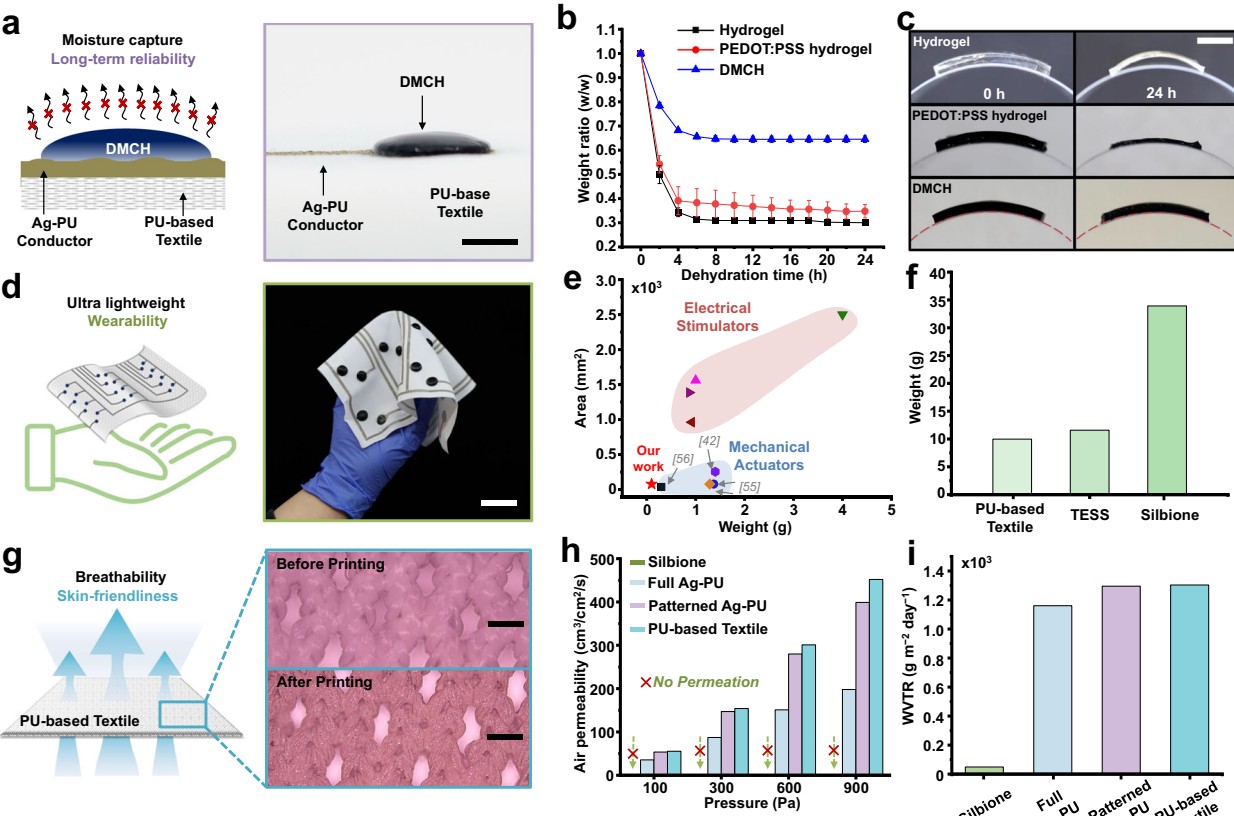

**Fig. 3 | Characteristics of the textile-based electrical stimulation feedback system. a** Side-view schematic and photograph demonstrating the dehydration resistance of DMCH. The hydrogel effectively retained moisture through its hygroscopic properties. Scale bar, 5 mm. **b** Moisture retention performance of DMCH (blue) compared with pure hydrogel (black) and PEDOT:PSS hydrogel (red) over 24 h under ambient conditions (21.3 °C, 66.3% relative humidity). DMCH exhibited minimal water loss, while the other hydrogels lost most of their moisture ($n = 8$; independent samples with identical thickness and diameter; data points represent means; error bars, s.d.). **c** Side-view images showing adhesion performance of DMCH (bottom), PEDOT:PSS hydrogel (middle), and pure hydrogel (top) on a curved surface (curvature = 0.033 mm⁻¹) after 24 h at 20 °C and 44.2% humidity. DMCH maintained conformal contact, whereas the other hydrogels showed delamination due to dehydration. Scale bar, 10 mm. **d** Schematic and photograph of a lightweight, wearable textile-based electrical stimulation device

featuring a 4 × 4 electrode array. Scale bar, 30 mm. **e** Comparison of the area and weight of a single electrode unit from this study with various mechanical actuators and electrical stimulators used in haptic interface applications. **f** Weight comparison of key packaging materials for wearable electronics, including bare PU-based textile, PU textile with printed Ag-PU conductor, integrated DMCH layer, and conventional Silbione encapsulant. **g** Schematic and SEM images showing the textile structure before (top) and after (bottom) Ag-PU conductor printing. The breathable textile retained its microporous architecture, maintaining air and moisture permeability. This experiment was performed once. Scale bars, 100 μm. **h, i** Air and moisture permeability evaluations for different packaging materials. **h** Air permeability measured under pressures of 100, 300, 600, and 900 Pa for bare PU textile, Ag-PU patterned textile, and Silbione reference. **i** Water vapor transmission rate (WVTR) measured over 24 h under ambient conditions (19.5 °C, 15% humidity).

respectively, compared with 0.26%/0.20% for the hydrogel and 0.17%/0.13% for the PEDOT:PSS hydrogel. Under high-humidity conditions, DMCH retained 1.23% moisture, higher than the 0.51% and 0.35% observed for the hydrogel and PEDOT:PSS hydrogel, respectively, reflecting the humidity-dependent, LiCl-driven hygroscopic characteristic.

To evaluate the conformability of hydrogel-based electrodes on curved surfaces, the contact area was assessed based on moisture retention data. Variations in perceived tactile sensations can result from impedance fluctuations at the electrode–skin interface, highlighting the importance of conformal adhesion for consistent performance. As shown in Fig. 3c, DMCH maintained full contact with a curved surface (curvature = 0.033 mm⁻¹) after 24 h, preserving both its structure and adhesion. In contrast, the pure hydrogel failed to conform fully, exhibiting visible gaps. This behavior is attributed to dehydration-induced shrinkage of the hydrogel's water-rich polymer network, which can lead to delamination from curved surfaces[41]. Because surface dehydration typically occurs more rapidly than interior drying, surface shrinkage stresses are intensified, promoting contraction and weakening interfacial adhesion. The PEDOT:PSS

hydrogel showed improved structural integrity during dehydration, maintaining adhesion and preventing collapse of the polymer network[42]. It shrank uniformly across the curved surface, thereby preserving contact. However, significant moisture loss led to substantial thickness reduction and a stiffened, less flexible state. In contrast, the ionic hydrogel, similar to DMCH, retained its moisture content and conformed well to curved surfaces over 24 h (Supplementary Fig. 15). These findings suggest that the DMCH-based electrode, due to its hygroscopic LiCl content, offers superior moisture retention and structural stability. This enhanced durability positions DMCH as a more reliable and long-lasting option for hydrogel-based electrotactile electrodes in wearable applications.

The use of PU-based textiles significantly reduces overall device weight, thereby minimizing physical load on the user and enabling the development of a lightweight, wearable stimulation system (Fig. 3d). In addition, distributing the stimulator array in alignment with regional variations in skin nerve density enhances the system's ability to replicate natural tactile sensations across diverse anatomical areas. Unlike highly sensitive regions such as the palm or tongue, the torso exhibits lower tactile sensitivity and higher perceptual thresholds, eliminating

the need for high spatial resolution in electrotactile stimulation. Consequently, torso applications commonly employ fewer, larger electrodes—typically exceeding 78 mm² in area[43,44]. Electrode size is a critical parameter in optimizing sensory feedback. Larger electrodes promote spatial summation, more readily activate underlying nerve fibers, and reduce skin impedance, though at the cost of reduced resolution. In contrast, smaller electrodes offer greater sensitivity across a broader frequency range and operate with lower power consumption. In this study, we employed small electrodes of approximately 78.54 mm², consistent with those used in previous forearm-based applications for stable and precise sensory feedback[45]. Electrode dimensions can vary based on anatomical location, local nerve density, and regional two-point discrimination thresholds.

Figure 3e compares the weight and area of the DMCH-based electrodes used in our system with those of mechanical actuators reported in previous studies[46–48], as well as commercial electrodes for TENS. The model information of the stimulators and actuators used for comparison are provided in the Methods section. Current commercial (e.g., bHaptics) and research-grade systems rely predominantly on arrays of mechanical actuators (ERM motors, LRAs, or pneumatic systems), which introduce substantial weight and bulk at high spatial densities. Accordingly, Fig. 3e positions the DMCH electrode not only as an alternative to conventional gel or TENS electrodes, but also as a lightweight, high-density building block for large-area haptic garments. In addition, Supplementary Table 2 compares DMCH with previously reported textile-based electrodes by summarizing electrode type, electrode area, and textile substrate type. Figure 3f presents a comparison of substrate and packaging material weights typically used to support electrodes and wiring in skin-interfaced electronics. At the same area of 3700 cm², our entire textile-based system remains significantly lighter—2.93 times lighter than a layer of standard silicone encapsulation material (33.93 g; thickness: 300 μm; modulus: 5 MPa; Silbione RTV 4420)[19]. The silicone layer thickness was chosen according to previously optimized encapsulation designs for ultrathin, stretchable electronics[20]. In total, the PU-based textile integrated with both DMCH-based electrodes and Ag-PU conductors weighs 11.57 g, while the bare textile weighs 9.89 g. This indicates that the additional functional components account for only 14.52% of the system's overall weight.

To ensure the long-term performance and user comfort of the electrical stimulation suit, we conducted a series of air and vapor permeability assessments. As shown in Fig. 3g, the micro-perforations in the PU-based textile remained unobstructed even after the Ag-PU conductor was printed onto the surface. This confirms that the breathability of the textile was preserved following the application of the conductive ink. To further evaluate vapor permeability, a PU-based textile patterned with Ag-PU conductors was positioned between two beakers in a custom setup. The lower beaker was filled with boiling water (100 °C), while the upper beaker was initially empty (Supplementary Fig. 16). Within 5 s, water vapor successfully permeated the textile and condensed on the interior walls of the upper beaker, demonstrating the excellent vapor permeability of the TESS. Quantitative air permeability was assessed using a standardized air permeability tester at various pressure levels (100, 300, 600, and 900 Pa) (Fig. 3h). Results showed that the textile patterned with Ag-PU textile retained high breathability, comparable to that of the unprinted PU-based textile (Supplementary Fig. 17). Although the presence of Ag-PU conductor led to an approximate 11.7% reduction in air permeability at 900 Pa compared to the bare textile, the performance remained significantly superior to that of conventional silicone-based materials such as Silbione. Fully printed textiles (Full Ag-PU) showed a further reduction, but still maintained acceptable levels of breathability for wearable applications. The water vapor transmission rate (WVTR) exhibited a similar trend (Supplementary Note 4). Under ambient conditions (37 °C and 19% relative humidity), the difference in WVTR

between the bare and patterned PU-based textiles was negligible—approximately 8 g m⁻² day⁻¹ (Fig. 3i).

## Sensory perception in multimodal electrical stimulation suit

In terms of tactile sensations based on electrical stimulation (i.e., electrotactile), human skin can be modeled as a parallel combination of resistor and capacitor[7,14], as shown in Fig. 4a. Electrical stimulation penetrates the high-impedance stratum corneum, allowing activation of mechanoreceptors such as Merkel's disks and Meissner's corpuscles through both direct current (DC) and alternating current (AC)[7,12–14]. Our system induces sensations—such as touch, tickling, roughness, and pressure—via controlled electrical pulse stimulation, offering a versatile and effective interface for tactile feedback.

We begin by outlining the configuration of the electrical stimulation electrode array, designed specifically for large-area applications across the body. The spacing between electrodes plays a crucial role in determining the nature of electrotactile stimulation. Shorter inter-electrode distances produce more localized sensations but require higher voltages to reach perceptual thresholds. Conversely, increasing the distance between electrodes allows the stimulation to diffuse more broadly, enabling effective activation at relatively lower voltages (Supplementary Fig. 18). Based on this principle, the center-to-center distance between the anode and cathode in the DMCH array was set to 20 mm, optimizing the trade-off between spatial resolution and stimulation voltage requirements[49]. The inter-electrode distance was selected considering the neural activation threshold[50–52], the spatial acuity of the forearm[53,54], and the need to reduce discomfort from movement, thereby ensuring wearability and interface stability. This spacing may require adjustment depending on anatomical site and user-specific variability. In addition, considering the upper arm—a region with relatively low tactile sensitivity—and interindividual differences in somatosensory acuity, we established a 50 mm spacing between adjacent electrode pairs (Supplementary Fig. 1b). This distance meets the two-point discrimination threshold for the upper arm (~45 mm)[55], ensuring reliable discrimination of distinct stimuli in torso-based applications. Figure 4b indicates the current profiles generated by pulse-shaped voltages at each electrode within the electrode array, configured as described above. By precisely controlling the current delivered through the skin, the system modulates the intensity and quality of the tactile sensations experienced by the user.

We conducted sensory perception tests to evaluate the effects the electrical stimulation on user sensation (Supplementary Fig. 19). Because high current densities can lead to adverse skin reactions such as burns or lesions[17,18], it is critical to define stimulation parameters within physiologically safe limits of voltage and current. Accordingly, we assessed both the perceptual threshold and the maximum tolerated current on the forearm (Fig. 4c), as well as the frequency-dependent perceptual threshold current (Fig. 4d). The perceptual threshold current is defined as the minimum current at which a user first detects a sensation. This threshold is closely linked to neural activation and provides a stable, reproducible reference for evaluating stimulation performance. As shown in Fig. 4c, the perceptual threshold exhibited low interindividual variability and a narrow margin of error, highlighting its reliability. In contrast, the maximum tolerated current showed greater variability across participants, reflecting subjective differences in sensory tolerance. Specifically, while some participants reported an increasing intensity of sensation, others experienced tickling or mild discomfort, and a few reported tolerating the stimulation up to the point of pain. These findings highlight the importance of precisely characterizing sensory parameters to ensure safe and personalized stimulation profiles (Supplementary Table 4). We also analyzed the correlation between stimulation frequency and perception threshold current across the skin. Although individual participants exhibited different absolute thresholds for sensation onset, their responses followed a consistent linear trend on a logarithmic

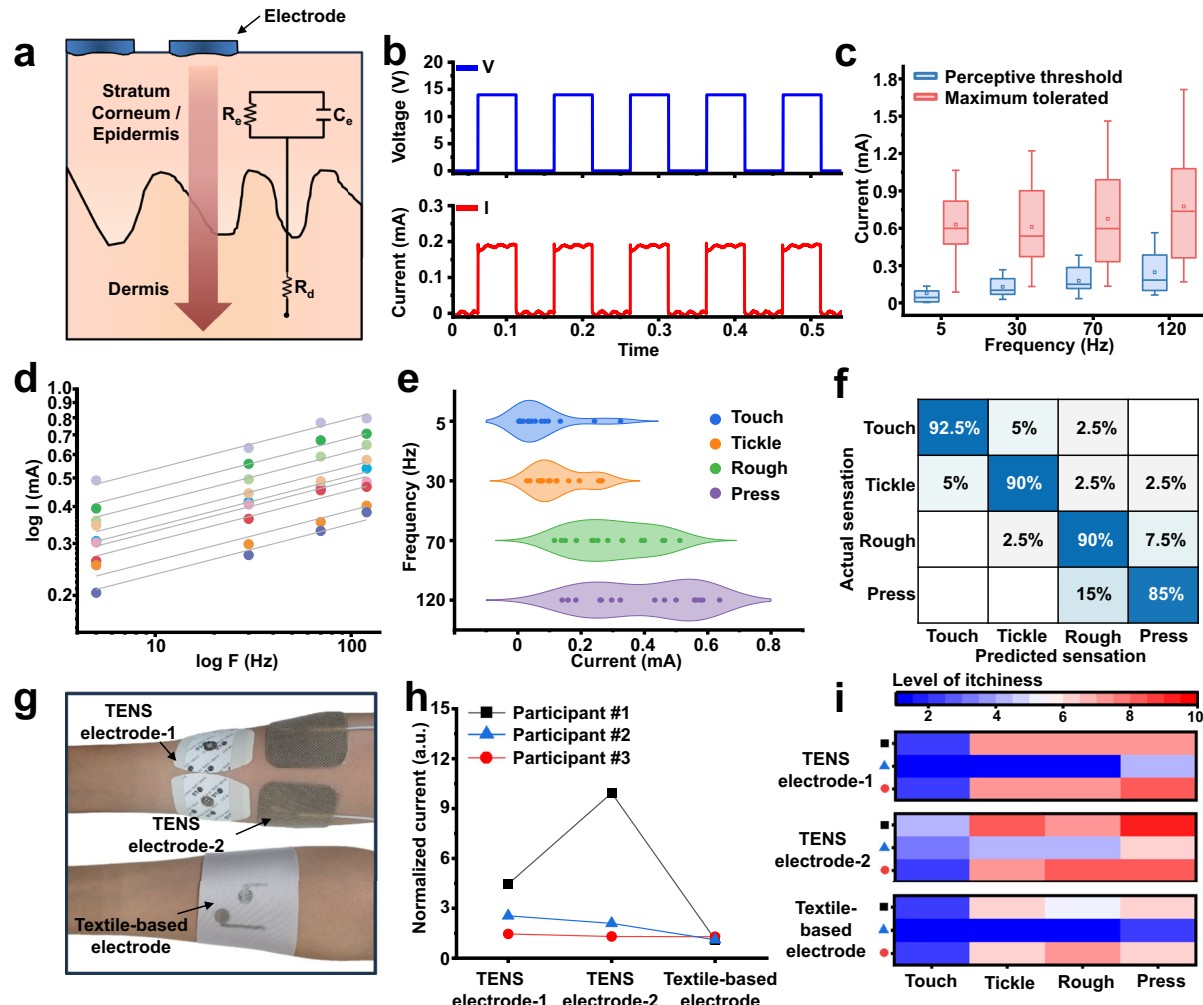

**Fig. 4 | User study of the multimodal textile-based electrical stimulation system. a** Schematic of the electrical impedance model of human skin, shown as a simplified equivalent circuit. **b** Representative waveforms of voltage (top) and current (bottom) signals applied to the skin during electrical stimulation. **c** Comparison of individual stimulus thresholds and maximum tolerable voltages across participants ($n = 15$). Squares indicate mean values; center lines show medians; box limits represent upper and lower quartiles; whiskers extend to $1.5 \times$ the interquartile range. **d** Log–log plot showing the linear relationship between perception threshold current and stimulation frequency. The fitted line has a constrained slope of 0.17076 with adjusted $R^2 > 0.9$ ($n = 9$). **e** Mapping of four distinct tactile sensations—touch, tickling, roughness, and pressure—based on stimulation current and frequency ($n = 15$; data shown as violin plots). **f** Confusion matrix showing recognition accuracy in 10 blind tests of forearm (posterior) stimulation without prior training. Participants identified sensations from four predefined categories. **g** Photograph of the forearm showing the placement of the textile-based electrode and two commercial TENS electrodes used to assess current variation before and after physical activity. **h** Normalized current measurements before and after a 10 min run on a track ($n = 3$). **i** Heat map showing subjective itchiness ratings following 10 min of exercise, evaluated for three different electrode types during delivery of various sensations ($n = 3$).

threshold current versus frequency scale, consistent with previous reports. Notably, all participants shared a common slope of 0.17076 (Fig. 4d). This trend may be attributed to the smoother sensory perception associated with higher-frequency stimulation, which generally requires greater current amplitudes for perceivable sensations[7].

Four discrete frequencies strategically selected based on established psychophysical principles: 5 Hz (touch), 30 Hz (tickling), 70 Hz (roughness), and 120 Hz (pressure). This selection leverages the finding that low-frequency stimulation (< 10 Hz) activates discrete mechanoreceptors to produce tapping sensations, whereas higher frequencies (> 100 Hz) induce sensory fusion, resulting in continuous pressure[56]. By manipulating the electrical parameters—specifically the amplitude and frequency of stimulation—various tactile sensations can be effectively rendered (Fig. 4e). Participants were able to distinguish discrete pulses at a low frequency of 5 Hz, which elicited light, tapping sensations reminiscent of gentle touch. Increasing the frequency to 30 Hz induced a tickling sensation that subtly propagated across the skin, with several participants likening it to the feeling of insects

crawling along the arm. At 70 Hz, the stimulation evoked a coarse tactile sensation, similar to the sensation of stroking animal fur. A further increase to 120 Hz produced a more intense pressure-like sensation, comparable to localized skin deformation. The average stimulation current showed a clear frequency-dependent increase. At 5 Hz, the mean current was 0.0776 mA (SD: $9.31 \times 10^{-5}$ A), rising to 0.3961 mA (SD: $1.72 \times 10^{-4}$ A) at 120 Hz. These results support the observation that higher frequencies require greater current amplitudes to produce perceivable tactile feedback. The increasingly broad and elevated threshold current ranges at higher frequencies may be due to a reduced sensitivity to incremental changes in sensation near 100 Hz, a phenomenon previously reported in the literature[24].

To assess the perceptual clarity of each rendered sensation, participants were asked to identify four distinct stimulation patterns based on predefined tactile descriptors: touch, tickling, roughness, and pressure (Supplementary Table 4). Ten participants were randomly presented with four pattern sequences and asked to match each to the most appropriate keyword. The classification task yielded a

maximum prediction accuracy of 92.5% and an average accuracy of 89.37% (Fig. 4f). Notably, many participants confused pressure with roughness. This observation is consistent with documented psychophysical limits in frequency discrimination. In the mid-to-high frequency range (> 50 Hz), the broad tuning of rapid-adapting afferents (e.g., Pacinian corpuscles) and temporal summation lead to overlapping neural recruitment, thereby reducing the perceptual separability of distinct frequencies[57]. As a result, users tend to report a more continuous vibration in which "roughness" and "pressure" converge toward similar percepts, with perceived intensity often providing a stronger cue than frequency in this regime. This reflects an inherent limitation of human tactile frequency resolution[35].

In addition, a user study was conducted to compare the performance of the TESS with that of commercial electrodes under perspiration-inducing conditions, focusing on the impact of sweat on sensory perception (Supplementary Note 5, Supplementary Fig. 36d). Sensory evaluation data for the commercial electrodes used in this comparison are presented in Supplementary Fig. 20a–c. Four participants were instructed to wear the TESS to their left forearm and two types of commercial TENS electrodes to their right forearm (Fig. 4g). They then engaged in a 10 min run on a track under conditions (13 °C and 66% relative humidity). TENS electrode-1 is a disposable electrode typically used for chronic pain relief, consisting of a two-layer design: a 47 × 36 mm adhesive layer and a central conductive region measuring 18 × 14 mm. TENS electrode-2, designed for both TENS and electronic muscle stimulation (EMS) applications, integrates a biocompatible adhesive hydrogel and a carbon-based conductive film into a single 50 × 50 mm unit. During the experiment, all electrodes were placed with a 10 mm spacing between their conductive edges for consistency in testing.

Post-exercise measurements revealed substantial inter-subject variability[58] and pronounced current amplification for the commercial electrodes. Specifically, the stimulation current increased by 1.4–4.4 × for TENS electrode 1 and by 1.3–9.9 × for TENS electrode 2 across participants, in some cases rendering stimulation difficult to tolerate due to elevated itchiness (Fig. 4h). This current increase is consistent with sweat accumulation at the skin–adhesive interface (Supplementary Fig. 21a, b). In contrast, the textile-based stimulation electrode showed only a 1.1–1.2 × increase in current across all participants, consistent with improved breathability and moisture management. Participants were also asked to rate the sensation of itchiness associated with each electrode on a 0–10 scale following stimulation (Fig. 4i). Both TENS electrodes caused higher levels of itchiness after sweating, particularly across wider skin areas. Participants also reported itchiness levels corresponding to the current amplification. In contrast, the textile-based electrode received consistently low itchiness scores, confirming its safe and comfortable operation under sweat-inducing conditions. The elevated current density in low-breathability electrodes promotes excessive spread of electrical signals across the skin, resulting in undesirable sensations such as itching or stinging. Furthermore, the strong adhesion of commercial electrodes can cause skin irritation or inflammation. The textile-based electrode avoids these drawbacks by maintaining conformal contact through garment compression rather than adhesives, ensuring stable electrotactile performance while enhancing comfort and minimizing skin irritation.

The TESS showed stable tactile performance under mechanical deformation. To evaluate its functionality under maximum strain conditions, we tested the system at the elbow—a joint characterized by frequent and significant movement and mechanical stress (Supplementary Fig. 22a)[59]. Starting from the perceptual threshold voltage (15 V), we incrementally increased the voltage by 1 V up to 25 V, measuring the resulting current through the skin at each level. Under 30% strain induced by elbow flexion, the textile-based electrode maintained a linear current response, exhibiting performance comparable

to that observed under unstrained, pristine conditions. To further assess durability, we conducted cyclic stretching tests on the posterior forearm, subjecting the textile embedded with Ag-PU conductors to 100 cycles of 30% strain (Supplementary Fig. 22b). The results showed negligible changes in current density across cycles.

## Development of large-area, full-body electrical stimulation suit

The TESS is highly adaptable and can be readily implemented by printing onto commercially available garments, such as compression sleeves (90% nylon, 10% PU; PS2000H_LG, 3 M) or standard clothing, thereby eliminating the need for custom-manufactured apparel (Supplementary Fig. 23). Given that the interface between DMCH and PU is governed by hydrogen bonding and polymer interpenetration, minor variations in PU content (10–13% in this study) do not significantly impact interfacial bonding strength. In our implementation, electrodes were printed on multiple regions of the body, including the chest, shoulders, arms, abdomen, and sides, resulting in a full-body suit that provides broad torso coverage (Fig. 5a). The base garment was an innerwear item (P0000BXF, GearX) composed of 88% polyester and 12% PU, size small. During garment fitting, the electrical stimulation suit maintained robust mechanical and electrical performance. The interface between the Ag-PU conductor and DMCH remained intact under stress, demonstrating reliable functionality and successful integration of electronic components with wearable textiles.

The suit delivers electrical stimulation through the application of controlled compression, eliminating the need for adhesives commonly used in conventional devices. Unlike traditional systems that rely on strong adhesive layers—often causing skin irritation, discomfort upon removal, and potential damage to embedded components—our suit secures its electrodes through garment pressure (Supplementary Fig. 33a, b). Because this wear modality can introduce positional shifts after repeated donning and doffing, we experimentally quantified the resulting positional displacement of the TESS suit, as shown in Supplementary Fig. 24. To address positional variations between wears (donning–doffing repeatability), inter-wear positional displacement was quantified over 10 repeated wears at four representative sites. Overall, the shift remained on the order of a single electrode footprint and within the same functional body part segment, and is therefore considered acceptable for the present implementation[60]. In addition, adhesion tests comparing the DMCH-based electrode with commercial TENS electrodes are presented in Supplementary Fig. 25. While both TENS electrodes achieved stable skin contact via high adhesion, peel-off tests revealed significant drawbacks: participants reported pain, visible red marks, and skin irritation following removal. These effects raise concerns about skin damage, increased risk of infection, and reduced reusability. In contrast, the DMCH-based electrode, secured solely by garment compression, left no visible marks or signs of irritation on the skin. Given the extensive surface area of the suit, garment pressure serves as an effective and scalable solution for electrode attachment. While our system uses a tight-fitting garment to establish a baseline contact pressure and mitigate gross instability while maintaining user comfort, residual variability in skin–electrode impedance remains unavoidable due to inter-individual differences in body geometry and site-dependent variations in electrode placement.

This mechanism is illustrated in Fig. 5b. Under low compression ($p_1$), the 2mm-thick DMCH electrode forms a soft, conformal interface with the skin ($a_1$), maintaining an inter-electrode distance of approximately 10 mm ($d_1$), thereby initiating a localized stimulation pathway. Conversely, under high compression ($p_2$), the DMCH's low-modulus allows it to expand its contact area ($a_2$) while reducing the inter-electrode distance ($d_2$), altering the electrical characteristics of stimulation. To experimentally validate the relationship between compression and sensory response, a resistive pressure sensor was developed and integrated into the system. This sensor was encapsulated to prevent current leakage through the skin. Its detailed

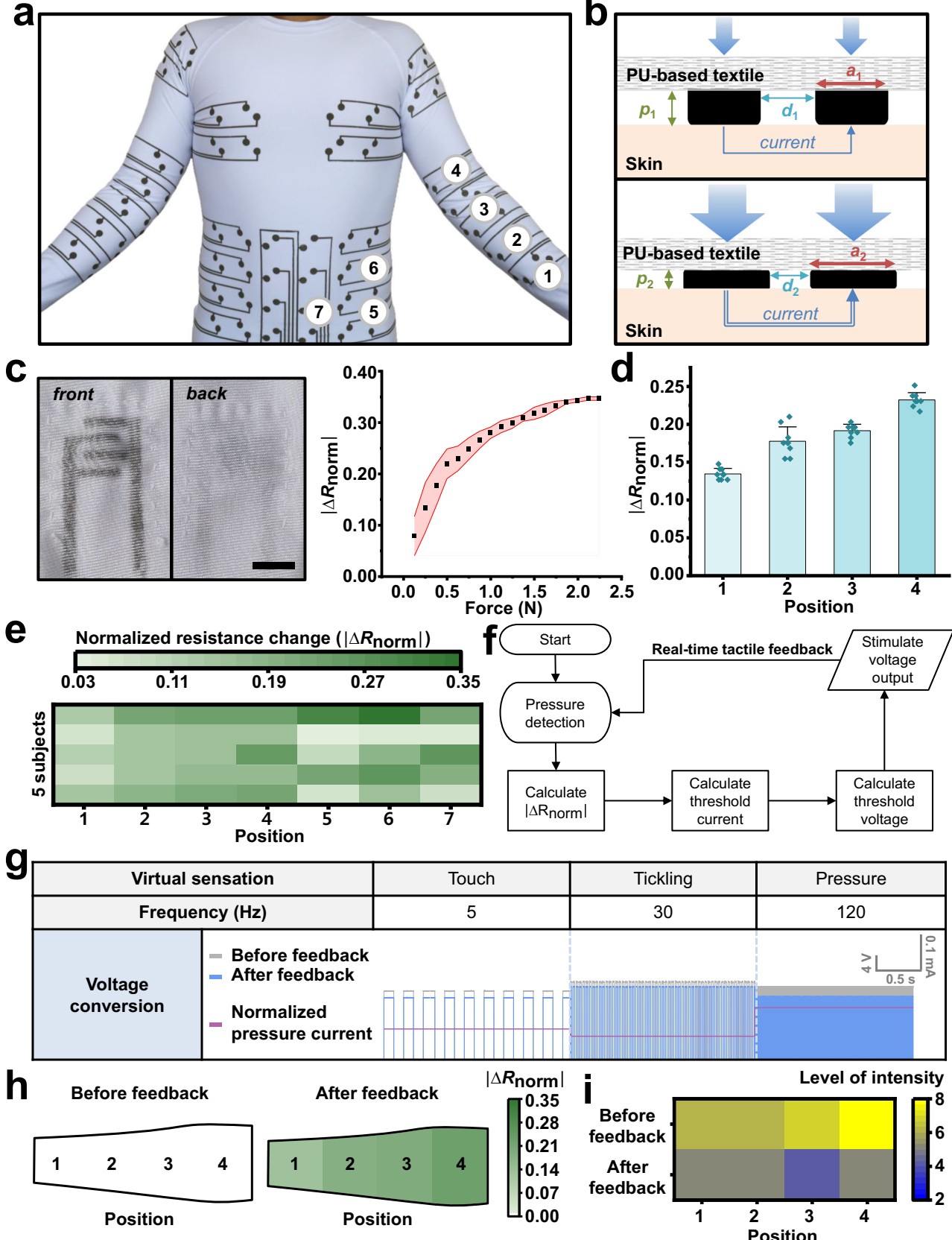

**Fig. 5 | User study on the full-body electrical stimulation suit. a** Photograph of a participant wearing the textile-based electrical stimulation suit designed for the chest, abdomen, sides, shoulders, and arms. The suit provides targeted compression and high spatial resolution for full-body stimulation. **b** Schematic illustrating the electrical stimulation mechanism, showing how current levels vary in response to changes in applied pressure ($p$), contact area ($a$), and interelectrode distance ($d$) under low-pressure (top) and high-pressure (bottom) conditions. **c** Photograph of the integrated pressure sensor embedded in PU-based fabric with a conductive hydrogel for real-time pressure sensing (left; Scale bar, 10 mm). Detection range of the sensor shown on the right ($n = 4$). **d** Normalized pressure changes measured by an interdigitated pressure sensor while participants wore a sleeve. Measurement locations included: wrist (1), mid-forearm (2), below the elbow (3), and above the elbow (4), using the elbow as the central reference point ($n = 9$ repetitions of donning and doffing; bar height: mean; error bars, s.d.). **e** Measured clothing pressure across different body locations (as shown in panel **a**) for five participants (three female and two male) with varying body types and ages. **f** Flowchart

illustrating the real-time electrical stimulation feedback system. **g** Three distinct tactile sensations were simulated through real-time tactile feedback. Wireless pressure sensors positioned across the body calculate garment pressure, triggering spatiotemporal activation of stimulation channels. Before feedback, the system considers both the threshold current and additional current variation with respect to the frequency. After feedback integrates all key parameters—including pressure-induced current adjustments—and enables real-time feedback and dynamic stimulation to accommodate individual users. **h** Heatmap of sleeve compression levels across arm positions 1–4 in a within-subject experiment. The sleeve on the left arm was operated without feedback compensation, whereas the sleeve on the right arm was operated with feedback compensation. **i** Heatmap of perceived intensity ratings across arm positions 1–4 in the same within-subject experiment (0–10 scale; 0: no sensation, 5: comfortable/moderate, 10: intolerable). The after-feedback condition shows more stable and comfortable perceived intensity compared to the before-feedback condition.

performance characteristics are shown in Fig. 5c. By normalizing resistance fluctuations, the sensor minimized variability, enabling accurate assessment of individual differences in garment pressure at the skin–fabric interface (Supplementary Fig. 34a, b). To evaluate sensor reliability, a single participant repeatedly donned and removed the garment nine times. The resulting standard deviation of 0.01898 (Fig. 5d) confirmed the sensor's consistency under identical pressure conditions. The pressure sensor operates as a voltage divider comprising a fixed 1 kΩ resistor and a variable resistor. The latter consists of a conductive gel deposited on an interdigitated electrode pattern, whose resistance changes with applied pressure. Higher pressure leads to a greater contact area between the gel and the electrode pattern, increasing current conduction and reducing resistance by expanding the available current path (Supplementary Fig. 26). Although reduced inter-electrode distance typically results in lower current flow at a given voltage, the increase in contact area under compression appears to have a more pronounced effect, ultimately enhancing current transmission.

Electrotactile sensation exhibits substantial interindividual variability in perceived intensity, primarily due to differences in the impedance at the electrode–skin interface. To quantitatively evaluate full-body electrotactile stimulation, we conducted a study involving five participants (three females and two males) with diverse body types. This investigation assessed the distribution of clothing associated with electrotactile stimuli. Figure 5e presents the clothing pressure applied at different body locations, measured using integrated pressure sensors. All participants wore suits of identical small size, and pressure readings were taken within a range considered comfortable by each wearer. A higher absolute value of the normalized resistance change ($\triangle R_{norm}$) corresponds to greater pressure. In general, pressure distributions varied with body type, with the arm regions (positions 1–4) exhibited a progressively increasing compression. To confirm that sensor resistance changes reflect pressure variations from local garment fit, we measured arm circumference at each sensor location (positions 1–4) for each participant using a flexible tape. Arm circumference increased from position 1 to position 4 and exhibited a matching position-dependent trend in the sensor responses (Supplementary Fig. 27). These results show that the sensors can resolve garment-pressure differences arising from inter-individual variations in body shape. Given the substantial variability in applied pressure, the results emphasize the need to address mismatches in electrotactile perception caused by differences in pressure distribution across users.

To ensure consistent haptic feedback under varying physical conditions, we implemented a real-time feedback control procedure, utilizing integrated pressure sensors (Fig. 5f). This feedback system continuously monitors the skin–electrode interface, accounting for pressure—a key factor influencing electrode adhesion and impedance.

The sensor translates voltage changes from a variable resistor into pressure values.

Achieving consistent sensory perception across users requires precise adjustment of the stimulation voltage to compensate for individual variability. A detailed description of the mechanism is provided in Supplementary Note 10 (Supplementary Figs. 29a, c, 30). Using this approach, pressure data from the forearm were captured via an integrated sensor, allowing the system to identify the stimulation site and output the appropriate voltage (Fig. 5g). By applying a predefined correction equation customized to each individual, the system dynamically adjusted the stimulation to deliver consistent tactile feedback. A within-subject comparison further validates the effectiveness of the proposed compensation strategy (Fig. 5h). In this experiment, sleeves were worn on both arms, and feedback compensation was applied only to the right arm (Fig. 5i). Participants rated perceived intensity on a 0–10 scale (0: no sensation, 5: comfortable/moderate, 10: intolerable) at positions 1–4. Without feedback, perceived intensity increased with garment pressure toward position 4; in contrast, the compensated condition produced more stable and comfortable sensations across positions, supporting that the proposed feedback improves the consistency of perceived intensity. In addition, Supplementary Table 5 provides a quantitative and qualitative comparison of TESS with representative haptic systems reported in prior work. This comparison highlights that TESS enables body-scale feedback, a capability that, to our knowledge, has not been demonstrated in previous electrotactile systems.

## Applications of textile-based electrical stimulation suit

The TESS has demonstrated the capability to deliver spatially distributed electrical stimulation across large areas of the skin using garment-induced compression, highlighting its potential to provide immersive, full-body tactile feedback for applications in gaming, training, and rehabilitation. We present three use cases of the suit in virtual reality (VR) and augmented reality (AR) environments that utilize large-area virtual tactile interaction: (1) simulating spatiotemporal bullet and grenade tactile impacts across the torso and arms in a VR gaming environment; (2) supporting treatment and training for essential tremor (ET) through real-time tremor detection using inertial measurement unit (IMU) sensors and Bluetooth low energy (BLE)-enabled wireless control in a VR setting; and (3) enabling dynamic AR experiences—such as the sensation of ants crawling along the arm—controlled through a graphical user interface (GUI) and camera input. In all three settings, the feedback mode accurately measured pressure and delivered appropriate electrical stimulation strength, ensuring precise electrotactile stimulation parameters across the full body.

Figure 6a depicts a participant wearing the TESS alongside a VR head-mounted display (HMD), experiencing simulated ballistic

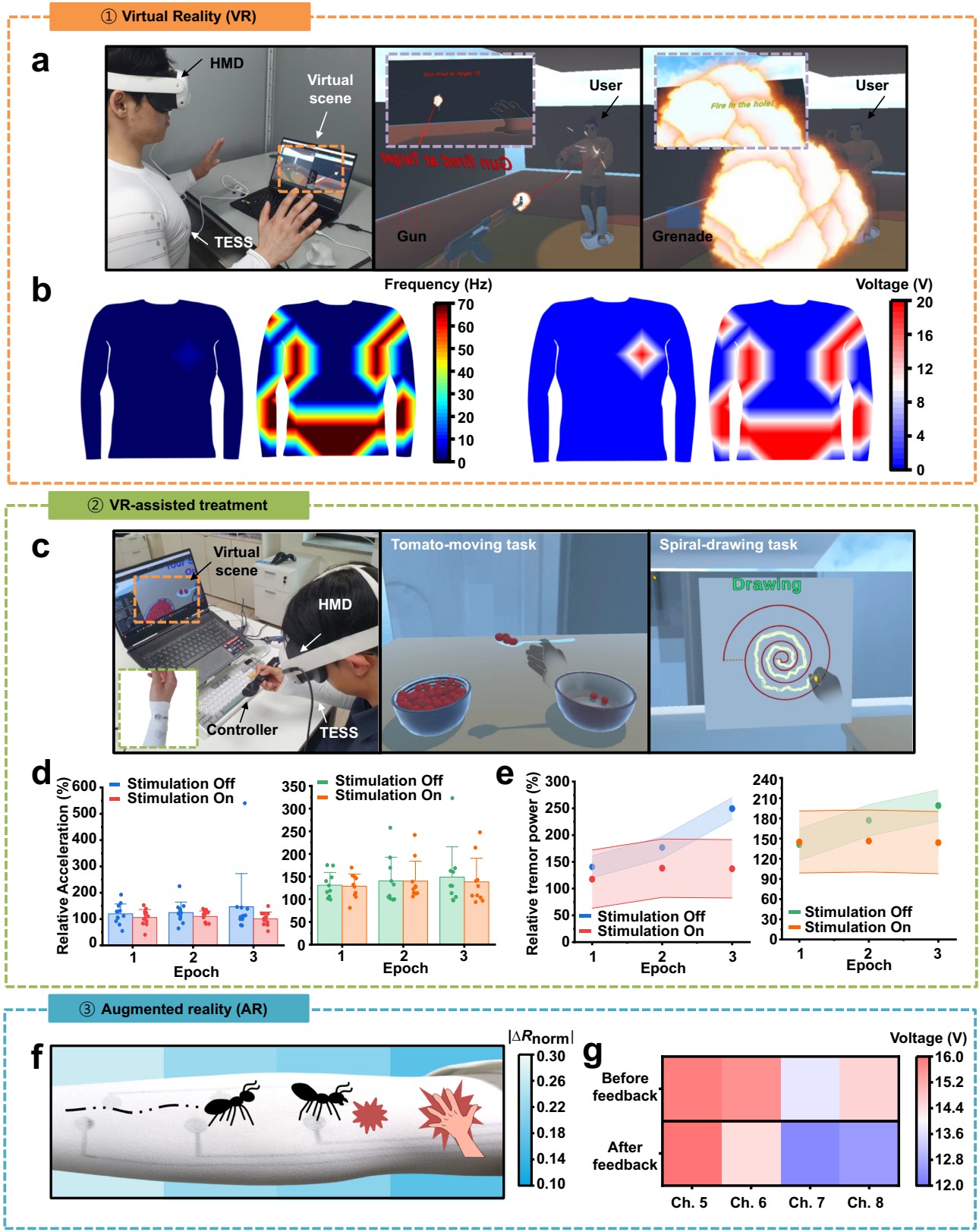

impacts in a virtual environment. The HMD tracks hand movements and synchronizes visual and auditory feedback with tactile output from the suit, providing real-time electrotactile stimulation. To enhance user immersion, participants can select between "bullet" and "grenade" modes. In bullet mode, 16 channels distributed across the chest, abdomen, upper arms, and forearms deliver two cycles of

intense, programmable 5 Hz stimulation, simulating the sharp, localized sensation of being shot. In grenade mode, all 16 channels positioned on the front torso are activated simultaneously with a coarse 70 Hz stimulus for approximately 8 s, replicating the sensation of a large-area impact. Figure 6b illustrates the voltage and frequency mapping of the suit during these ballistic events in the virtual reality

**Fig. 6 | Real-time electrical stimulation feedback interface. a** A participant wearing the TESS experiences immersive gameplay through integrated visual, auditory, and tactile feedback. The system simulates bullet (middle) and grenade (right) impacts during a shooting game. Insets show the user's in-game perspective for each mode. **b** Voltage and frequency distribution maps of the TESS suit, illustrating the stimulation range, intensity, and frequency for the bullet (left) and grenade (right) modes. **c** The participant wears the TESS on the radial nerve while performing two VR-based tasks: tomato-moving (middle) and spiral-drawing (right) tasks, as shown in the VR environment (left). The system records hand tremors via controller sensors. **d** Comparison of normalized acceleration amplitudes at the dominant frequency before and after electrical stimulation during the tomato-moving (left; $n = 12$ participants; dots represent individual participants; bars represent mean values; error bars, s.d.) and spiral-drawing (right; $n = 10$ participants; dots represent individual participants; bars represent mean values; error bars, s.d.) tasks, tracked over multiple epochs. **e** Normalized average tremor power and standard deviation across epochs for the tomato-moving (left; $n = 12$ participants) and spiral-drawing (right; $n = 10$ participants) tasks, showing changes in tremor intensity with and without stimulation. **f** Augmented reality (AR) simulation of crawling ants achieved through feedback. The AR scenario (right) is rendered on a mobile device, with virtual ants crawling along the arm. Wireless pressure sensors positioned across the arm calculate garment pressure. **g** The dynamically adjusted stimulation intensity via feedback is compared to the before-feedback intensity in the AR-based ant simulation.

(VR) environment. This VR-based electrotactile impact system not only enhances immersion by targeting a broader range of body regions but also demonstrates significant potential for high-fidelity applications in combat simulation, virtual gaming, and tactical training scenarios.

In another demonstration, we introduce a VR-based diagnostic and therapeutic system that integrates a virtual reality platform with the electrical stimulation suit, employing IMU sensors and BLE wireless controllers. Tremor is a common motor symptom observed in neurological conditions such as Parkinson's disease (PD) and Essential Tremor (ET). Previous studies[61–63] have demonstrated that skin surface stimulation can effectively reduce tremor severity, likely by modulating both peripheral and central components of the tremor network through activation of sensory afferent fibers near the wrist. To evaluate this system, healthy participants without known tremor disorders were recruited. Artificial tremors were induced via repetitive weight-lifting exercises, combined with wrist weights between experimental epochs[64,65]. Participants wore an HMD and used two types of controllers—a spoon-shaped controller and a marker-style controller—to quantitatively capture hand tremor severity (Fig. 6c and Supplementary Fig. 28).

Within the VR environment, participants completed two motor tasks designed to assess tremor after fatigue induction: (1) a tomato-moving task using a spoon and (2) an Archimedean spiral-drawing task (Supplementary Note 9, 11). They completed five sets of dumbbell exercises targeting the biceps and shoulders/forearm muscles, respectively, using 70% of their maximum load to induce muscle fatigue. To establish a baseline, participants first performed the tasks after the exercise without stimulation (Supplementary Fig. 35).

To further enhance tremor induction, participants performed both tasks while wearing a 1 kg wrist weight and the textile-based electrical stimulation device on their dominant wrist[66]. A total of twelve participants (mean age = 26.83 years; four females; Supplementary Fig. 36a) and ten participants (mean age = 26.80 years; three females; Supplementary Fig. 36b) were enrolled in the respective task groups. For stimulation, one electrode was positioned over the radial nerve near the wrist, while a second electrode was placed approximately 4 cm away (center-to-center), ensuring targeted stimulation of radial nerve branches. To effectively reduce tremors, a 100 Hz electrical stimulation signal with 37.5% duty cycle was applied at an intensity sufficient to elicit a perceptible sensation in participants[67]. Figure 6d presents the dominant acceleration amplitude for both the tomato-moving (left) and spiral-drawing (right) tasks. The dominant frequency—defined as the frequency with the highest power spectral density (PSD)—was identified for each cycle, and the corresponding acceleration amplitude was normalized to each participant's baseline. Tremor power was calculated within the frequency band of interest (4–12 Hz), commonly associated with tremor activity[67]. In the tomato-moving task, peak tremor amplitude progressively increased across the three experimental epochs when electrical stimulation was not applied. However, when electrical stimulation was administered, the peak amplitude remained relatively constant throughout the epochs.

In the spiral-drawing task, the effect of electrical stimulation was less pronounced, though a slight reduction in tremor amplitude was observed following stimulation. These differences may be attributed to the variations in muscle engagement. Figure 6e shows the average tremor power across the epochs for both tasks. In the absence of stimulation, tremor power consistently increased over time, indicative of fatigue-induced tremor. In contrast, with electrical stimulation, tremor power remained relatively stable, suggesting that the electrical input effectively counteracted tremor progression, even under induced fatigue conditions. These findings support the potential of the TESS for applications in the diagnosis, monitoring, and treatment of neurological disorders such as PD and ET.

To demonstrate the system's capabilities in an AR environment, we developed a dynamic simulation of an ant crawling on a participant's arm (Fig. 6f). The simulation sequence included a tickling sensation (~4 s) as the ant moved, a touch sensation (~2 s) representing a bite, and a pressure sensation (~2 s) when the ant was "caught." Two feedback scenarios were evaluated. The system without feedback relied solely on pre-calibrated threshold and impedance data, along with frequency-based current variations. While this method delivered participant-specific electrotactile feedback, it failed to fully compensate for variations in garment pressure, limiting consistency and precision. In this feedback system, feedback from integrated pressure sensors was incorporated into the stimulation control loop. This allowed the system to adapt the stimulation intensity based not only on individual impedance and frequency response, but also on real-time garment pressure (Fig. 6g). The result was a more stable, comfortable, and personalized electrotactile experience. In addition, the feedback mechanism enabled real-time detection and compensation for pressure fluctuations caused by body movement or differences in garment compression, further enhancing the reliability and safety of the system.

## Discussion

The textile-based electrical stimulation suit introduced in this study is lightweight, breathable, and resistant to dehydration. It offers seamless, flexible integration with the skin over large body areas, enabling rapid, real-time, and high-resolution tactile stimulation. By targeting whole-body skin interactions—an area largely underexplored in previous research—this system achieves high-density, spatiotemporally precise tactile feedback across nearly all anatomical regions while maintaining excellent breathability and effective sweat management. Crucially, the system accounts for individual variations in skin impedance and body morphology. Through experimental validation and computational modeling, we demonstrate enhanced efficiency in electrical tactile perception. A case study featuring a real-time, pressure-based feedback strategy further illustrates how the system dynamically adjusts stimulation parameters based on localized garment pressure. This holistic approach not only enables personalized tactile feedback across the entire body but also points to promising applications for full-body haptic interfaces.

As a result, the TESS platform opens new possibilities for immersive sensory experiences beyond conventional fingertip or palm-based stimulation. Its ability to deliver precise, adaptive stimulation over large skin areas with feedback control supports applications in entertainment, remote collaboration, and military simulation. These capabilities provide a pathway toward integrating electrical tactile feedback into virtual, augmented, and mixed reality systems, extending conventional haptic perception toward an artificial 'sixth sense.' The system's robust control architecture and its capacity to accommodate interindividual variability in tactile perception represent significant advances with broad implications. Beyond haptic feedback, the TESS device also holds potential for therapeutic electrical stimulation in tissue regeneration, wound healing, pain management, and cosmetic treatments. In integrated VR or AR settings, TESS could deliver realistic tactile feedback alongside therapeutic stimulation, highlighting its multifunctionality. As a next-generation wearable platform, TESS offers a durable, scalable, user-centric solution for consistent electrical stimulation—enhancing comfort and interaction. This approach enhances the usability of electrical stimulation in healthcare and rehabilitation and supports its adoption in immersive digital environments.

## Methods

### Fabrication of stretchable Ag-PU conductor ink
To create the viscous conductive ink, 0.5 g of PU (HydroMed™ D3, AdvanSource Biomaterials Corp.) was dissolved in 8 ml of tetramethylene oxide (34865, Sigma-Aldrich) and stirred continuously for 8 h. After complete dissolution, 7 g of Ag flakes (average diameter ≈1.3 μm, DSF-500 MWZ-S, Daejoo Electronic Materials Co., Ltd.) were added to the solution, and the mixture was further stirred for 4 h to achieve uniform dispersion.

### Stencil printing of textile permeable ink
The resulting Ag-PU ink was stencil-printed onto a stretchable textile substrate (87% polyester, 13% spandex; weight ≈190 g/m²). A stainless-steel stencil mask (SUS, thickness ≈100 μm), laser-cut to define the desired pattern, was used for printing. The viscous ink was applied onto the stencil, and a constant shear force was applied using a glass slide to ensure even deposition through the mask.

### Fabrication of the DMCH
The DMCH was synthesized by first mixing 22 g of deionized (DI) water, 10 g of PEDOT:PSS solution (PH1000, Heraeus), and 6.36 g of acrylamide (2-Propenamide, A8887, Sigma-Aldrich) for 2 h. Subsequently, 4 g of lithium chloride (L4408, Sigma-Aldrich), 0.038 g of ammonium persulfate (248614, Sigma-Aldrich), and 0.004 g of N,N'-methylenebisacrylamide (M7279, Sigma-Aldrich) were sequentially added, with each component mixed for 30 min. Finally, 3 drops of N,N,N',N'-tetramethylethylenediamine (T7024, Sigma-Aldrich) were added, and the mixture was quickly stirred before being poured into a mold and cured for 30 min.

### Assembly of multimodal electrical stimulation system
To assemble the textile-based electrical stimulation device, the cured DMCH hydrogel was laminated onto the textile substrate pre-patterned with the Ag-PU conductor ink. The assembly was then cured in an oven (OF-02PW, JEIO TECH) at 60 °C for 10 min. Electrical connections were established using silicone hook-up wires (Hook-up Wire Kit, CBAZY), which were bonded to the conductive traces with Ecoflex (Ecoflex 00–30, Smooth-On) and connected to an external control circuit (Supplementary Notes 7 and 8, and Supplementary Figs. 31 and 32) to complete the multimodal electrical stimulation system.

### Comparison of the area and weight of electrodes
The mechanical actuators evaluated in this study included various vibrotactile motors: an eccentric rotating mass (ERM) motor (C0720B003D, Jinlong Machinery & Electronics), a linear resonant actuator (LRA) (C10-100, Precision Microdrives™), an additional ERM (910-108.002, Precision Microdrives™), and other vibrotactile actuators referenced in [ref. 46]. Electrical stimulation electrodes included Ag/AgCl Monitoring Electrodes (2223H, 3 M), Ag/AgCl Disposable ECG Electrodes (L-150X, VitrodeL), TENS electrode-1 (LG Electro Pad, 26419561, LG Electronics Inc.), and TENS electrode-2 (PI-5500, Chunghoon Co., Ltd.), designed for both TENS and EMS applications.

### Air permeability measurement
Air permeability was evaluated under pressure levels of 100, 300, 600, and 900 Pa using a standard air permeability tester (FX 3300, Textest Instruments). Each sample measured 15 cm × 15 cm.

### Dehydration test
Dehydration behavior was assessed for three hydrogel types: (1) a control hydrogel lacking both LiCl and PEDOT:PSS, prepared by replacing these components with an equivalent amount of DI water; (2) a PEDOT:PSS hydrogel, which omitted only LiCl; and (3) the full DMCH formulation. Eight samples of each hydrogel type were tested over a 24 h period at room temperature (21.3 °C, 66.3% humidity). Samples were placed on individual microscope glass slides (Marienfeld Superior), and weight measurements were recorded at two-hour intervals. Final calculations excluded the weight of the glass slides.

### Human participant sensory perception test
A total of 15 volunteers of diverse genders, aged in their 20 s to 30 s (mean age: 25.95 years; three females; Supplementary Fig. 36c), participated in the electrical stimulation study. All participants underwent current threshold measurements, recognition confusion assessments, and classification of four distinct tactile sensations. Of these, nine participants were additionally involved in evaluating perceptual threshold currents across a range of frequencies. Threshold voltage and current measurements were collected from the anterior region of the left forearm using a digital multimeter (DMM751, Keithley Instruments). Before testing, participants were seated in a relaxed position to minimize external influences on perception. To prevent skin fatigue and ensure safety during repeated electrical exposure, participants were given a 30 min rest period after every 20 min stimulation session.

The perceptual threshold was defined as the minimum current intensity required to induce a distinct, recognizable sensation without causing discomfort or stinging. For the sensation classification task, participants were not informed of the electrical parameters associated with each of the four sensations, ensuring an unbiased assessment based on subjective perception. In the recognition confusion assessment, participants were first exposed to a single presentation of each sensation at its median stimulation intensity (as determined in the prior classification task), followed by a rest period. Subsequently, the sensations were applied in random order while participants were blindfolded and asked to identify the perceived sensation. No hypothesis testing was performed (one-/two-sided not applicable). All procedures involving device testing and sensory data collection were conducted under protocols approved by the institutional review board, with informed consent obtained from all participants. The perception experiments were performed at Hanyang University (approved by its Institutional Review Board, no. HYUIRB-202511-004). Informed consent was obtained from all participants prior to the experiments, and no compensation was provided for their participation.

## EDS-mapping

EDS-mapping images were obtained using a field-emission scanning electron microscope (FE-SEM; JSM-7610F-Plus, JEOL). Prior to imaging, samples were cryogenically frozen with liquid nitrogen and sectioned using a precision blade to expose clean cross-sectional surfaces.

## Fabrication of the clothing pressure sensor

The pressure sensor was created by screen-printing Ag-PU conductive ink onto a textile substrate using a stencil designed with an inter-digitated electrode pattern. A convex-shaped conductive gel layer was then applied facing the printed ink surface. To improve the signal-to-noise ratio (SNR), the gel was formulated using half the PEDOT:PSS concentration found in the DMCH formulation. Subsequently, a PU-based textile that was not coated with ink was placed on the top and sewn for encapsulation. The change in resistance according to the force of the completed pressure sensor was recorded by placing a hydrogel that acted as an artificial skin underneath and placing a load on it. The pressure sensor was evaluated using the normalized resistance change ($\Delta R_{norm}$), defined as:

$$\Delta R_{norm} = \frac{R - R_0}{\bar{R}_0} \tag{1}$$

where $R$ is the measured resistance, $R_0$ is the initial resistance, is the stabilized resistance value in a weightless state, and $\bar{R}_0$ is the average initial resistance. The average value of $\bar{R}_0$ was determined to be 833.98924 $\Omega$.

## Implementation of feedback using clothing pressure

To implement the feedback system, we first measured the threshold current at various positions along the arm, as well as the actual current flowing through the skin while the sleeve was worn. Based on these measurements, a linear fitting model was established, assuming a direct relationship between applied voltage and skin current, as illustrated in Supplementary Fig. 29b. The voltage–current relationship, taking into account participant-specific skin impedance, was defined by the following equation:

$$v = 12.2137 \, I + 4.4519 \, \text{V} \tag{2}$$

where $I$ is the current and $v$ is the voltage (Supplementary Table 3a).

The additional current due to pressure is defined by an expression based on the normalized resistance change ($\Delta R_{norm}$) (Supplementary Fig. 26 and Supplementary Table 3b):

$$I_{pressure} = 1.54 \, R_{norm} - 0.2 \, \text{mA} \tag{3}$$

Thus, the required current is:

$$I_{total} = I_{th} + I_{freq} - I_{pressure} \, \text{mA} \tag{4}$$

The total required current ($I_{total}$) is defined as the sum of the inherent threshold current (which accounts for the participant's skin impedance) and the additional current required based on the frequency ($I_{freq}$) minus the additional current induced by the pressure ($I_{pressure}$). $I_{freq}$ considering the average differences in the threshold current across frequencies, as shown in Fig. 4d.

## Virtual environment and visualization

The VR scenes were implemented by the authors in Unity. The visual assets (e.g., props such as a tomato, desk, human avatar, and explosion effects) were obtained from the Unity Asset Store. The hand model/hand-tracking visualization was implemented using the Meta Quest SDK/library. All other elements, including the experimental logic and scene composition, were created by the authors.

## Reporting summary

Further information on research design is available in the Nature Portfolio Reporting Summary linked to this article.

## Data availability

The data generated in this study are provided in the Supplementary Information/Source Data file. Source data are provided in this paper.

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

## Acknowledgements

This work was supported by the National Research Foundation of Korea (NRF) grants NRF-2022R1C1C1003994 and RS-2024-00411764 funded by the Ministry of Science (MSIT) of the Korean Government. This work was supported by the Institute of Information & Communications Technology Planning & Evaluation (IITP) under the artificial intelligence semiconductor support program to nurture top talents grant IITP-(2026)-RS-2023-00253914 funded by MSIT. This work was supported by the "Regional Innovation System & Education (RISE)" through the Seoul RISE Center grant 2026-RISE-01-027-04 funded by the Ministry of Education (MOE) and the Seoul Metropolitan Government.

## Author contributions

J.H.H., S.H.K., J.-H.K. and Y.H.J. conceived the study, designed the research, analyzed the data, and wrote the manuscript. J.H.H., S.H.K. and J.-H.K. designed the device and fabricated it. J.Y.Y. and J.-H.K. designed the operational protocols and graphical user interfaces. J.H.H. and S.H.K. performed experimental validation and analysis. J.H.H., S.H.K., J.-H.K., J.-Y.Y., J.S., G.C., J.M.L., B.C., S.P., J.K., S.M.W., J.K., D.-W.P. and Y.H.J. carried out the technical revisions of the manuscript.

## Competing interests

The authors declare no competing interests.

## Additional information

[1]Department of Electronic Engineering, Hanyang University, Seoul, Republic of Korea. [2]Department of Chemical Engineering, University of Seoul, Seoul, Republic of Korea. [3]School of Electrical and Computer Engineering, University of Seoul, Seoul, Republic of Korea. [4]Center for Semiconductor Research, University of Seoul, Seoul, Republic of Korea. [5]Department of Semiconductor Convergence Engineering, Sungkyunkwan University, Suwon, Republic of Korea. [6]Department of Artificial Intelligence Semiconductor Engineering, Hanyang University, Seoul, Republic of Korea. [7]Department of Electrical and Computer Engineering, Ajou University, Suwon, Republic of Korea. [8]Department of Chemical and Biomolecular Engineering, Yonsei University, Seoul, Republic of Korea. [9]Department of Electrical and Computer Engineering, Sungkyunkwan University, Suwon, Republic of Korea. [10]Institute of Nano Science and Technology, Hanyang University, Seoul, Republic of Korea. [11]These authors contributed equally: Jin Hee Hwang, Sun Hong Kim, Ju-Hwan Kim, Jae-Young Yoo. ✉e-mail: jkim448@hanyang.ac.kr; dwpark31@uos.ac.kr; yjung@hanyang.ac.kr

