## [Transparent Peer Review file · Nature Communications]

A Lightweight, Durable Full-Body Electrical Stimulation Suit for Haptic Feedback and Therapeutic Applications

Corresponding Author: Professor Yei Hwan Jung

Version 0:

Reviewer comments:

Reviewer #1

(Remarks to the Author)

Hwang et al. introduced a novel textile-based full-body electrical stimulation suit (TESS) designed for immersive haptic feedback and therapeutic applications. The suit integrates lightweight, breathable materials with soft, stretchable electrodes and a real-time calibration system, ensuring consistent stimulation across diverse body types and conditions. By delivering electrotactile sensations, TESS shows certain applications in XR environments and rehabilitation therapies. There are some questions and comments.

1. TESS was equipped with an integrated real-time calibration system that dynamically monitors garment pressure to ensure consistent tactile feedback across the body. However, in Fig. 1b and Supplementary Fig. 23, pressure sensors and stimulators are placed on different sites. Considering the irregular curvature of human skin, can measurement precisely indicate the pressure on the stimulators?
2. Does the Ag-PU conductive layer form an electrical connection with the skin during use. Could this connection impact the electrical stimulation process or alter the stimulation parameters?
3. The study reported a conductivity of 5×10^3 S/cm for the ink (80% Ag volume fraction). How does this compare to other similar inks? What is the absolute resistance of the conductive layer in the constructed TESS?
4. The authors considered the area of DMCH, but it is unclear how the thickness of DMCH was determined. Additionally, does the thickness of DMCH influence the electrical properties of the electrode or the contact pressure correction applied in this study?
5. Fig. 3e compares the weight and area of DMCH electrodes with mechanical actuators and commercial TENS electrodes. However, the source of the commercial electrodes is not specified and should be clarified. Since these comparison parameters mainly relate to electrode density, comparing the DMCH electrodes with widely reported textile-based electrodes would more effectively highlight their advantages. Additionally, as electrical performance is a key consideration, it should also be included in the comparison for a more comprehensive evaluation.
6. The authors describe the effect of inter-electrode distance on perception threshold (Supplementary Fig. 11). The reasons for this are unclear and require further explanation. Furthermore, the authors also need to clarify how the center-to-center distance between the anode and cathode was determined and how this affected the experimental results.
7. Fig. 5e shows the resistance variation of the pressure sensor at arm positions 1 to 4. What is the range of pressure variation caused by individual body shape differences and positional variations between wears at different positions on the human body? Can all of this be detected by the reported sensor?
8. In Fig. 5g, the authors adjusted the stimulation voltage based on pressure data to compensate for individual differences, aiming to achieve consistent sensory perception across users. However, the amount of compensation requires further statistical analysis to validate its effectiveness. Additionally, is there any existing research supporting the possibility of achieving consistent electrical stimulation effects by standardizing the current output?
9. In the applications of the TESS, it is important to evaluate the impact of the integrated calibration system based on pressure on the effectiveness of various applications. However, the current experiments do not appear to address this aspect.
10. The study reported the reconstruction of various tactile scenarios using four specific stimulation frequencies. However, the mechanism linking these frequencies to distinct tactile perceptions remains unclear. Does it simply demonstrate that users can differentiate between electrical stimulations of varying frequencies? Additionally, as noted in row 397, many participants were confused by 70 Hz and 120 Hz, associating them with pressure and roughness. Could this confusion be attributed to human insensitivity or reduced perceptual resolution within this frequency range?

11. The user study should include detailed information on the age and gender distribution of the participants. This is important as such factors can influence tactile perception.

12. There are a few minor issues that should be addressed. There seems to be a watermark in the top left corner of Fig. 2f. Additionally, stratum corneum belongs to epidermis, so the skin model in Fig. 4a needs to be adjusted. A scale bar is preferred in Supplementary Fig. 16.

Reviewer #2

(Remarks to the Author)

This manuscript presents a full-body stimulation suit with good performance and mechanical/electrical properties and complementary study on the practical application, which is interesting and has great application potential in medical and extended reality fields. However, the scientific novelty is not clear enough. Is the novelty of this work in material design, stimulation performance, or whole wearable system design? There are many works on textile wearable electronics, and what is the key innovation of this work? What is the scientific point of the key innovation? I suggest addressing the following questions for further consideration of its suitability for publication.

1. Please specify the advantage of the Ag-PU and DMCH electrode. How about the environmental stability of the DMCH electrode? Such as with long-term exposure in air, high temperature/humidity, etc.
2. Biocompatibility of the materials should be discussed, for instance, in the introduction.
3. The authors studied the sweat effect on the current and stimulation, based on the 10-min running condition. It will be better if the quantitative evaluation can be done to study the effect of sweat.
4. As an electronic cloth, how is the abrasion resistance, and is it washable?
5. Textile is a type of material with complex features in material and structure, how is the printing accuracy of the conductive array on textiles? Any solutions could improve it and enhance the function of the electrical-stimulation suit? Please cite some important references these two years and illustrate the scientific points.
6. Is this electrode recipe and system design applicable to other textile substrates that have diverse materials, or different woven structures? Please show some examples and discuss.
7. In practical application, there will be displacements between the cloth and human skin to affect the electrical stimulation and feedback effect, how to avoid it?
8. How is the feedback accuracy and efficiency of this electrical stimulation suit with other reported works? It is better to make a figure or table for quantitative comparison.
9. Line 200, is there any evidence or reference for the hydrogen bond between DMCH and Ag-PU?
10. Regarding to line 139, will the water absorption and hydrophilic property of HydroMed cause the oxidation and degradation of Ag particles? Maybe the resistance of the printed electrode can be measured after keeping in a wet condition for a relatively long time to see the durability.
11. Line 47, XR and should be defined at its first appearance in the main text (not only defined in the abstract).
12. In Figure 2a, the red area in the SEM is not very accurate, and the Ag-PU electrode layer has not been labeled. It is suggested to present Figure 2 in three lines. Additionally, the scale bars in Figure 1b, 2a, 2d, 5c should be added. In Figure S2, what is the element that each color represents in the EDS?

Version 1:

Reviewer comments:

Reviewer #1

(Remarks to the Author)

The authors have addressed all my concerns. It's a very interesting work by using full body ES. I recommend the acceptance of the manuscript as is.

Reviewer #2

(Remarks to the Author)

The authors have addressed my previous comments and improved the manuscript. However, there are still some writing and formatting issues in the revised manuscript that should be addressed to ensure a high-quality presentation, meeting the high standards of Nat. Commun., Please see the issues as follows.

1. Too many references are cited to support the revision, which highly reduces the continuity of the manuscript. It is suggested to concisely integrate the revisions in the text, and relocate the current detailed optimization-related discussion and the corresponding references to Supporting Information, such as the points as follows:
 - (1) The discussion of printing accuracy (pages 6-7);
 - (2) The conductivity comparison of Ag-PU conductors with other Ag-based inks for e-textiles (page 8);
 - (3) The optimization of the DMCH electrode thickness (page 11);
2. The subtitles in Results can be refined to improve the narrative logic of the entire article, for instance, the section title of "Material characterization" on page 6 is more suitable to occur in the Experimental Section. More accurate titles closely related to the Results and Discussion should be used.
3. The Conclusion is not concise enough. Please improve it.
4. The Methods Section contains too much content, making it hard to get the important experimental details. It is suggested

to retain experiments related to the e-textile assembly and its crucial performance evaluation. All other details should be moved to the Supporting Information.

5. The scale bar unit should be μm instead of um , for example, Figure S8

6. The numbering in Figure S29 and Figure S33 is inconsistent with that in the figure caption. Please carefully check.

Response to Referees Letter

Referee #1:

Summary Comments: *Hwang et al. introduced a novel textile-based full-body electrical stimulation suit (TESS) designed for immersive haptic feedback and therapeutic applications. The suit integrates lightweight, breathable materials with soft, stretchable electrodes and a real-time calibration system, ensuring consistent stimulation across diverse body types and conditions. By delivering electrotactile sensations, TESS shows certain applications in XR environments and rehabilitation therapies. There are some questions and comments.*

Our response:

We thank the reviewer for pointing out the key contributions of this manuscript and for providing insightful comments that are helpful in improving our work. We have addressed all of these comments, as listed below and in revisions to the manuscript.

Comment #1: *TESS was equipped with an integrated real-time calibration system that dynamically monitors garment pressure to ensure consistent tactile feedback across the body. However, in Fig. 1b and Supplementary Fig. 23, pressure sensors and stimulators are placed on different sites. Considering the irregular curvature of human skin, can measurement precisely indicate the pressure on the stimulators?*

Our response (Comment #1): We sincerely thank the reviewer for this insightful comment. In our TESS, each stimulator is paired with a pressure sensor placed adjacent to the stimulation site to monitor the local garment pressure. We acknowledge that, in the original manuscript, the photographs shown in the Supplementary Fig. 33 (previously Supplementary Fig. 23, now Supplementary Fig. 33 in the revised manuscript) did not match the schematic illustrations in Fig. 1b. In the revised Supplementary Fig. 33, we have replaced the photographs so that they correspond to the layout illustrated in Fig. 1b.

Furthermore, to minimize any residual mismatch between the stimulation site and the measured local pressure, we redesigned the sleeve layout to position the stimulator and pressure sensor in closer proximity than in the original design. Specifically, the center-to-center distance was reduced from ~50 mm to a minimum of 20 mm after redesign (a 60 % reduction).

In the revised manuscript, we also report results from an additional experiment designed to assess whether a single pressure reading can reliably represent the garment pressure around a given arm cross-section. We selected four cross-sectional positions along the arm—position 1 at the wrist (smallest circumference) through position 4 at the upper arm (largest circumference), so that arm circumference increased monotonically from position 1 to position 4. At each position, we measured garment pressure at four circumferential locations (anterior, medial/inner, posterior, and lateral/outer). Across all positions, the pressure variation within a given cross-section was small. The largest within-section deviation occurred at the wrist (Supplementary Fig. 34), but even there the variability remained within the sensor tolerance (standard deviation = 0.028 at position 1). Accordingly, these small differences are not expected to produce a meaningful change in perceived stimulation intensity.

These findings support our assumption that circumferential pressure differences at a given arm position are negligible. This interpretation is also consistent with prior reports. Verzwylt *et al.* (2023) showed that medial–lateral differences are typically within 10–20 % of the overall pressure and that average within-cross-section differences are generally small, while Mitsuno *et al.* (2023) reported near-uniform pressure distribution along a given arm circumference. Therefore, in practical setting, a single-point measurement is commonly used as a

representative value for a cross-section^{R3, R4}.

We agree with the reviewer that garment pressure can exhibit circumferential non-uniformity within the same arm cross-section. Accordingly, we have revised the manuscript to explicitly acknowledge this potential non-uniformity and to clarify that our measurements show the within-cross-section variation remains within the sensor tolerance (Fig. 5c), and is therefore unlikely to meaningfully affect the perceived stimulation intensity.

- R1. Verzwylt, Z. J. *et al.* Optimising application of a two-layer compression bandage using continuous, multi-point sub-bandage pressure monitoring in healthy volunteers. *Journal of Wound Management* **24** (2023).
- R2. Mitsuno, T. T. Gradient of clothing pressure for comfortable support wear. *Journal of Textile Engineering & Fashion Technology* **9**, 86-89 (2023).
- R3. Wicaksono, I. *et al.* A tailored, electronic textile conformable suit for large-scale spatiotemporal physiological sensing in vivo. *npj Flexible Electronics* **4**, 1-13 (2020).
- R4. Zhao, L., Li, X., Yu, J., Li, C. & Li, G. Compression sleeves design based on Laplace laws. *Journal of Textile Engineering & Fashion Technology* **2**, 314-320 (2017).

Authors' modification to the manuscript:

(Page 38-39 in Revised Supplementary Information; add Supplementary Fig. 33 and 34)

Supplementary Fig. 33 | Photograph of a participant wearing a forearm-based TESS system featuring an array of four electro tactile stimulator and pressure sensor pairs printed on a sleeve-type pressure garment

a, A photograph of the garment worn on a participant with a thinner arm, resulting in relatively low contact pressure measured by the sensors. **b**, A photograph of the same garment worn on a participant with a thicker arm, resulting in relatively high contact pressure. Scale bar, 20 mm.

Supplementary Fig. 34 | Pressure variation analysis across the arm cross-section.

a, Pressure was measured at four circumferential locations (anterior, medial/inner, posterior, and lateral/outer) using the pressure sensor. **b**, Normalized resistance change (ΔR_{norm}) and arm circumference measured at four arm positions (1–4). The circumferential distances from the anterior location toward the inner direction to the remaining sensor locations were measured for each position (1: 0, 30, 70, 100 mm; 2: 0, 40, 95, 150 mm; 3: 0, 50, 100, 160 mm; 4: 0, 70, 120, 170 mm).

(Page 3; line 32 in Revised Supplementary Information; add Supplementary Note 1)

“For demonstration purposes, an electrode-array-based electrotactile device was mounted on the arm (Supplementary Fig. 33a,b). Each stimulator is paired with a pressure sensor placed adjacent to the stimulation site to monitor the local garment pressure. To assess whether a single-point pressure reading can represent the garment pressure around a given arm cross-section, we conducted measurements along the arm at four cross-sectional positions (from the wrist with the smallest circumference to the upper arm with the largest circumference). At each position, garment pressure was recorded at four circumferential locations (anterior, medial/inner, posterior, and lateral/outer). Across all positions, the within-cross-section variability was small; the largest deviation was observed at the wrist, yet the variability remained within the sensor tolerance (standard deviation = 0.028 at position 1; Supplementary Fig. 34a,b). These differences are not expected to produce a meaningful change in perceived stimulation intensity, supporting the use of a single-point pressure measurement as a representative value for garment pressure within a cross-section^{R3, R4}. A robust skin–electrode interface reduces contact impedance, allowing more consistent and effective tactile feedback.”

Comment #2: *Does the Ag-PU conductive layer form an electrical connection with the skin during use. Could this connection impact the electrical stimulation process or alter the stimulation parameters?*

Our response (Comment #2):

We thank the reviewer for this valuable comment. In our current design, the Ag-PU textile conductor does not form intimate, sustained contact with the skin because the textile surface is intrinsically irregular and lacks adhesive characteristics. This limits interfacial charge transfer and increases susceptibility to motion-induced artifacts, thereby reducing effective current delivery. In addition, the Ag-PU conductor is embedded within the textile architecture, which further constrains conformal skin contact. In contrast, the DMCH layer has an extremely low modulus comparable to that of skin, enabling more reliable and conformal contact at the skin–electrode interface.

To further investigate this point, we performed additional measurements and consistently

observed low current densities for the Ag-PU conductor across all tested frequencies, with values of ~ 0.015 mA/cm² at 1 Hz, ~ 0.014 mA/cm² at 10 Hz, and ~ 0.015 mA/cm² at 100 Hz. This additional finding was included in Fig. 2f, in the revised manuscript. By comparison, the DMCH exhibited a substantially higher current density at 1 Hz—approximately 23.3× greater than that of the Ag-PU conductor. These results indicate that the Ag-PU layer delivers currents below the perceptual threshold and is therefore unlikely to elicit a noticeable sensation.

Based on these findings, we conclude that an additional encapsulation step for the Ag-PU conductor is not necessary. Moreover, encapsulation would likely compromise the breathability of the textile, which is a key design consideration for wearable use.

Authors' modification to the manuscript:

(Page 11; line 250 in Revised Manuscript)

“To compare the individual and combined effects of ionic salt and conductive polymer on electrode performance, current densities were measured for four electrode types: **a Ag-PU conductor**, a pure hydrogel, an ionic hydrogel incorporating LiCl to enhance ionic conductivity, and the DMCH formulation containing both LiCl and PEDOT:PSS (Fig. 2f). **Although the Ag-PU conductor is directly coated on the textile for interconnects, it does not produce electro-tactile sensations on the skin. When used as a standalone electrode, it delivers substantially lower current than hydrogel-based electrodes because the Ag-PU interface exhibits much higher skin–electrode impedance than wet hydrogel contacts, limiting charge transfer and effective current delivery. As a result, the current density remains below the perceptual threshold and is not expected to elicit a noticeable sensation. In contrast, current density increased progressively from the pure hydrogel, to the ionic hydrogel, and reached its maximum in DMCH, thus allowing efficient current delivery.**”

(Page 57 in Revised Manuscript)

Fig. 2 | Characterization of the electrical stimulation electrode

a, Schematic diagram of a single electrical stimulator showing the DMCH and the Ag-PU

conductor printed on a PU-based textile. **b**, Cross-sectional SEM image showing seamless integration of the DMCH on the Ag-PU conductor printed onto a PU-based textile. No interfacial gaps were observed. Scale bars, 100 μm . **c**, Schematic showing the mechanism of robust adhesion between DMCH and the Ag-PU conductor. **d**, Photographs of the peeling test used to compare the peeling force of the DMCH/Ag-PU conductor with that of the DMCH/silicon elastomer. **e**, Peeling forces per width of DMCH attached to the Ag-PU conductor and silicon elastomer. **f**, **Current density measured on skin as a function of frequency for the Ag-PU conductor (used as the interconnect) and three electrical stimulation electrodes with different hydrogel compositions.** **g**, Stress as a function of strain for conductive hydrogel (red), hydrogel (black), and ionic hydrogel (blue). The DMCH is capable of withstanding strains of up to 600 %. **h**, Impedance spectra of various hydrogel electrodes with different compositions: pure hydrogel (black), ionic hydrogel (blue), and DMCH (red). The DMCH formed by mixing PEDOT and ions exhibited the lowest impedance, owing to the doping effect of the ions on PEDOT.

Comment #3: *The study reported a conductivity of 5×10^3 S/cm for the ink (80% Ag volume fraction). How does this compare to other similar inks? What is the absolute resistance of the conductive layer in the constructed TESS?*

Our response (Comment #3):

We thank the reviewer for this comment. To benchmark the performance of our Ag-PU conductor, we have added Supplementary Table 1, which compares its conductivity with previously reported textile-based Ag inks. The conductivity of our Ag-PU conductor is comparable to representative values reported in the literature. Importantly, beyond conductivity alone, our system introduces a new class of wearable haptic interface technology that addresses several unmet needs, as summarized below.

1. Robust interfacial adhesion between metallic composite and ionic electrodes.

Most reported Ag composites employ hydrophobic elastomeric binders such as Ecoflex, fluorinated rubber, or PDMS^{R1-R3}. Because these matrices have limited affinity for hydrophilic hydrogels, stable integration often requires additional interfacial treatments (e.g., benzophenone/UV activation^{R1}); otherwise, the hydrogel layer can delaminate under mechanical loading. In contrast, our Ag-PU conductor enables strong hydrogel bonding without such surface treatments. Robust interfacial coupling is achieved through a simple thermal bonding step (60 °C for 10 min), which promotes hydrogen-bond-mediated interactions and suppresses delamination even under strains up to 270 %.

2. Room-temperature process and high throughput.

Many previously reported high-performance textile conductors require high-temperature curing/sintering and/or multi-step, complex patterning and transfer processes^{R4,R5}. In contrast, our Ag-PU conductor is fabricated via a room-temperature, high-throughput printing process without high-temperature post-processing, providing a streamlined and scalable route toward textile-integrated wearable systems.

We also evaluated the absolute resistance of the conductive layer. Five samples were fabricated for each pattern width (1, 2, and 3 mm) at a fixed pattern length of 30 mm, and the absolute resistance was measured (Supplementary Fig. 5). As expected, the resistance decreased with increasing pattern width, with average values of 3.62 Ω (1 mm), 1.90 Ω (2 mm), and 1.54 Ω (3 mm).

In conclusion, while the conductivity of our Ag-PU conductor is comparable to that of representative textile-based Ag inks reported in prior work, our design addresses practical constraints that are central to garment-scale deployment, including robust interfacial adhesion, textile integration, and process simplification.

- R1. Kim, S. H. *et al.* Ultrastretchable conductor fabricated on skin-like hydrogel–elastomer hybrid substrates for skin electronics. *Advanced Materials* **30**, 1800109 (2018).
- R2. Jin, H. *et al.* Highly durable nanofiber-reinforced elastic conductors for skin-tight electronic textiles. *Acs Nano* **13**, 7905-7912 (2019).
- R3. Matsuhisa, N. *et al.* Printable elastic conductors with a high conductivity for electronic textile applications. *Nature communications* **6**, 7461 (2015).
- R4. Zhu, C.-H., Li, L.-M., Wang, J.-H., Wu, Y.-P. & Liu, Y. Three-dimensional highly conductive silver nanowires sponges based on cotton-templated porous structures for stretchable conductors. *RSC Advances* **7**, 51-57 (2017).
- R5. Lv, J. *et al.* Printed sustainable elastomeric conductor for soft electronics. *Nature Communications* **14**, 7132 (2023).

Authors' modification to the manuscript:

(Page 8; line 165 in Revised Manuscript)

“Given that skin stretchability varies across the body—reaching up to 70 % in joint regions^{25,26}—an Ag-PU conductor with 80 % Ag content, offering 200 % stretchability, is sufficient to ensure robust and reliable performance in wearable electronic applications. **Moreover, cyclic tensile testing at 30 % strain confirmed the mechanical and electrical stability of the optimized Ag-PU conductor over 1,000 cycles (Supplementary Fig. 4d). Supplementary Figure 5 shows the absolute resistance of the printed Ag-PU conductors as a function of pattern geometry. For the 2 mm trace width used throughout this study, a 30 mm-long conductor exhibits an average resistance of ~1.9 Ω. We also benchmarked the conductivity of our Ag-PU conductors against previously reported Ag-based inks for textile electronics (Supplementary Table 1) confirming that our conductivity is comparable to representative textile-based Ag inks reported in prior work. Beyond conductivity, our ink enables hydrogen-bond-mediated adhesion to hydrophilic hydrogels without the additional surface treatments often required for hydrophobic Ag composites^{R1–R3}, and it can be fabricated without high-temperature post-processing, supporting scalable integration in textile-based wearable systems.** To evaluate durability, 13 Ag-PU conductor patterns (30 mm × 2 mm) were fabricated on the textile and subjected to a washing test consisting of four 10-min cycles under magnetic stirring in water at 650 rpm (Supplementary Fig. 6).”

(Page 10 in Revised Supplementary Information; add Supplementary Fig. 5)

Supplementary Fig. 5 | Absolute resistance of Ag-PU conductors with varying printed patterns

Absolute resistance of 30 mm-long Ag-PU conductors as a function of printed pattern width (1, 2, and 3 mm).

(Page 42; in revised Supplementary Information; add Supplementary Table. 1)

reference	Materials	Conductivity (S cm ⁻¹)	Heat treatment
[R1]	Ag nanowire/ polydimethylsiloxane	56.82	100 °C for 2h
[R2]	Ag flake/ vegetable oil-based polyurethane	12,833	60 °C for 3h
[R3]	Ag nanowire/ thermoplastic polyurethane	5,114	120 °C for 30min
[R4]	Ag nanowire/ thermoplastic polyurethane	3,668	80 °C
[R5]	Ag nanowire/ double-covered yarn (cotton fiber)	4,018	80 °C for 4h
[R6]	Ag nanoparticle/ polydopamine	4,058	50 °C for a day
[R7]	Poly(ethylene glycol) methyl ether appended to thioctic acid-modified AgNWs/ thermoplastic polyurethane	14,205	155 °C for 30min
[R8]	Ag nanoparticle/ poly (styrene-block-butadiene-block-styrene)	5,400	No
[R9]	Ag powder/ poly(vinylidene fluoride-co-hexafluoropropylene)	3,246	No
[R9]	Ag flake/ poly(vinylidene fluoride-co-hexafluoropropylene)	3,116	No
This work	Ag flake/ thermoplastic polyurethane	5,411	No

Supplementary Table 1 | Conductivity comparison of Ag-based textile inks.

Reported conductivities of Ag-based inks on textile substrates, benchmarked against this work.

- R1. Zhu, C.-H., Li, L.-M., Wang, J.-H., Wu, Y.-P. & Liu, Y. Three-dimensional highly conductive silver nanowires sponges based on cotton-templated porous structures for stretchable conductors. *RSC Advances* **7**, 51-57 (2017).
- R2. Lv, J. *et al.* Printed sustainable elastomeric conductor for soft electronics. *Nature Communications* **14**, 7132 (2023).
- R3. Zhao, H. *et al.* Ultrastretchable and washable conductive microtextiles by coassembly of silver nanowires and elastomeric microfibers for epidermal human-machine interfaces. *ACS Materials Letters* **3**, 912-920 (2021).
- R4. Zhu, H.-W. *et al.* Printable elastic silver nanowire-based conductor for washable

- electronic textiles. *Nano Research* **13**, 2879-2884 (2020).
- R5. Cheng, Y., Wang, R., Sun, J. & Gao, L. Highly conductive and ultrastretchable electric circuits from covered yarns and silver nanowires. *ACS nano* **9**, 3887-3895 (2015).
- R6. Niu, B., Hua, T. & Xu, B. Robust deposition of silver nanoparticles on paper assisted by polydopamine for green and flexible electrodes. *ACS Sustainable Chemistry & Engineering* **8**, 12842-12851 (2020).
- R7. Lu, Y. *et al.* High-performance stretchable conductive composite fibers from surface-modified silver nanowires and thermoplastic polyurethane by wet spinning. *ACS applied materials & interfaces* **10**, 2093-2104 (2018).
- R8. Park, M. *et al.* Highly stretchable electric circuits from a composite material of silver nanoparticles and elastomeric fibres. *Nature nanotechnology* **7**, 803-809 (2012).
- R9. La, T. G. *et al.* Two-layered and stretchable e-textile patches for wearable healthcare electronics. *Advanced healthcare materials* **7**, 1801033 (2018).

Comment #4: *The authors considered the area of DMCH, but it is unclear how the thickness of DMCH was determined. Additionally, does the thickness of DMCH influence the electrical properties of the electrode or the contact pressure correction applied in this study?*

Our response (Comment #4):

We appreciate the reviewer's valuable comments, which address a critical aspect of our DMCH electrode design, the thickness optimization. The optimized thickness was determined by correlating electrical stability with its mechanical integrity under compressive loading.

1. Thickness-dependent electrical stability of DMCH under pressure

To quantify the thickness-dependent response of DMCH under compression, we used a universal testing machine (UTM) to compress DMCH samples with thicknesses of 1, 2, 3, and 5 mm up to 5 N while simultaneously measuring the through-thickness (vertical) resistance. As shown in Supplementary Figure 11, the 1 mm-thick DMCH became electrically unmeasurable beyond 1.3 N because the layer was too thin to maintain mechanical integrity under higher compressive loads. In contrast, DMCH samples with thicknesses ≥ 2 mm remained mechanically intact and electrically stable under forces up to 5 N, providing a robust conduction path across the full operating range. Notably, the 2 mm-thick DMCH exhibited the smallest resistance change at 5 N, which is important for electrodes subjected to sustained garment pressure. By comparison, the 3 mm- and 5 mm-thick DMCH samples showed higher absolute resistance and greater thickness, which can compromise wear comfort and conformability in garment-integrated implementations.

2. Thickness-dependent conductivity and stabilization threshold

We examined the intrinsic electrical properties of DMCH as a function of thickness. As shown in Supplementary Figure 12, the electrical characterization indicates a clear stabilization threshold. The 1 mm-thick DMCH exhibits substantially lower conductivity ($\sigma \approx 0.025 \text{ mS}\cdot\text{cm}^{-1}$) compared with the thickness-insensitive regime observed for thicker DMCH ($\sigma \approx 0.12\text{--}0.14 \text{ mS}\cdot\text{cm}^{-1}$). This behavior is consistent with prior reports on PEDOT:PSS-based electrodes, where the establishment of continuous conducting pathways and the relative contribution of interfacial resistances can be thickness dependent. In thin layers, an insufficiently percolated network of PEDOT-rich domains, combined with a larger relative influence of interfacial resistances, can lead to an apparently reduced conductivity^{R1–R6}. In contrast, the conductivity stabilizes from 2 mm onward, indicating that 2 mm is the minimum thickness required to form a robust and reliable conductive path. Based on these combined

mechanical and electrical results, we selected 2 mm-thick DMCH as the optimal thickness, as it ensures electrical reliability under compressive loading up to 5 N (Supplementary Figs. 11 and 12).

We observed that, even with the electrode thickness fixed at 2 mm, the measured current still varied with clothing pressure (Supplementary Fig. 26). Accordingly, our system monitors contact pressure and updates the stimulation parameters to support consistent tactile feedback. We have revised the manuscript to clarify the thickness-dependent electrical stability and characteristics of DMCH and to describe how these effects are incorporated into the contact-pressure correction.

- R1. Maeda, R. *et al.* The conducting fibrillar networks of a PEDOT: PSS hydrogel and an organogel prepared by the gel-film formation process. *Nanotechnology* **32**, 135403 (2021).
- R2. Greczynski, G., Kugler, T. & Salaneck, W. Characterization of the PEDOT-PSS system by means of X-ray and ultraviolet photoelectron spectroscopy. *Thin Solid Films* **354**, 129-135 (1999).
- R3. Andrei, V. *et al.* Size Dependence of Electrical Conductivity and Thermoelectric Enhancements in Spin-Coated PEDOT: PSS Single and Multiple Layers. *Advanced Electronic Materials* **3**, 1600473 (2017).
- R4. Nardes, A., Kemerink, M. & Janssen, R. Anisotropic hopping conduction in spin-coated PEDOT: PSS thin films. *Physical Review B—Condensed Matter and Materials Physics* **76**, 085208 (2007).
- R5. Carter, J. L., Kelly, C. A., Marshall, J. E. & Jenkins, M. J. Effect of thickness on the electrical properties of PEDOT: PSS/Tween 80 films. *Polymer Journal* **56**, 107-114 (2024).
- R6. Dijk, G., Ruigrok, H. J. & O'Connor, R. P. Influence of PEDOT: PSS coating thickness on the performance of stimulation electrodes. *Advanced Materials Interfaces* **7**, 2000675 (2020).

Authors' modification to the manuscript:

(Page 10; line 235 in Revised Manuscript)

“For instance, the DMCH–Ag-PU laminate withstood stretching up to 2.7 times its original length without delamination, whereas the Ag–silicone interface exhibited weak adhesion and early failure (Fig. 2e).

The DMCH electrode thickness was optimized by balancing mechanical integrity and electrical stability under compression. As shown in Supplementary Figure 11, the 1 mm DMCH could not sustain stable measurements above 1.3 N, whereas the 2 mm DMCH remained mechanically intact and electrically stable up to 5 N, exhibiting the smallest resistance change. Consistently, Supplementary Figure 12 shows that the 1 mm DMCH has lower conductivity ($\sigma \approx 0.025 \text{ mS}\cdot\text{cm}^{-1}$) than the thickness-insensitive regime observed for thicker DMCH ($\sigma \approx 0.12\text{--}0.14 \text{ mS}\cdot\text{cm}^{-1}$), consistent with prior reports on PEDOT:PSS-based electrodes^{R1–R6}. Together, these results indicate that 2 mm is the minimum thickness required to ensure a robust conductive path and reliable electrical performance under compressive loading up to 5 N.

To evaluate the electrical performance of the developed electrodes on skin, two electrodes (cathode and anode) were applied to the skin surface, and a voltage was used to induce current.”

(Page 24; line 560 in Revised Manuscript)

“This mechanism is illustrated in Figure 5b. Under low compression (p_1), the 2 mm-thick DMCH electrode forms a soft, conformal interface with the skin (a_1), maintaining an inter-electrode distance of approximately 10 mm (d_1), thereby initiating a localized stimulation pathway.”

(Page 31 in Revised Supplementary Information)

Supplementary Fig. 26 | Normalized additional currents induced by clothing pressure
Normalized additional current flows depending on the pressure, which induces variability in the haptic feedback intensity. A total of 8 repetitions were performed.

(Page 16-17 in Revised Supplementary Information; add Supplementary Fig. 11 and 12)

Supplementary Fig. 11 | Through-thickness (vertical) resistance as a function of applied compressive force for DMCH electrodes of varying thicknesses

Electrical characterization of the DMCH hydrogel electrode under a constant applied pressure of 5 N, where the through-thickness (vertical) resistance was measured using a universal testing machine (UTM).

Supplementary Fig. 12 | Comparison of the conductivity of DMCH electrodes with different thicknesses.

Measured electrical conductivity of the DMCH hydrogel as a function of thickness.

Comment #5: Fig. 3e compares the weight and area of DMCH electrodes with mechanical actuators and commercial TENS electrodes. However, the source of the commercial electrodes is not specified and should be clarified. Since these comparison parameters mainly relate to electrode density, comparing the DMCH electrodes with widely reported textile-based electrodes would more effectively highlight their advantages. Additionally, as electrical performance is a key consideration, it should also be included in the comparison for a more comprehensive evaluation.

Our response (Comment #5):

We thank the reviewer for this insightful comment and appreciate the opportunity to clarify the rationale for including mechanical actuators in Fig. 3e. Our overarching goal is to realize a body-scale suit that leverages electrical stimulation to provide multiple wearable-interaction capabilities, including full-body haptic feedback delivered with high spatiotemporal density. In this context, we included mechanical actuators as a benchmark because most current commercial (e.g., vibrotactile and electromechanical systems such as bHaptics) and research-grade haptic platforms rely predominantly on arrays of mechanical actuators (e.g., ERM motors, LRAs, or pneumatic actuators). These components typically dominate the overall weight and local bulk of the suit, and they fundamentally constrain achievable spatial resolution and wearability.

Specifically, a prior study²¹ adopted for comparison in Figure 3e, utilizing 36 mechanical actuators (0.73 units/cm²) reported a total system weight of 33.68 g (single actuator weight: 0.3 g) confined to a limited area of 49.32 cm². In contrast, TESS demonstrates weight efficiency, weighing only 11.57 g with 32 stimulators across a vastly larger area of 3700 cm². Furthermore, another referenced study⁴² utilizing a 32 actuator array yielded a total system weight of 130 g, further exemplifying the weight limitations of mechanical approaches. These comparisons indicate that high-density mechanical arrays inherently incur substantial weight accumulation, whereas TESS achieves a considerably lighter form factor despite extensive coverage. For this reason, Figure 3e was designed to position the DMCH not only as an alternative to TENS electrodes, but as a lightweight that can replace mechanical actuator. Detailed model information for the actuators and stimulators used in the comparison are provided in the Methods section. In the revised manuscript, we explicitly state in Figure 3e that commercial

electrodes or actuators are used for comparison.

We agree with the reviewer that comparing our DMCH with widely reported textile-based electrodes, including electrode area, provides a more comprehensive context. Accordingly, we have added Supplementary Table 2, which summarizes representative textile-based electrodes from prior literature, including electrode type, electrode area, and textile substrate type. We believe these additions clarify why mechanical actuators were included in Figure 3e and, at the same time, more clearly demonstrate that the proposed DMCH is not only electrically effective but also particularly advantageous for realizing lightweight, large-area, textile-based haptic suits.

Finally, detailed specifications, including the electrode source (company and model) and the electrode/actuator type, are provided in the Methods section.

21. Jung, Y. H. *et al.* A wireless haptic interface for programmable patterns of touch across large areas of the skin. *Nature Electronics* **5**, 374-385 (2022).
42. Yu, X. *et al.* Skin-integrated wireless haptic interfaces for virtual and augmented reality. *Nature* **575**, 473-479 (2019).

Authors' modification to the manuscript:

(Page 15; line 355 in Revised Manuscript)

“Figure 3e compares the weight and area of the DMCH-based electrodes used in our system with those of mechanical actuators reported in previous studies^{42,55,56}, as well as commercial electrodes for TENS. The model information of the stimulators and actuators used for comparison are provided in the Methods section. Current commercial (e.g., bHaptics) and research-grade systems rely predominantly on arrays of mechanical actuators (ERM motors, LRAs, or pneumatic systems), which introduce substantial weight and bulk at high spatial densities. Accordingly, Figure 3e positions the DMCH electrode not only as an alternative to conventional gel or TENS electrodes, but also as a lightweight, high-density building block for large-area haptic garments. In addition, Supplementary Table 2 compares DMCH with previously reported textile-based electrodes by summarizing electrode type, electrode area, and textile substrate type. Figure 3f presents a comparison of substrate and packaging material weights typically used to support electrodes and wiring in skin-interfaced electronics.”

(Page 33; line 776 in Revised Manuscript)

“Comparison of the area and weight of electrodes

The mechanical actuators evaluated in this study included various vibrotactile motors: an eccentric rotating mass (ERM) motor (C0720B003D, Jinlong Machinery & Electronics), a linear resonant actuator (LRA) (C10-100, Precision Microdrives™), an additional ERM (910-108.002, Precision Microdrives™), and other vibrotactile actuators referenced in [ref. 42]. Electrical stimulation electrodes included Ag/AgCl Monitoring Electrodes (2223H, 3M), Ag/AgCl Disposable ECG Electrodes (L-150X, VitrodeL), TENS electrode-1 (LG Electro Pad, 26419561, LG Electronics Inc.), and TENS electrode-2 (PI-5500, Chunghoon Co., Ltd.), designed for both TENS and EMS applications.”

(Page 43 in Revised Supplementary Information; add Supplementary Table 2)

reference	Electrode type	Textile substrate	Electrode area
[R1]	Silver/ carbon paste	Polyester 76 %, spandex 24 %	314 mm ²
[R2]	Silver/ carbon paste	Polyester 88 %, spandex 12 %	314 mm ²
[R3]	Screen-printed silver ink	Cotton knit/ polyester	600 mm ²
This work	Ag-PU conductor/ DMCH	Polyester 87 %, spandex 13 %	78.53 mm ²

Supplementary Table 2 | Comparison of reported textile-based electrodes (type, area, substrate)

Representative textile-based electrodes reported in literature, including electrode type, electrode area, and substrate type.

- R1. Kim, S., Lee, S. & Jeong, W. EMG measurement with textile-based electrodes in different electrode sizes and clothing pressures for smart clothing design optimization. *Polymers* **12**, 2406 (2020).
- R2. Kim, H., Rho, S., Han, S., Lim, D. & Jeong, W. Fabrication of textile-based dry electrode and analysis of its surface EMG signal for applying smart wear. *Polymers* **14**, 3641 (2022).
- R3. Nigusse, A. B., Malengier, B., Mengistic, D. A., Tseghai, G. B. & Van Langenhove, L. Development of washable silver printed textile electrodes for long-term ECG monitoring. *Sensors* **20**, 6233 (2020).

Comment #6: *The authors describe the effect of inter-electrode distance on perception threshold (Supplementary Fig. 11). The reasons for this are unclear and require further explanation. Furthermore, the authors also need to clarify how the center-to-center distance between the anode and cathode was determined and how this affected the experimental results.*

Our response (Comment #6):

We thank the reviewer for this insightful comment regarding the electrode design parameters. The optimization of the inter-electrode distance is critical for defining the geometry of the current path and the resulting neural activation profile. In this work, we established a 20 mm center-to-center distance based on three key scientific considerations:

1. Modulation of neural activation threshold

The spacing between the anode and cathode determines the depth of current penetration. Our investigation confirms that widening the inter-electrode distance extends the current distribution vertically through the skin layers. This expanded profile facilitates the recruitment of a larger volume of mechanoreceptors located in deeper tissues. Consequently, this electrode geometry effectively lowers the perception threshold, allowing the system to elicit clear tactile sensations at reduced stimulation voltages compared to narrower configurations^{R1-R3}.

2. Anatomical spatial acuity

The electrode design must align with the physiological characteristics of the target anatomy. Unlike the fingertips, which possess high receptor density, other skin parts of the body such as the forearm exhibit a relatively high two-point discrimination threshold^{R4}. Therefore, ultra-compact high-density arrays yield diminishing returns in spatial resolution for this region^{R5-R8}.

A 20 mm spacing provides sufficiently distinct haptic cues that align with the spatial acuity of the forearm, avoiding unnecessary complexity while ensuring signal distinguishability.

3. Wearability and interface stability

Finally, this configuration enhances the robustness of the wearable interface under dynamic conditions. By operating at a lower perception threshold (achieved through the optimized spacing), the system minimizes the risk of discomfort caused by motion artifacts. Specifically, it mitigates sharp pain caused by current concentration during partial electrode delamination or skin deformation, ensuring a stable and comfortable user experience in practical wearable scenarios.

In the revised manuscript, we explicitly justify our selection of a 20 mm center-to-center distance (using 10 mm diameter electrodes). Additionally, we have corrected the error in Supplementary Figure 18 (previously Supplementary Fig. 11, now Supplementary Fig. 18 in the revised manuscript) to ensure the accuracy of the data presented.

- R1. Gomez-Tames, J. D., Gonzalez, J. & Yu, W. in *2012 Annual International Conference of the IEEE Engineering in Medicine and Biology Society*. 3576-3579 (IEEE).
- R2. Tanaka, Y., Shen, A., Kong, A. & Lopes, P. in *Extended Abstracts of the 2023 CHI Conference on Human Factors in Computing Systems*. 1-5.
- R3. Kajimoto, H. in *Pervasive Haptics: Science, Design, and Application* 79-96 (Springer, 2016).
- R4. Solomonow, M., Lyman, J. & Freedy, A. Electrotactile two-point discrimination as a function of frequency, body site, laterality, and stimulation codes. *Annals of biomedical engineering* **5**, 47-60 (1977).
- R5. Štrbac, M. *et al.* Integrated and flexible multichannel interface for electrotactile stimulation. *Journal of neural engineering* **13**, 046014 (2016).
- R6. Chai, G., Zhang, D. & Zhu, X. Developing non-somatotopic phantom finger sensation to comparable levels of somatotopic sensation through user training with electrotactile stimulation. *IEEE Transactions on Neural Systems and Rehabilitation Engineering* **25**, 469-480 (2016).
- R7. Franceschi, M. *et al.* in *2015 37th Annual International Conference of the IEEE Engineering in Medicine and Biology Society (EMBC)*. 4554-4557 (IEEE).
- R8. Cheng, S. & Zhang, D. in *2017 10th International Conference on Human System Interactions (HSI)*. 120-124 (IEEE).

Authors' modification to the manuscript:

(Page 23 in Revised Supplementary Information)

Supplementary Fig. 18 | The effect of inter-electrode distance on perception threshold

As the distance between electrodes increases, the current path flows deeper into the skin, lowering the perception threshold.

(Page 18; line 410 in Revised Manuscript)

“Based on this principle, the center-to-center distance between the anode and cathode in the DMCH array was set to 20 mm, optimizing the trade-off between spatial resolution and stimulation voltage requirements⁴³. **The inter-electrode distance was selected considering the neural activation threshold^{R1-R3}, the spatial acuity of the forearm^{R4-R8}, and the need to reduce discomfort from movement, thereby ensuring wearability and interface stability.** This spacing may require adjustment depending on anatomical site and user-specific variability.”

Comment #7: *Fig. 5e shows the resistance variation of the pressure sensor at arm positions 1 to 4. What is the range of pressure variation caused by individual body shape differences and positional variations between wears at different positions on the human body? Can all of this be detected by the reported sensor?*

Our response (Comment #7):

Thank you for this valuable comment. Within the sensing range characterized in Figure 5c, the proposed sensor can resolve garment-pressure variations arising from differences in body shape and measurement location. Specifically, we measured garment pressure at arm positions 1–4 and at representative torso locations in five participants with diverse body types (Fig. 5e). Across these measurements, the sensor outputs remained within the characterized sensing range and consistently preserved location-dependent differences, indicating that the sensor’s dynamic range and sensitivity are well matched to pressure levels typically encountered in compression garments and sleeves.

To assess the reliability of the pressure-sensing values in Figure 5e, we measured arm circumference at positions 1–4 (from the thinner distal region to the thicker proximal region) in the same five participants and compared these measurements with the corresponding resistance changes of the pressure sensors. Arm circumference increased from position 1 to position 4, and the sensor resistance exhibited a consistent position-dependent trend across participants (Supplementary Fig. 27). These results support that the sensor reliably resolves the garment-pressure gradient along the arm.

To address positional variations between wears (donning–doffing repeatability), we quantified inter-wear positional displacement over 10 repeated wears at four representative sites (Supplementary Fig. 24). The displacement was 12.2 ± 6.2 mm on the arm, 7.9 ± 3.4 mm on the abdomen, 10.6 ± 3.1 mm on the flank, and 5.7 ± 1.5 mm on the shoulder. These values indicate that torso and shoulder placement are reproduced within ~ 1 cm, whereas the arm shows the largest variability. This variability remains confined to a spatial region on the order of a single electrode footprint (10 mm diameter) and within the same functional segment of the limb^{R1}. We therefore regard this degree of positional variability as acceptable, while acknowledging that improving placement repeatability remains an important direction for future refinement.

We revised the manuscript to explicitly state that garment pressure variations arising from individual body-shape differences and positional changes across multiple body sites (arm and torso) fall within the sensing range characterized in Figure 5c, and that the reported sensor can resolve these variations, as demonstrated in Figure 5e.

R1. Jumet, B. *et al.* Fluidically programmed wearable haptic textiles. *Device 1* (2023).

Authors' modification to the manuscript:

(Page 32 in Revised Supplementary Information; add Supplementary Fig. 27)

Supplementary Fig. 27 | Relationship between arm circumference and garment pressure sensing

Arm circumference and corresponding resistance changes of the pressure sensors at positions 1–4 along the arm for five participants. The arm circumference was measured using a flexible measuring tape.

(Page 29 in Revised Supplementary Information; add Supplementary Fig. 24)

Supplementary Fig. 24 | Positional displacement of the TESS after repeated donning and doffing.

Average displacement of the electrode interface after 10 donning/doffing cycles across four major body positions.

Authors' modification to the manuscript:

(Page 25; line 589 in Revised Manuscript)

“In general, pressure distributions varied with body type, with the arm regions (positions 1–4) exhibited a progressively increasing compression. To confirm that sensor resistance changes reflect pressure variations from local garment fit, we measured arm circumference at each sensor location (positions 1–4) for each participant using a flexible tape. Arm circumference increased from position 1 to position 4 and exhibited a matching position-dependent trend in the sensor responses (Supplementary Fig. 27). These results show that the sensors can resolve garment-pressure differences arising from inter-individual variations in body shape. Given the substantial variability in applied pressure, the results emphasize the need to address mismatches in electrotactile perception caused by differences in pressure distribution across users.”

(Page 23; line 540 in Revised Manuscript)

“Unlike traditional systems that rely on strong adhesive layers—often causing skin irritation, discomfort upon removal, and potential damage to embedded components—our suit secures its electrodes through garment pressure. **Because this wear modality can introduce positional shifts after repeated donning and doffing, we experimentally quantified the resulting positional displacement of the TESS suit, as shown in Supplementary Figure 24. To address positional variations between wears (donning–doffing repeatability), inter-wear positional displacement was quantified over 10 repeated wears at four representative sites. Overall, the shift remained on the order of a single electrode footprint and within the same functional body part segment, and is therefore considered acceptable for the present implementation^{R1}. Additionally, adhesion tests comparing the DMCH-based electrode with commercial TENS electrodes are presented in Supplementary Figure 25.**”

Comment #8: *In Fig. 5g, the authors adjusted the stimulation voltage based on pressure data to compensate for individual differences, aiming to achieve consistent sensory perception across users. However, the amount of compensation requires further statistical analysis to validate its effectiveness. Additionally, is there any existing research supporting the possibility of achieving consistent electrical stimulation effects by standardizing the current output?*

Our response (Comment #8):

We thank the reviewer for this insightful comment. We agree that a more quantitative analysis is needed to support the effectiveness of the compensation strategy in Figure 5g. We performed a within-subject comparison in which the participant wore sleeves on both arms (Fig. 5h). The left arm was operated without feedback compensation, whereas the right arm was operated with feedback compensation, and the participant rated perceived intensity on a 0–10 scale (0: no sensation, 5: comfortable/moderate, 10: intolerable). As garment pressure increased from position 1 to position 4, the uncompensated condition showed a corresponding increase in perceived intensity, with the strongest sensation reported at position 4. In contrast, under the compensated condition, the participant reported more comfortable and stable sensations across positions (Fig. 5i). These results are consistent with pressure-induced increases in unintended current under the uncompensated condition and indicate that the proposed compensation improves the consistency of perceived intensity.

In addition, prior studies have shown that standardizing (regulating) the stimulation current is an effective strategy for achieving more consistent perceived intensity in electrotactile systems. In electrotactile and transcutaneous stimulation, the skin–electrode impedance can vary continuously with contact pressure, hydration, and skin condition, which in turn alters the delivered current when voltage is held constant. Accordingly, multiple studies have concluded that current regulation is generally more robust for maintaining consistent sensation. For example, prior work^{R1} experimentally demonstrated that, under fixed-voltage stimulation, the same nominal amplitude can produce noticeably different tactile sensations as contact pressure changes, because the delivered current depends directly on the skin–electrode impedance. Similarly, another study¹⁷ identified electrode–skin impedance fluctuations as a primary source of intensity variation and proposed a closed-loop controller that updates stimulation parameters (e.g., current amplitude and pulse duration) based on measured impedance to regulate perceived sensation. We acknowledge that perfectly invariant perception cannot be guaranteed due to psychophysical variability; however, pressure monitoring enables us to standardize the delivered stimulation current by compensating for pressure-induced impedance changes, thereby providing a more stable perceptual baseline than voltage-controlled stimulation

without feedback.

The need for compensation is particularly pronounced in our TESS architecture. Unlike conventional systems that rely on adhesive electrodes, TESS employs DMCH where electrical contact is established via garment-applied pressure. In this configuration, the skin–electrode impedance is intrinsically coupled to contact pressure. Because contact pressure varied with body location and inter-individual body shape, the proposed feedback compensation is required to suppress location- and subject-specific pressure variations and thereby maintain consistent stimulation intensity.

In the revised manuscript, we have incorporated prior work showing that standardizing the current level is an effective strategy for maintaining consistent perceived intensity in electrotactile systems.

- R1. Lim, K. *et al.* Interference haptic stimulation and consistent quantitative tactility in transparent electrotactile screen with pressure-sensitive transistors. *Nature communications* **15**, 7147 (2024).
- 17 Akhtar, A., Sombeck, J., Boyce, B. & Bretl, T. Controlling sensation intensity for electrotactile stimulation in human-machine interfaces. *Science robotics* **3**, eaap9770 (2018).

Authors' modification to the manuscript:

(Page 63 in Revised Manuscript)

Fig. 5 | User study on the full-body electrical stimulation suit

a, Photograph of a participant wearing the textile-based electrical stimulation suit designed for the chest, abdomen, sides, shoulders, and arms. The suit provides targeted compression and high spatial resolution for full-body stimulation. **b**, Schematic illustrating the electrical stimulation mechanism, showing how current levels vary in response to changes in applied pressure (p), contact area (a), and interelectrode distance (d) under low-pressure (top) and high-pressure (bottom) conditions. **c**, Photograph of the integrated pressure sensor embedded in PU-based fabric with a conductive hydrogel for real-time pressure sensing (left). Detection range of the sensor shown on the right. Inset: image of a 4×1 unencapsulated pressure sensor array and a 4×3 tactile stimulator array. Scale bar, 10 mm. **d**, Normalized pressure changes measured by an interdigitated pressure sensor while participants wore a sleeve. Measurement locations included: wrist (1), mid-forearm (2), below the elbow (3), and above the elbow (4), using the elbow as the central reference point ($n = 9$ repetitions of donning and doffing; bar height: mean;

error bars, s.d.). **e**, Measured clothing pressure across different body locations (as shown in panel a) for five participants (three female and two male) with varying body types and ages. **f**, Flowchart illustrating the real-time electrical stimulation feedback system. A mobile device sets the default resistance of the pressure sensor. Real-time clothing pressure at each arm location is captured via four ADC inputs, and normalized pressure values are used to calculate the required stimulation voltage. The selected stimulation channel determines frequency, and the output voltage is computed using a pre-calibrated formula. **g**, Three distinct tactile sensations were simulated through real-time tactile feedback. Wireless pressure sensors positioned across the body calculate garment pressure, triggering spatiotemporal activation of stimulation channels. Before feedback, the system considers both the threshold current and additional current variation with respect to the frequency. After feedback integrates all key parameters—including pressure-induced current adjustments—and enables real-time feedback and dynamic stimulation to accommodate individual users. **h**, Heatmap of sleeve compression levels across arm positions 1–4 in a within-subject experiment. The sleeve on the left arm was operated without feedback compensation, whereas the sleeve on the right arm was operated with feedback compensation. **i**, Heatmap of perceived intensity ratings across arm positions 1–4 in the same within-subject experiment (0–10 scale; 0: no sensation, 5: comfortable/moderate, 10: intolerable). The after-feedback condition shows more stable and comfortable perceived intensity compared to the before-feedback condition.

(Page 26; line 608 in Revised Manuscript)

“By applying a predefined correction equation customized to each individual, the system dynamically adjusted the stimulation to deliver consistent tactile feedback. A within-subject comparison further validates the effectiveness of the proposed compensation strategy (Fig. 5h). In this experiment, sleeves were worn on both arms, and feedback compensation was applied only to the right arm (Fig. 5i). Participants rated perceived intensity on a 0–10 scale (0: no sensation, 5: comfortable/moderate, 10: intolerable) at positions 1–4. Without feedback, perceived intensity increased with garment pressure toward position 4; in contrast, the compensated condition produced more stable and comfortable sensations across positions, supporting that the proposed feedback improves the consistency of perceived intensity. In addition, Supplementary Table 5 provides a quantitative and qualitative comparison of TESS with representative haptic systems reported in prior work.”

Comment #9: *In the applications of the TESS, it is important to evaluate the impact of the integrated calibration system based on pressure on the effectiveness of various applications. However, the current experiments do not appear to address this aspect.*

Our response (Comment #9):

We thank the reviewer for this insightful comment. We agree that it is essential to evaluate how the pressure-based integrated calibration system in TESS influences the effectiveness of its applications. In response, we added a user experiment to quantify how the pressure-based integrated calibration affects functional stimulation consistency under realistic garment-pressure gradients. Specifically, using bilateral sleeves in a within-subject design, we compared perceived-intensity maps across Positions 1–4 with the calibration disabled versus enabled, showing that the calibration maintains a more uniform and comfortable perceptual profile (Fig. 5h,i).

Additionally, this integrated calibration framework was used in the body-scale demonstrations in Figure 6, including a VR bullet/grenade game, VR-assisted treatment for essential tremor reduction, and augmented-reality applications, where stable and repeatable

stimulation across body locations is required. We hope these results further support the practical utility of the proposed pressure-based integrated compensation system for robust, body-scale electro-tactile applications.

Authors' modification to the manuscript:
(Page 63 in Revised Manuscript)

Fig. 5 | User study on the full-body electrical stimulation suit

a, Photograph of a participant wearing the textile-based electrical stimulation suit designed for the chest, abdomen, sides, shoulders, and arms. The suit provides targeted compression and high spatial resolution for full-body stimulation. **b**, Schematic illustrating the electrical stimulation mechanism, showing how current levels vary in response to changes in applied pressure (p), contact area (a), and interelectrode distance (d) under low-pressure (top) and high-pressure (bottom) conditions. **c**, Photograph of the integrated pressure sensor embedded in PU-

based fabric with a conductive hydrogel for real-time pressure sensing (left). Detection range of the sensor shown on the right. Inset: image of a 4×1 unencapsulated pressure sensor array and a 4×3 tactile stimulator array. Scale bar, 10 mm. **d**, Normalized pressure changes measured by an interdigitated pressure sensor while participants wore a sleeve. Measurement locations included: wrist (1), mid-forearm (2), below the elbow (3), and above the elbow (4), using the elbow as the central reference point ($n = 9$ repetitions of donning and doffing; bar height: mean; error bars, s.d.). **e**, Measured clothing pressure across different body locations (as shown in panel a) for five participants (three female and two male) with varying body types and ages. **f**, Flowchart illustrating the real-time electrical stimulation feedback system. A mobile device sets the default resistance of the pressure sensor. Real-time clothing pressure at each arm location is captured via four ADC inputs, and normalized pressure values are used to calculate the required stimulation voltage. The selected stimulation channel determines frequency, and the output voltage is computed using a pre-calibrated formula. **g**, Three distinct tactile sensations were simulated through real-time tactile feedback. Wireless pressure sensors positioned across the body calculate garment pressure, triggering spatiotemporal activation of stimulation channels. Before feedback, the system considers both the threshold current and additional current variation with respect to the frequency. After feedback integrates all key parameters—including pressure-induced current adjustments—and enables real-time feedback and dynamic stimulation to accommodate individual users. **h**, Heatmap of sleeve compression levels across arm positions 1–4 in a within-subject experiment. The sleeve on the left arm was operated without feedback compensation, whereas the sleeve on the right arm was operated with feedback compensation. **i**, Heatmap of perceived intensity ratings across arm positions 1–4 in the same within-subject experiment (0–10 scale; 0: no sensation, 5: comfortable/moderate, 10: intolerable). The after-feedback condition shows more stable and comfortable perceived intensity compared to the before-feedback condition.

(Page 26; line 608 in Revised Manuscript)

“By applying a predefined correction equation customized to each individual, the system dynamically adjusted the stimulation to deliver consistent tactile feedback. A within-subject comparison further validates the effectiveness of the proposed compensation strategy (Fig. 5h). In this experiment, sleeves were worn on both arms, and feedback compensation was applied only to the right arm (Fig. 5i). Participants rated perceived intensity on a 0–10 scale (0: no sensation, 5: comfortable/moderate, 10: intolerable) at positions 1–4. Without feedback, perceived intensity increased with garment pressure toward position 4; in contrast, the compensated condition produced more stable and comfortable sensations across positions, supporting that the proposed feedback improves the consistency of perceived intensity. In addition, Supplementary Table 5 provides a quantitative and qualitative comparison of TESS with representative haptic systems reported in prior work.”

Comment #10: *The study reported the reconstruction of various tactile scenarios using four specific stimulation frequencies. However, the mechanism linking these frequencies to distinct tactile perceptions remains unclear. Does it simply demonstrate that users can differentiate between electrical stimulations of varying frequencies? Additionally, as noted in row 397, many participants were confused by 70 Hz and 120 Hz, associating them with pressure and roughness. Could this confusion be attributed to human insensitivity or reduced perceptual resolution within this frequency range?*

Our response (Comment #10):

We appreciate the reviewer’s insightful comment regarding frequency-dependent tactile

perception. To elicit distinct tactile percepts, we selected four discrete stimulation frequencies—5 Hz (touch), 30 Hz (tickling), 70 Hz (roughness), and 120 Hz (pressure), based on established psychophysical principles^{R1,R2}. Prior literature consistently reports that low-frequency electrotactile stimulation (<10 Hz) produces temporally discrete pulses that are perceived as tapping-like sensations, whereas mid-range frequencies (~30–70 Hz) introduce stronger temporal modulation that is often perceived as vibration or roughness. At higher frequencies (>100 Hz), temporal fusion tends to occur, yielding more continuous sensations commonly described as smooth pressure or superficial tingling^{R3,R4}.

With respect to the observed confusion between 70 Hz and 120 Hz, we attribute this outcome to inherent limitations in human frequency discrimination at mid-to-high electrotactile frequencies. At elevated frequencies, temporal summation and the broad bandwidth of rapid-adapting afferents (e.g., Pacinian corpuscles) lead to overlapping neural recruitment, thereby reducing qualitative separability of stimuli. Consistent with prior studies^{R5}, perceptual resolution degrades in this regime (>50 Hz), where distinct stimulation frequencies can converge toward similar percepts. Despite these psychophysical constraints, our system leverages frequency modulation to render multiple perceptual categories and achieves a minimum classification accuracy of 85 %. To the best of our knowledge, TESS represents the first full-body haptic suit that delivers these distinct frequency-encoded percepts through a unified textile interface.

- R1. Graczyk, E. L., Christie, B. P., He, Q., Tyler, D. J. & Bensmaia, S. J. Frequency shapes the quality of tactile percepts evoked through electrical stimulation of the nerves. *Journal of Neuroscience* **42**, 2052-2064 (2022).
- R2. Djozic, D. J., Bojanic, D., Krajoski, G., Popov, N. & Ilic, V. in *2015 IEEE 15th International Conference on Bioinformatics and Bioengineering (BIBE)*. 1-5 (IEEE).
- R3. Ray, R. K., Kumar Vasudevan, M. & Manivannan, M. Electrotactile displays: taxonomy, cross-modality, psychophysics and challenges. *Frontiers in Virtual Reality* **5**, 1406923 (2024).
- R4. Yang, S. *et al.* A Comprehensive Survey of Electrical Stimulation Haptic Feedback in Human-Computer Interaction. *arXiv preprint arXiv:2504.21477* (2025).
- R5. Paredes, L. P., Dosen, S., Rattay, F., Graimann, B. & Farina, D. The impact of the stimulation frequency on closed-loop control with electrotactile feedback. *Journal of neuroengineering and rehabilitation* **12**, 35 (2015).

Authors' modification to the manuscript:

(Page 19; line 447 in Revised Manuscript)

“Four discrete frequencies were strategically selected based on established psychophysical principles: 5 Hz (touch), 30 Hz (tickling), 70 Hz (roughness), and 120 Hz (pressure). This selection leverages the finding that low-frequency stimulation (<10 Hz) activates discrete mechanoreceptors to produce tapping sensations, whereas higher frequencies (>100 Hz) induce sensory fusion, resulting in continuous pressure^{R1,R2}. By manipulating the electrical parameters—specifically the amplitude and frequency of stimulation—various tactile sensations can be effectively rendered (Fig. 4e). Participants were able to distinguish discrete pulses at a low frequency of 5 Hz, which elicited light, tapping sensations reminiscent of gentle touch. Increasing the frequency to 30 Hz induced a tickling sensation that subtly propagated across the skin, with several participants likening it to the feeling of insects crawling along the arm. At 70 Hz, the stimulation evoked a coarse tactile sensation, similar to the sensation of stroking animal fur. A further increase to 120 Hz produced a more intense pressure-like sensation, comparable to localized skin deformation. The average stimulation current showed

a clear frequency-dependent increase. At 5 Hz, the mean current was 0.0776 mA (SD: 9.31×10^{-5} A), rising to 0.3961 mA (SD: 1.72×10^{-4} A) at 120 Hz. These results support the observation that higher frequencies require greater current amplitudes to produce perceivable tactile feedback. The increasingly broad and elevated threshold current ranges at higher frequencies may be due to a reduced sensitivity to incremental changes in sensation near 100 Hz, a phenomenon previously reported in the literature²⁴.

To assess the perceptual clarity of each rendered sensation, participants were asked to identify four distinct stimulation patterns based on predefined tactile descriptors: touch, tickling, roughness, and pressure (Supplementary Table 2). Ten participants were randomly presented with four pattern sequences and asked to match each to the most appropriate keyword. The classification task yielded a maximum prediction accuracy of 92.5 % and an average accuracy of 89.37 % (Fig. 4f). Notably, many participants confused pressure with roughness. This observation is consistent with documented psychophysical limits in frequency discrimination. In the mid-to-high frequency range (>50 Hz), the broad tuning of rapid-adapting afferents (e.g., Pacinian corpuscles) and temporal summation lead to overlapping neural recruitment, thereby reducing the perceptual separability of distinct frequencies⁴⁵. As a result, users tend to report a more continuous vibration in which “roughness” and “pressure” converge toward similar percepts, with perceived intensity often providing a stronger cue than frequency in this regime. This reflects an inherent limitation of human tactile frequency resolution^{R5}.”

Comment #11: *The user study should include detailed information on the age and gender distribution of the participants. This is important as such factors can influence tactile perception.*

Our response (Comment #11):

Thank you for your valuable comment. We agree that reporting participant age and gender is important for reproducibility and generalizability. Accordingly, we have revised the manuscript to include the number of participants, gender distribution, and age distribution for all user studies conducted in this work.

Authors’ modification to the manuscript:

(Page 41 in Revised Supplementary Information; add Supplementary Fig. 36)

Supplementary Fig. 36 | Age distribution of participants in the user study

Histogram showing the number of participants as a function of age (years) for all user studies conducted in this work. **a**, Tomato-moving task. **b**, Spiral-drawing task. **c**, Human participant sensory perception test. **d**, Sweat test.

Authors' modification to the manuscript:

(Page 29; line 670 in Revised Manuscript)

“A total of twelve participants (mean age = 26.83 years; four females; **Supplementary Fig. 36a**) and ten participants (mean age = 26.80 years; three females; **Supplementary Fig. 36b**) were enrolled in the respective task groups.”

Authors' modification to the manuscript:

(Page 34; line 812 in Revised Manuscript)

“A total of 15 volunteers of diverse genders, aged in their 20s to 30s (mean age: 25.95 years; **three females; Supplementary Fig. 36c**), participated in the electrical stimulation stimulation study.”

Authors' modification to the manuscript:

(Page 35; line 835 in Revised Manuscript)

“Four participants (both male and female) in their 20s, with a mean age of 24.75 years (**two females; Supplementary Fig. 36d**), took part in a user test designed to evaluate the effects of perspiration on electrical stimulation.”

Comment #12: *There are a few minor issues that should be addressed. There seems to be a watermark in the top left corner of Fig. 2f. Additionally, stratum corneum belongs to epidermis, so the skin model in Fig. 4a needs to be adjusted. A scale bar is preferred in Supplementary Fig. 16.*

Our response (Comment #12):

We thank the reviewer for this comment. We have revised the manuscript and figures to reflect these additional suggestions. We removed the watermark in Figure 2f, corrected the mislabeled skin model in Figure 4a, and added a scale bar to Supplementary Figure 16.

Authors' modification to the manuscript:

(Page 57 in Revised Manuscript)

Fig. 2 | Characterization of the electrical stimulation electrode

a, Schematic diagram of a single electrical stimulator showing the DMCH and the Ag-PU conductor printed on a PU-based textile. **b**, Cross-sectional SEM image showing seamless integration of the DMCH on the Ag-PU conductor printed onto a PU-based textile. No interfacial gaps were observed. Scale bars, 100 μm . **c**, Schematic showing the mechanism of robust adhesion between DMCH and the Ag-PU conductor. **d**, Photographs of the peeling test used to compare the peeling force of the DMCH/Ag-PU conductor with that of the DMCH/silicon elastomer. Scale bars, 20 mm. **e**, Peeling forces per width of DMCH attached to the Ag-PU conductor and silicon elastomer. **f**, Current density measured on skin as a function of frequency for the Ag-PU conductor (used as the interconnect) and three electrical stimulation electrodes with different hydrogel compositions. **g**, Stress as a function of strain for conductive hydrogel (red), hydrogel (black), and ionic hydrogel (blue). The DMCH is capable of withstanding strains of up to 600%. **h**, Impedance spectra of various hydrogel electrodes with different compositions: pure hydrogel (black), ionic hydrogel (blue), and DMCH (red). The DMCH formed by mixing PEDOT and ions exhibited the lowest impedance, owing to the doping effect of the ions on PEDOT.

(Page 61 in Revised Manuscript)

Fig. 4 | User study of the multimodal textile-based electrical stimulation system

a, Schematic of the electrical impedance model of human skin, shown as a simplified equivalent circuit. **b**, Representative waveforms of voltage (top) and current (bottom) signals applied to the skin during electrical stimulation. **c**, Comparison of individual stimulus thresholds and maximum tolerable voltages across participants ($n = 15$). Squares indicate mean values; center lines show medians; box limits represent upper and lower quartiles; whiskers extend to $1.5 \times$ the interquartile range. **d**, Log-log plot showing the linear relationship between perception threshold current and stimulation frequency. The fitted line has a constrained slope of 0.17076 with adjusted $R^2 > 0.9$ ($n = 9$). **e**, Mapping of four distinct tactile sensations—touch, tickling, roughness, and pressure—based on stimulation current and frequency ($n = 15$; data shown as violin plots). **f**, Confusion matrix showing recognition accuracy in 10 blind tests of forearm (posterior) stimulation without prior training. Participants identified sensations from four predefined categories. **g**, Photograph of the forearm showing placement of the textile-based electrode and two commercial TENS electrodes used to assess current variation before and after physical activity. **h**, Normalized current measurements before and after a 10-min run on a track ($n = 3$). **i**, Heat map showing subjective itchiness ratings following 10 min of exercise, evaluated for three different electrode types during delivery of various sensations ($n = 3$).

Supplementary Fig. 25 | Comparison of skin peel-off tests of electrodes a-b, The strong adhesiveness causes the skin to stretch when the electrode is removed, which is irritating to the skin. **(a)** Photograph showing the peel-off force of a commercial electrode-1. The electrode is divided into a conductive part in the center and an adhesive part surrounding it. **(b)** Photograph showing the peel-off force of a commercial electrode-2. This electrode has all the conductive and adhesive layers combined. **C,** Photograph showing the peel-off force of a DMCH. DMCH does not have strong adhesive properties and is not irritating to the skin. The diameter of the electrode used in the photograph is 40 mm. **Scale bars, 10 mm.**

Referee #2:

Summary Comments: *This manuscript presents a full-body stimulation suit with good performance and mechanical/electrical properties and complementary study on the practical application, which is interesting and has great application potential in medical and extended reality fields. However, the scientific novelty is not clear enough. Is the novelty of this work in material design, stimulation performance, or whole wearable system design? There are many works on textile wearable electronics, and what is the key innovation of this work? What is the scientific point of the key innovation? I suggest addressing the following questions for further consideration of its suitability for publication.*

Our response: We thank the reviewer for raising this important point. We agree that the scientific novelty should be stated more explicitly. The key innovation of this work is a materials-to-systems co-design that enables a lightweight, breathable, and scalable electrotactile textile suit, and our novelty spans (i) material design, (ii) a pressure sensor-based haptic feedback compensation system, and (iii) whole-body scale wearable system integration, as summarized below.

(i) Large area, breathable conductors with robust hydrogel integration

We thank the reviewer for this important comment regarding the novelty in materials design. We introduce a large-area, printable Ag-PU conductor that is directly patterned on commercial textile substrates, while preserving key textile attributes including flexibility and breathability. Importantly, our conductor is engineered to form strong interfacial bonding with the hydrogel-based electrode layer, enabling robust integration of conductive routing and soft electrodes without heavy encapsulation. While prior textile electronics have demonstrated body-scale sensing suits or textile networks, these systems often integrate functional devices on non-breathable plastic substrates and/or rely on knitting or embroidery of conductive yarns for electrical interconnects^{R1-R3}. In these cases, the resulting implementations can remain partially rigid and are not optimized for delivering stimulation across the whole body, due to limited electrode–skin interfacing and/or insufficient stimulation performance at scale. We acknowledge that there are prior studies that print Ag-based inks directly onto textiles. However, these reports typically do not address robust integration with hydrogel electrodes such as DMCH through interfacial bonding, and, to our knowledge, they have not demonstrated this conductor–hydrogel integration in a truly large-area, garment-scale format. In contrast, our approach is designed around large-area printed Ag ink conductors directly patterned on commercial textiles and interfaced with hydrogel-bonded stimulation electrodes, providing a materials platform that is directly aligned with scalable, garment-level electrotactile systems.

(ii) Pressure-dependent feedback control for consistent electrotactile output

Contact pressure strongly affects electrotactile stimulation because it changes how well the electrode contacts the skin, which in turn alters the skin–electrode impedance and the amount of current delivered. Prior electrotactile systems have addressed this issue through (1) fingertip/hand-based interfaces that directly measure touch pressure and use it for calibration or control^{R4} and/or (2) approaches that monitor skin–electrode impedance, often using adhesive gel electrodes, to regulate stimulation under interface fluctuations^{R5}. However, these implementations are typically localized, and, to our knowledge, an analogous pressure-aware compensation strategy has not been demonstrated in a large-area, garment-scale electrotactile system that must operate across multiple body sites. In contrast to adhesive-based electrodes, our system establishes electrical contact through garment-applied pressure across diverse body locations (including the hand, arm, and torso). Because the skin–electrode impedance is coupled to contact pressure in this configuration, we incorporate pressure monitoring and

pressure-dependent haptic feedback compensation to standardize delivered stimulation conditions and maintain consistent perceptual output across body locations and users. This pressure-aware compensation constitutes a key systems-level advance, directly addressing a practical challenge that is less emphasized in prior garment-based electrohaptic implementations while enabling large-area deployment.

(iii) Whole wearable system design

At the system level, we demonstrate a body-scale electrohaptic textile system with integrated conductors, electrodes, and sensing, designed for high-area coverage while remaining lightweight and breathable. Prior haptic demonstrations are often confined to localized body regions, such as the hand, or cover only limited portions of the body, which inherently restricts the achievable sense of full-body immersion. In contrast, our system integrates textile-scale routing, stimulation electrodes, and sensing to enable whole-body haptic garments. This body-scale electrohaptic architecture extends beyond hand-localized interfaces and enables distributed, immersive tactile feedback across the body.

We have revised the manuscript to address the reviewer's comments, and we hope the three key innovations of this work—materials design, feedback control, and system-level integration—are now more clearly articulated.

- R1. Lin, R. *et al.* Wireless battery-free body sensor networks using near-field-enabled clothing. *Nature communications* **11**, 444 (2020).
- R5. Wicaksono, I. *et al.* A tailored, electronic textile conformable suit for large-scale spatiotemporal physiological sensing in vivo. *npj Flexible Electronics* **4**, 1-13 (2020).
- R2. Lee, S. *et al.* A body-scale textile-based electromyogram monitoring system with coaxially shielded conductive yarns. *Science Advances* **11**, eadx4518 (2025).
- R3. Lim, K. *et al.* Interference haptic stimulation and consistent quantitative tactility in transparent electrohaptic screen with pressure-sensitive transistors. *Nature communications* **15**, 7147 (2024).
- R4. Kajimoto, H., Kawakami, N., Maeda, T. & Tachi, S. in *Proc. World Multiconference on Systemics, Cybernetics and Informatics (SCI2001)*. 95-99.

Comment #1: Please specify the advantage of the Ag-PU and DMCH electrode. How about the environmental stability of the DMCH electrode? Such as with long-term exposure in air, high temperature/humidity, etc.

Our response (Comment #1):

We appreciate the reviewer's valuable comment regarding environmental stability. In response, we performed a dehydration study under three conditions to evaluate the environmental stability of the DMCH: room temperature (20 °C, 46 % relative humidity), elevated temperature (60 °C, 46% relative humidity), and high humidity (20 °C, 84 % relative humidity). We monitored the time-dependent mass change of four hydrogel formulations, hydrogel, PEDOT:PSS hydrogel, ionic hydrogel (with lithium chloride, LiCl), and DMCH, over 156 h (one week), recording measurements every 12 h.

Quantitative analysis shows that Ionic Hydrogel and DMCH, both containing LiCl, exhibited comparable moisture-retention profiles across all three environments. At room temperature, DMCH and Ionic hydrogel maintained average moisture contents of 0.68 % and 0.65 %, respectively, whereas Hydrogel and PEDOT:PSS hydrogel maintained lower moisture contents of 0.26 % and 0.17 %, respectively.

Under high-temperature conditions, DMCH and Ionic Hydrogel maintained moisture contents of 0.58 % and 0.57 %, respectively, whereas Hydrogel and PEDOT:PSS hydrogel

showed lower moisture retention of 0.20 % and 0.13 %, respectively. Under high-humidity conditions, DMCH and Ionic hydrogel exhibited moisture absorption of 1.23 % and 1.18 %, while Hydrogel and PEDOT:PSS hydrogel retained 0.51 % and 0.35 %, respectively.

The slight moisture uptake observed for DMCH and the ionic hydrogel arises from the presence of LiCl, which is hygroscopic and can absorb ambient moisture, leading to a gradual increase in water content over time. Accordingly, the mass-change behavior of LiCl-containing hydrogels is governed more strongly by humidity than by temperature. Importantly, the measurable moisture uptake occurs only under extremely humid conditions (e.g., 84% relative humidity in our test) and remains small; thus, it is not expected to meaningfully affect electrical stimulation performance. More importantly, DMCH and the ionic hydrogel exhibit improved long-term stability under conditions relevant to practical use, particularly at room and elevated temperatures. Under these conditions, both materials reach an equilibrium state within ~12 h, whereas the hydrogel and PEDOT:PSS hydrogel fail to maintain hydrogel functionality after prolonged exposure to heat and humidity.

Authors' modification to the manuscript:

(Page 13; line 308 in Revised Manuscript)

“DMCH exhibited a distinct behavior: after an initial moisture loss during the first ~4 h, it reached equilibrium and maintained a stable moisture content throughout the remainder of the 24-h period. To assess long-term environmental stability, we compared the dehydration behavior of DMCH with three other hydrogels (hydrogel, PEDOT:PSS hydrogel, and ionic hydrogel) under room temperature (20 °C, 46 % relative humidity), high temperature (60 °C, 46 % relative humidity), and high humidity (20 °C, 84 % relative humidity) for 156 h (Supplementary Fig. 14). Quantitatively, DMCH maintained an average moisture content of 0.68 % and 0.58 % at room temperature and high temperature, respectively, compared with 0.26 %/0.20 % for the hydrogel and 0.17 %/0.13 % for the PEDOT:PSS hydrogel. Under high-humidity conditions, DMCH retained 1.23 % moisture, higher than the 0.51 % and 0.35 % observed for the hydrogel and PEDOT:PSS hydrogel, respectively, reflecting the humidity-dependent, LiCl-driven hygroscopic characteristic.”

(Page 19 in Revised Supplementary Information; add Supplementary Fig. 14)

Supplementary Fig. 14 | Environmental stability of DMCH and reference hydrogels

Time-dependent moisture content of four hydrogels (Hydrogel, PEDOT:PSS hydrogel, ionic hydrogel with LiCl, and DMCH) over 156 h under room-temperature, high-temperature, and high-humidity conditions.

Comment #2: Biocompatibility of the materials should be discussed, for instance, in the introduction.

Our response (Comment #2):

Thank you for raising this critical point. To assess skin biocompatibility, one participant wore the sleeve containing DMCH and the printed Ag-PU conductor for 24 h during daily activities (Supplementary Fig. 13), with placement on the non-dominant arm to minimize positional changes. No visible skin irritation or abnormal marks were observed after removal. In addition, prior studies have widely reported polyacrylamide-based hydrogels incorporating PEDOT:PSS or LiCl for bioelectronics, supporting the general biocompatibility of the DMCH constituents^{R1–R3}.

For the Ag-PU conductor, HydroMed²⁴ is reported as biocompatible. However, because the conductor contains silver flakes and partial surface exposure of Ag cannot be fully excluded, we acknowledge that definitive biocompatibility of the Ag-PU composite cannot be claimed based on the present data. Nevertheless, the Ag-PU conductor is embedded within the textile architecture and is therefore unlikely to maintain prolonged, highly conformal contact with the skin compared to the protruding DMCH electrode layer. Additionally, prior studies have demonstrated the use of silver–polymer composites for skin-interfacing applications, particularly when Ag fillers are embedded within elastomeric matrices to mitigate direct exposure^{R4–R6}. We have revised the manuscript to clarify the current evidence and limitations.

- R1. Xie, Z. *et al.* Ultrastretchable, self-healable and adhesive composite organohydrogels with a fast response for human–machine interface applications. *Journal of Materials Chemistry C* **10**, 8266-8277 (2022).
- R2. Yun, T. G. *et al.* All-transparent stretchable electrochromic supercapacitor wearable patch device. *Acs Nano* **13**, 3141-3150 (2019).
- R3. Zhang, M. *et al.* Strong, conductive, and freezing-tolerant polyacrylamide/PEDOT:PSS/cellulose nanofibrils hydrogels for wearable strain sensors. *Carbohydrate Polymers* **305**, 120567 (2023).
- R4. Boda, U. *et al.* Screen-printed corrosion-resistant and long-term stable stretchable electronics based on AgAu microflake conductors. *ACS Applied Materials & Interfaces* **15**, 12372-12382 (2023).
- R5. Seo, H. *et al.* Self-packaged stretchable printed circuits with ligand-bound liquid metal particles in elastomer. *Nature Communications* **16**, 4944 (2025).
- R6. Ding, C. *et al.* Durability study of thermal transfer printed textile electrodes for wearable electronic applications. *ACS Applied Materials & Interfaces* **14**, 29144-29155 (2022).
24. Kim, J., Duong, H. D. & Rhee, J. I. Preparation and characterization of electrospun fluorescent fiber mats as temperature sensors using various polymers. *Polymer Testing* **122**, 108019 (2023).

Authors' modification to the manuscript:

(Page 18 in Revised Supplementary Information; add Supplementary Fig. 13)

Supplementary Fig. 13 | 24-h skin-contact biocompatibility assessment of DMCH and Ag-PU conductor

Photographs of the skin after 24 h continuous contact with DMCH and the Ag-PU conductor.

(Page 12; line 266 in Revised Manuscript)

“Consequently, the DMCH-based electrode—integrating both PEDOT:PSS and LiCl—demonstrated significantly higher current density compared to the ionic hydrogel and exhibited the lowest impedance across the low-frequency range (1–150 Hz) typical of TENS (Fig. 2g).

Biocompatibility was evaluated using a 24 h skin-contact test in which a participant wore a sleeve containing DMCH and the printed Ag-PU conductor during daily activities (Supplementary Fig. 13). No visible skin irritation was observed after removal, consistent with prior reports supporting the biocompatibility of polyacrylamide-based hydrogels incorporating PEDOT:PSS or LiCl^{R1-R3}. Because the Ag-PU conductor is embedded within the textile architecture, it is less able to maintain sustained, conformal skin contact than the protruding DMCH electrode layer. Prior studies have similarly used Ag-polymer composites for skin-interfacing applications, typically embedding Ag fillers within elastomeric matrices to minimize direct exposure^{R4-R6}.”

(Page 9; line 203 in Revised Manuscript)

“In this study, DMCH was employed as the base electrode. DMCH is synthesized by incorporating the conductive polymer poly(3,4-ethylenedioxythiophene):polystyrene sulfonate (PEDOT:PSS) with hygroscopic aqueous ionic salts, such as lithium chloride (LiCl), resulting in a **biocompatible**^{R1-R3} hydrogel with high electrical conductivity and low impedance. Structurally, DMCH consists of a cross-linked polymer network that retains a significant amount of water within its porous matrix²⁹.”

Comment #3: *The authors studied the sweat effect on the current and stimulation, based on the 10-min running condition. It will be better if the quantitative evaluation can be done to study the effect of sweat.*

Our response (*Comment #3*):

We thank the reviewer for this helpful comment. We agree that a quantitative evaluation of the sweat effect can further highlight the advantages of our breathable TESS. In the revised manuscript, we performed an experiment under environmental conditions (13 °C, 66 % relative humidity), in which participants ran on a track for 10 min followed by 10 min of rest.

To quantify sweat-induced effects, we analyzed the normalized stimulation current for each participant. The results revealed pronounced inter-subject variability for the commercial TENS electrodes. Specifically, for Participant 1, the current increased by 9.9× for TENS electrode-2 and 4.4× for TENS electrode-1. Participants 2 and 3 also showed increases ranging from 1.3× to 2.5× across the TENS electrodes. Such variability is expected because sweat secretion rate and electrolyte concentration can differ substantially among individuals^{R1}. In contrast, our textile-based electrode remained stable, with current increases of only 1.1×, 1.2×, and 1.1× for Participants 1–3, respectively.

Following these measurements, we performed a sensory evaluation that specifically assessed sweat-induced itchiness, distinct from stimulation intensity. On a scale from 0 (no sensation) to 10 (intolerable itchiness), participants reported elevated discomfort and itchiness with both TENS electrodes after sweating, with itchiness increasing at higher frequencies. By contrast, the textile-based electrode consistently maintained low itchiness scores, supporting stable and comfortable operation under sweat-inducing conditions.

We documented visible sweat accumulation by photography, quantified the corresponding current changes, and performed a post-activity sensory test. We revised the manuscript to report these sweat-induced changes in current and perceived sensation on a per-participant basis. We

believe these additional results strengthen the conclusion that TESS maintains comparatively stable electro-tactile output and user comfort under sweat-inducing conditions.

R1. Baker, L. B. Sweating rate and sweat sodium concentration in athletes: a review of methodology and intra/interindividual variability. *Sports Medicine* **47**, 111-128 (2017).

Authors' modification to the manuscript:

(Page 21; line 493 in Revised Manuscript)

“Post-exercise measurements revealed substantial inter-subject variability^{R1} and pronounced current amplification for the commercial electrodes. Specifically, the stimulation current increased by 1.4–4.4× for TENS electrode 1 and by 1.3–9.9× for TENS electrode 2 across participants, in some cases rendering stimulation difficult to tolerate due to elevated itchiness. This current increase is consistent with sweat accumulation at the skin–adhesive interface (Supplementary Fig. 21a,b). In contrast, the textile-based stimulation electrode showed only a 1.1–1.2× increase in current across all participants, consistent with improved breathability and moisture management. Participants were also asked to rate the sensation of itchiness associated with each electrode on a 0–10 scale following stimulation (Fig. 4i). Both TENS electrodes caused higher levels of itchiness after sweating, particularly across wider skin areas. Participants also reported itchiness levels corresponding to the current amplification. In contrast, the textile-based electrode received consistently low itchiness scores, confirming its safe and comfortable operation under sweat-inducing conditions.”

(Page 61 in Revised Manuscript)

Fig. 4 | User study of the multimodal textile-based electrical stimulation system

a, Schematic of the electrical impedance model of human skin, shown as a simplified equivalent circuit. **b**, Representative waveforms of voltage (top) and current (bottom) signals applied to the skin during electrical stimulation. **c**, Comparison of individual stimulus thresholds and maximum tolerable voltages across participants ($n = 15$). Squares indicate mean values; center lines show medians; box limits represent upper and lower quartiles; whiskers extend to $1.5\times$ the interquartile range. **d**, Log–log plot showing the linear relationship between perception threshold current and stimulation frequency. The fitted line has a constrained slope of 0.17076 with adjusted $R^2 > 0.9$ ($n = 9$). **e**, Mapping of four distinct tactile sensations—touch, tickling, roughness, and pressure—based on stimulation current and frequency ($n = 15$; data shown as violin plots). **f**, Confusion matrix showing recognition accuracy in 10 blind tests of forearm (posterior) stimulation without prior training. Participants identified sensations from four predefined categories. **g**, Photograph of the forearm showing placement of the textile-based electrode and two commercial TENS electrodes used to assess current variation before and after physical activity. **h**, Normalized current measurements before and after a 10-min run on a track ($n = 3$). **i**, Heat map showing subjective itchiness ratings following 10 min of exercise, evaluated for three different electrode types during delivery of various sensations ($n = 3$).

(Page 26 in Revised Supplementary Information; add Supplementary Fig. 21)

Supplementary Fig. 21 | Sweat accumulation after running

a, A photograph of the forearm before exercise. **b**, A photograph of the forearm after 10 min of running followed by 10 min of rest ($13\text{ }^{\circ}\text{C}$, 66% relative humidity), showing visible sweat accumulation.

Comment #4: *As an electronic cloth, how is the abrasion resistance, and is it washable?*

Our response (Comment #4):

We thank the reviewer for this important comment. Washability is a critical consideration for textile-based systems. In the revised manuscript, we printed 13 Ag-PU conductor patterns (length 30 mm, width 2 mm) on the textile substrate and subjected them to a washing test. Samples underwent four wash cycles of magnetic stirring in water (650 rpm, 10 min/cycle), which resulted in an approximately $4.8\times$ increase in conductor resistance.

We also repeated the same washing protocol after laminating DMCH onto the Ag-PU conductor to evaluate the durability of the integrated interface. Among the eight integrated

DMCH/Ag-PU samples, delamination occurred in seven samples during the second wash cycle, and the remaining sample failed during the first cycle. Because DMCH is a water-containing hydrogel electrode, direct water exposure during washing readily compromises interfacial bonding and leads to delamination. By contrast, the Ag-PU conductor showed resistance to abrasion and washing-induced mechanical stress under the present protocol. In line with prior strategies for improving the washability of Ag-based textile conductors, we expect durability can be further enhanced by strengthening conductor–textile interfacial adhesion^{R1} and by adopting patterning approaches that reduce wash-induced mechanical damage^{R2}. We have added these results to the revised manuscript to clearly delineate the current durability limitations and the associated pathways for improvement.

- R1. Bae, K. *et al.* Washable heat-resistant and inkjet-printed devices on cotton fabric for wearable applications. *Nature Communications* **16**, 8615 (2025).
- R2. Ding, C. *et al.* Durability study of thermal transfer printed textile electrodes for wearable electronic applications. *ACS Applied Materials & Interfaces* **14**, 29144-29155 (2022).

Authors' modification to the manuscript:

(Page 11 in Revised Supplementary Information; add Supplementary Fig. 6)

Supplementary Fig. 6 | Washability of screen-printed Ag-PU textile conductors

Change in resistance of Ag-PU conductors (30 mm length × 2 mm width) on textile after repeated washing.

(Page 8; line 175 in Revised Manuscript)

“Beyond conductivity, our ink enables hydrogen-bond-mediated adhesion to hydrophilic hydrogels without the additional surface treatments often required for hydrophobic Ag composites^{60–62}, and it can be fabricated without high-temperature post-processing, supporting scalable integration in textile-based wearable systems. To evaluate durability, 13 Ag-PU conductor patterns (30 mm × 2 mm) were fabricated on the textile and subjected to a washing test consisting of four 10-min cycles under magnetic stirring in water at 650 rpm (Supplementary Fig. 6).”

Comment #5: Textile is a type of material with complex features in material and structure, how is the printing accuracy of the conductive array on textiles? Any solutions could improve it and enhance the function of the electrical-stimulation suit? Please cite some important references these two years and illustrate the scientific points.

Our response (Comment #5):

We thank the reviewer for this helpful comment regarding printing accuracy of conductive arrays on textiles and potential strategies to further enhance electrical-stimulation suits. To assess printing accuracy, we screen-printed the Ag–PU conductor into 30 mm-long traces using stencil masks with nominal line widths of 100, 250, and 500 μm (Supplementary Fig. 3). All printed traces remained electrically conductive, and the resistance decreased with increasing line width, with average values of $\sim 40\ \Omega$ (100 μm), $\sim 18\ \Omega$ (250 μm), and $\sim 10\ \Omega$ (500 μm). Optical microscopy further showed printed widths of 108–233 μm for the 100 μm mask, 250–370 μm for the 250 μm mask, and widths close to the nominal value with comparatively uniform edges for the 500 μm mask. Collectively, these results demonstrate reliable electrical interconnection with sub-millimetre printed features on textiles.

Additionally, we agree that the intrinsic structural complexity of textiles poses significant challenges for high-fidelity printing of conductive patterns. Textiles are porous and three-dimensional, and they often exhibit substantial surface roughness; these features can lead to nonuniform ink deposition and reduced adhesion, particularly for fine-line features. Such effects can directly impact textile-based electrical-stimulation suits by causing nonuniform current distribution and increased variability in skin–electrode contact impedance. In future work, we anticipate improving printability on textiles by pursuing the strategies outlined below.

First, recent work on self-adhesive elastic conductive ink (ECI) demonstrates that tailoring ink rheology and interfacial adhesion can significantly improve pattern fidelity on porous fabrics. Authors report an ECI with high permeability into the textile thickness, but low lateral diffusivity, enabling direct printing of fine conductive traces on a wide range of textiles without severe edge bleeding or delamination under strain as shown in Figs below.

(Figure.5 in Ref, *ACS Nano*, 2024,18, 34750–34762)

Second, hybrid patterning based on in-textile photolithography has demonstrated sub-100 μm -scale metal features within woven fabrics while preserving air and moisture permeability. In this approach, metal is conformally deposited along individual fibers inside the textile scaffold, enabling line/space dimensions of ~ 100 –200 μm with low resistance and good wash durability. This capability surpasses the practical resolution limit of conventional on-textile screen printing ($\sim 400\ \mu\text{m}$) and would enable denser interconnects and electrode arrays for spatially resolved electrical stimulation. We anticipate that applying a thin composite coating to the textile could make this in-textile photolithography approach compatible with our platform, providing finer conductive features while maintaining the flexibility required for wearable suits.

(Figure.1 in Ref, *Nature Communications*, 2024, 15, 887)

Third, thermal transfer printing offers a practical route to decouple high-resolution circuit fabrication from the intrinsic roughness of textiles. In this study, silver-flake/TPU electrodes are first patterned on a smooth release film and then laminated onto various fabrics by hot pressing. The resulting textile electrodes exhibit high conductivity ($\sim 5 \times 10^4$ S/cm), strong adhesion, excellent washing resistance, and patterning resolution on the order of 40 μm , all while retaining fabric stretchability and comfort. This approach can be directly applied to electrical-stimulation suits to provide even distribution of current across the electrodes and maintain consistent, stable contact with the skin

(Figure.2 in Ref, *ACS Appl. Mater. Interfaces*, 2022,14, 29144–29155)

Together, these three strategies, such as rheology-engineered inks, in-textile photolithographic hybrid patterning, and thermal transfer of pre-patterned stretchable conductors, show that the printing accuracy of conductive arrays on textiles can be substantially improved despite the substrate complexity. For future developments of our electrical-stimulation suit, adopting such approaches would allow finer-pitch electrode arrays with more homogeneous impedance, which is expected to enhance stimulation selectivity, and improve overall user comfort and device reliability.

In the revised manuscript, we added a discussion of practical approaches for improving the printing accuracy of conductive arrays on textiles and reported the printing accuracy of our electrodes based on additional characterization results.

- R1. Zhu, L. *et al.* Self-Adhesive Elastic Conductive Ink with High Permeability and Low Diffusivity for Direct Printing of Universal Textile Electronics. *ACS nano* 18, 34750-34762 (2024).

- R2. Wang, P. *et al.* Well-defined in-textile photolithography towards permeable textile electronics. *Nature communications* **15**, 887 (2024).
- R3. Ding, C. *et al.* Durability study of thermal transfer printed textile electrodes for wearable electronic applications. *ACS Applied Materials & Interfaces* **14**, 29144-29155 (2022).

Authors' modification to the manuscript:

(Page 8 in Revised Supplementary Information; add Supplementary Fig. 3)

Supplementary Fig. 3 | Printing accuracy evaluation via resistance comparison across pattern widths

a, Optical microscopy images of printed widths of 108–233 μm (100 μm mask), 250–370 μm (250 μm mask), and ~500 μm with relatively uniform edges (500 μm mask). **b**, The resistance of 30 mm-long printed traces measured as a function of nominal pattern width (100, 250, and 500 μm).

(Page 6; line 134 in Revised Manuscript)

“During the coating process, the ink’s capillary action promotes deep penetration into the PU textile substrate, resulting in a durable bonding layer²³ Cross-sectional scanning electron microscopy (SEM) analysis confirmed that the Ag-PU conductors infiltrate the textile to a depth of approximately 160 μm, effectively anchoring within the 360 μm-thick PU matrix to form mechanically robust interfaces (Supplementary Fig. 2).

To assess printing accuracy on the textile substrate, Ag-PU conductors (30 mm trace length) were screen-printed using stencil masks with nominal line widths of 100, 250, and 500 μm (Supplementary Fig. 3a,b). All traces remained electrically conductive, and resistance decreased with increasing line width, confirming reliable interconnection and defining the achievable printing fidelity for textile-based conductive arrays. While prior studies report improved pattern fidelity using approaches such as rheology-engineered inks, in-textile photolithography, or thermal transfer printing^{R1–R3}, these methods typically require additional materials, specialized equipment, and/or extra processing steps. Screen printing was therefore selected as a practical compromise between scalability, throughput, and cost for large-area textile fabrication, while remaining compatible with commercial fabrics.”

Comment #6: *Is this electrode recipe and system design applicable to other textile substrates that have diverse materials, or different woven structures? Please show some examples and discuss.*

Our response (Comment #6):

We thank the reviewer for this thoughtful comment regarding the generalizability of our electrode recipe and system design across diverse textile substrates. In the revised manuscript, we first characterized the textiles used in this study by optical microscopy (Supplementary Fig.

8). We evaluated three substrates with distinct compositions: Textile 1 (87 % polyester, 13 % spandex), Textile 2 (90 % nylon, 10 % polyurethane), and Textile 3 (88 % polyester, 12 % polyurethane).

To assess compatibility of the Ag-PU conductor with these varying textile structures, we fabricated five replicate patterns (length 30 mm, width 2 mm) on each substrate under identical processing conditions and quantified electrical resistance. The mean resistances were 1.90 Ω (Textile 1), 1.96 Ω (Textile 2), and 2.08 Ω (Textile 3). The small spread across substrates indicates that the electrical performance of the Ag-PU conductor is maintained across these textile architectures under matched fabrication conditions.

We attribute this substrate-independent behaviour to the Ag-PU composite forming a conformal, percolated conductive network in which the effective path resistance is governed primarily by printed feature geometry and filler loading, rather than by the underlying textile topography. Accordingly, the Ag-PU conductor exhibits performance that is comparatively insensitive to substrate choice, supporting its applicability across diverse materials and weave structures.

Authors' modification to the manuscript:

(Page 13 in Revised Supplementary Information; add Supplementary Fig. 8)

Supplementary Fig. 8 | Electrical performance of Ag-PU conductors on different textile substrates

a, Optical microscopy images of the textile substrate. The substrates were: Textile 1 (87% polyester, 13% spandex), Textile 2 (90% nylon, 10% polyurethane), and Textile 3 (88% polyester, 12% polyurethane). **b**, Absolute resistance of Ag-PU patterns (length 30 mm, pattern width 2 mm) printed on three textile substrates with varying compositions and weave/knit structures. The resulting mean resistances (1.90 Ω , 1.96 Ω , and 2.08 Ω , respectively) indicate that the conductor maintains consistent electrical performance across different substrate structures.

(Page 9; line 194 in Revised Manuscript)

“Consequently, higher Ag volume fractions lead to improved initial conductivity and faster resistance stabilization. Additionally, as shown in Supplementary Figure 8a,b, when the same Ag-PU conductor was screen-printed onto three textile substrates with distinct woven structures, the resulting traces exhibited similar absolute resistances (1.90 Ω , 1.96 Ω , and 2.08 Ω for Textiles 1–3, respectively), indicating that the electrical performance in this work is largely insensitive to the underlying textile substrate.”

Comment #7: *In practical application, there will be displacements between the cloth and*

human skin to affect the electrical stimulation and feedback effect, how to avoid it?

Our response (*Comment #7*):

We thank the reviewer for this thoughtful comment. We agree that displacement between the garment and the skin is a critical engineering challenge that can degrade electrical-stimulation stability. Our system uses a tight-fitting, highly stretchable base garment (e.g., a small-sized undergarment or sleeve) to prioritize comfort while establishing a consistent baseline contact pressure.

Nevertheless, residual variations in skin–electrode impedance remain inevitable due to inter-individual body geometry and dynamic motion during use. To address this variability, we implemented an integrated feedback compensation system that continuously monitors the interface state and adaptively adjusts stimulation parameters. By compensating for fluctuations that cannot be fully eliminated through passive fit alone, the proposed framework supports precise and consistent electrotactile feedback under practical use conditions. We have revised the manuscript accordingly to incorporate these clarifications.

Authors' modification to the manuscript:

(Page 24; line 555 in Revised Manuscript)

“Given the extensive surface area of the suit, garment pressure serves as an effective and scalable solution for electrode attachment. **While our system uses a tight-fitting garment to establish a baseline contact pressure and mitigate gross instability while maintaining user comfort, residual variability in skin–electrode impedance remains unavoidable due to inter-individual differences in body geometry and site-dependent variations in electrode placement.**”

Comment #8: *How is the feedback accuracy and efficiency of this electrical stimulation suit with other reported works? It is better to make a figure or table for quantitative comparison.*

Our response (*Comment #8*):

We thank the reviewer for this helpful suggestion. In the revised manuscript, we added Supplementary Table 5 to provide a quantitative comparison between TESS and representative haptic systems reported in prior work. This table summarizes, for each system, whether feedback compensation is implemented, the number and area of stimulation sites, the targeted body region (e.g., fingertip, hand, arm), and the substrate type and stretchability.

Our review indicates that most haptic systems that implement feedback compensation are demonstrated in highly localized body regions, such as the hand or fingertip, where dense stimulation matrices can be implemented over a small area. In contrast, our TESS is implemented as a body-scale suit targeting larger regions (arm and torso), where the two-point discrimination threshold is substantially larger than that of the fingertip/hand. This difference in anatomical targets necessitates a different design space for feedback compensation, in which the primary requirements shift from local, high-spatial-resolution control to large-area coverage and uniformity.

To the best of our knowledge, we did not identify prior examples of a body-scale haptic suit that integrates electrotactile feedback using a textile-based, stretchable electrode system. Accordingly, we revised the manuscript to position our work as an initial system-level demonstration toward this direction, while noting that task-specific feedback accuracy metrics are not yet standardized across studies and remain an important topic for future work.

Authors' modification to the manuscript:

(Page 46 in Revised Supplementary Information; add Supplementary Table 5)

reference	Feedback compensation	Stimuli point	Target	Stimulation coverage area	Device substrate	Haptic method	stretchability
[R1]	x	21	Forearm	Medium	silicone rubber	electrotactile	High
[R2]	x	9	Hand	Medium	Silicone	Pneumatic & vibrotactile	High
[R3]	Current monitoring	25	Fingertip	Small	Flexible PCB	electrotactile	Medium
[R4]	x	32	Hand	Medium	PDMS	electrotactile	Medium
[R5]	x	6	Hand	Medium	Fabric	electrotactile	Medium
[R6]	x	4	Arm	Small	Polyimide	electrotactile	Medium
[R7]	x	7	Fingertip	Small	Silicone membrane	electrostatic	High
[R8]	x	25	Any part	Medium	PET & Al	electrostatic	Medium
[R9]	Detecting pressure through the pressure-sensitive transistor	100	Fingertip	Small	Glass	electrotactile	Low
This work	Resistive pressure sensing	4	Full-body	Large	PU-based textile	electrotactile	High

Supplementary Table 5 | Comparison of TESS with previously reported haptic systems. Summary of representative haptic systems reported in the literature, including whether feedback compensation is implemented, the number and area of stimulation sites, the targeted body region (e.g., fingertip, hand), and the device substrate type and stretchability.

- R1. Shi, Y. *et al.* Self-powered electro-tactile system for virtual tactile experiences. *Science Advances* **7**, eabe2943 (2021).
- R2. Liu, M. *et al.* Tactile sensing and rendering patch with dynamic and static sensing and haptic feedback for immersive communication. *ACS Applied Materials & Interfaces* **16**, 53207-53219 (2024).
- R3. Lin, W. *et al.* Super-resolution wearable electrotactile rendering system. *Science advances* **8**, eabp8738 (2022).
- R4. Yao, K. *et al.* Encoding of tactile information in hand via skin-integrated wireless haptic interface. *Nature Machine Intelligence* **4**, 893-903 (2022).
- R5. Xu, G. *et al.* Self-powered electrotactile textile haptic glove for enhanced human-machine interface. *Science advances* **11**, eadt0318 (2025).
- R6. Xu, B. *et al.* An epidermal stimulation and sensing platform for sensorimotor prosthetic control, management of lower back exertion, and electrical muscle activation. *Advanced Materials (Deerfield Beach, Fla.)* **28**, 4462 (2015).
- R7. Chen, S., Chen, Y., Yang, J., Han, T. & Yao, S. Skin-integrated stretchable actuators toward skin-compatible haptic feedback and closed-loop human-machine interactions. *npj Flexible Electronics* **7**, 1 (2023).
- R8. Leroy, E. & Shea, H. Hydraulically amplified electrostatic taxels (HAXELs) for full body haptics. *Advanced Materials Technologies* **8**, 2300242 (2023).
- R9. Lim, K. *et al.* Interference haptic stimulation and consistent quantitative tactility in transparent electrotactile screen with pressure-sensitive transistors. *Nature communications* **15**, 7147 (2024).

“Without feedback, perceived intensity increased with garment pressure toward position 4; in contrast, the compensated condition produced more stable and comfortable sensations across positions, supporting that the proposed feedback improves the consistency of perceived intensity. In addition, Supplementary Table 5 provides a quantitative and qualitative comparison of TESS with representative haptic systems reported in prior work. This comparison highlights that TESS enables body-scale feedback, a capability that, to our knowledge, has not been demonstrated in previous electrotactile systems.”

Comment #9: Line 200, is there any evidence or reference for the hydrogen bond between DMCH and Ag-PU?

Our response (Comment #9):

We appreciate the reviewer’s insightful question regarding the evidence for hydrogen bonding between DMCH and Ag-PU. Herein, we provide relevant references and related results that support the presence of hydrogen bonding interactions as a key factor enhancing the interfacial adhesion and mechanical integrity of the Ag-PU composite system. In previous work (*Nano-Micro Lett.*, **2023**, 15, 8), self-adhesive elastic conductive inks highlights that the hydroxyl (-OH), amino (-NH₂), and carbonyl groups (-CO) present in Carboxyethyl Chitin (CECT) can form strong hydrogen bonds with the amino groups (-NH₂) of PAAM, leading to enhanced mechanical strength and improved interfacial bonding in their composites. This interaction was confirmed via spectroscopic shifting of N-H and C=O vibrational peaks, indicating stable hydrogen bonding networks between PAAM and TPU components. Based on this result, Carbonyl (C=O) group presented in our HydroMed polymer can form strong hydrogen bond with amino group(-NH₂) in PAAM. Accordingly, we revised the manuscript to incorporate mechanistic discussion and supporting literature evidence that clarify the role of hydrogen bonding in our interfacial design.

(Figures 1,2 in Ref, *Nano-Micro Lett.*, **2023**, 15, 8)

R1. Zhang, J., Hu, Y., Zhang, L., Zhou, J. & Lu, A. Transparent, ultra-stretching, tough, adhesive carboxyethyl chitin/polyacrylamide hydrogel toward high-performance soft electronics. *Nano-micro letters* **15**, 8 (2023).

Authors’ modification to the manuscript:

(Page 10; line 231 in Revised Manuscript)

“The polymer chains become grafted onto the elastomeric surface of the Ag-PU, forming hydrogen bonds between urethane groups (NH-COO), which further reinforce the interface with heat (Fig. 2c)^{R1}. A peel test confirmed the mechanical robustness of the laminated interface, demonstrating superior adhesion compared to conventional Ag-ink/silicone rubber (Ecoflex-Ag) composites (Fig. 2d).”

Comment #10: Regarding to line 139, will the water absorption and hydrophilic property of HydroMed cause the oxidation and degradation of Ag particles? Maybe the resistance of the printed electrode can be measured after keeping in a wet condition for a relatively long time to see the durability.

Our response (Comment #10):

We thank the reviewer for this important comment. We agree that the water absorption and hydrophilic nature of HydroMed could, in principle, accelerate Ag degradation by sustaining a wet interface and facilitating ionic transport. Prior studies^{R1,R2} have reported that Ag-based conductors can exhibit resistance increase under prolonged humidity or liquid exposure, and that chloride-containing electrolytes can further promote Ag oxidation and AgCl formation, both of which can degrade electrical performance.

To directly evaluate durability under wet exposure in our system, we quantified the resistance change of the printed Ag-PU conductor after a controlled water-exposure protocol. Specifically, we printed 13 Ag-PU conductor patterns (length 30 mm, width 2 mm) on the textile substrate and subjected them to a wet-condition protocol comprising four cycles of magnetic stirring in water (650 rpm, 10 min/cycle). After this wet exposure, the conductor resistance increased by approximately 4.8×. The patterns remained electrically continuous after the wet exposure, indicating a measurable level of resistance to moisture/oxidation-driven degradation under the present conditions. Consistent with established strategies for improving the water durability of Ag-based textile conductors, further improvements are expected by strengthening conductor–textile interfacial adhesion^{R3} and by adopting patterning routes that reduce water-induced mechanical damage^{R4}. We added these results to the revised manuscript to clarify the current durability limitations under wet exposure.

- R1. Jiu, J. *et al.* The effect of light and humidity on the stability of silver nanowire transparent electrodes. *Rsc Advances* **5**, 27657-27664 (2015).
- R2. Mayousse, C., Celle, C., Frackiewicz, A. & Simonato, J.-P. Stability of silver nanowire based electrodes under environmental and electrical stresses. *Nanoscale* **7**, 2107-2115 (2015).
- R3. Bae, K. *et al.* Washable heat-resistant and inkjet-printed devices on cotton fabric for wearable applications. *Nature Communications* **16**, 8615 (2025).
- R4. Ding, C. *et al.* Durability study of thermal transfer printed textile electrodes for wearable electronic applications. *ACS Applied Materials & Interfaces* **14**, 29144-29155 (2022).

Authors' modification to the manuscript:

(Page 11 in Revised Supplementary Information; add Supplementary Fig. 6)

Supplementary Fig. 6 | Washability of screen-printed Ag-PU textile conductors

Change in resistance of Ag-PU conductors (30 mm length × 2 mm width) on textile after repeated washing.

(Page 8; line 175 in Revised Manuscript)

“Beyond conductivity, our ink enables hydrogen-bond-mediated adhesion to hydrophilic hydrogels without the additional surface treatments often required for hydrophobic Ag composites^{60–62}, and it can be fabricated without high-temperature post-processing, supporting scalable integration in textile-based wearable systems. To evaluate durability, 13 Ag-PU conductor patterns (30 mm × 2 mm) were fabricated on the textile and subjected to a washing test consisting of four 10-min cycles under magnetic stirring in water at 650 rpm (Supplementary Fig. 6).”

Comment #11: Line 47, XR and should be defined at its first appearance in the main text (not only defined in the abstract).

Our response (Comment #11):

We thank the reviewer for this helpful comment. We agree that extended reality (XR) should be explicitly defined for clarity, and we have revised the manuscript to define XR at its first occurrence in the main text, consistent with the definition provided in the Abstract.

Authors’ modification to the manuscript:

(Page 3; line 46 in Revised Manuscript)

“Electrical stimulation offers significant potential across diverse applications, including haptic feedback in extended reality (XR) systems and therapeutic interventions. When integrated with virtual or augmented reality, XR-assisted therapy combines immersive environments with targeted electrical stimulation to enhance rehabilitation, pain management, and neuromuscular recovery^{1,2,3}.”

Comment #12: In Figure 2a, the red area in the SEM is not very accurate, and the Ag-PU electrode layer has not been labeled. It is suggested to present Figure 2 in three lines. Additionally, the scale bars in Figure 1b, 2a, 2d, 5c should be added. In Figure S2, what is the element that each color represents in the EDS?

Our response (Comment #12):

We thank the reviewer for this helpful comment. We agree that the original SEM image was not optimally suited for that figure panel. Accordingly, we replaced Figure 2a with a clearer

schematic illustration, and we believe that microstructural details are now sufficiently conveyed by the updated Figure 2b and Supplementary Figure 2.

In addition, as suggested, we reformatted Figure 2 into three rows and added scale bars to Figures 1b, 2a, 2d, and 5c to improve clarity. We also revised the caption of Supplementary Figure 2 to clarify that yellow corresponds to the Ag L series and the green corresponds to the O K series in the EDS mapping.

Authors' modification to the manuscript:

(Page 57 in Revised Manuscript)

Fig. 2 | Characterization of the electrical stimulation electrode

a, Schematic diagram of a single electrical stimulator showing the DMCH and the Ag-PU conductor printed on a PU-based textile. **b**, Cross-sectional SEM image showing seamless integration of the DMCH on the Ag-PU conductor printed onto a PU-based textile. No interfacial gaps were observed. Scale bars, 100 μm . **c**, Schematic showing the mechanism of robust adhesion between DMCH and the Ag-PU conductor. **d**, Photographs of the peeling test used to compare the peeling force of the DMCH/Ag-PU conductor with that of the DMCH/silicon elastomer. Scale bars, 20 mm. **e**, Peeling forces per width of DMCH attached to the Ag-PU conductor and silicon elastomer. **f**, Current density measured on skin as a function of frequency for the Ag-PU conductor (used as the interconnect) and three electrical stimulation electrodes with different hydrogel compositions. **g**, Stress as a function of strain for conductive hydrogel (red), hydrogel (black), and ionic hydrogel (blue). The DMCH is capable of withstanding strains of up to 600%. **h**, Impedance spectra of various hydrogel electrodes with different compositions: pure hydrogel (black), ionic hydrogel (blue), and DMCH (red). The DMCH formed by mixing PEDOT and ions exhibited the lowest impedance, owing to the doping effect of the ions on PEDOT.

Fig. 1 | Design of a large-area textile-based electrical stimulation feedback suit
a, Conceptual illustration of the textile-based electrical stimulation feedback suit. The suit is constructed using a lightweight, breathable polyurethane (PU)-based textile, which delivers four distinct electrotactile stimuli to the skin through a dehydration-resistant hydrogel under appropriate compression. The system enables immersive interaction within virtual reality (VR) environments. **b**, Flowchart illustrating the real-time haptic feedback mechanism on the forearm. Integrated resistive pressure sensors detect localized pressure variations and adjust stimulation intensity accordingly, providing personalized tactile feedback based on the user's unique sensory response. **Scale bars, 5 mm.** **c**, Illustration and photograph of a participant wearing the feedback suit. Due to variations in body shape, different pressure levels (low on the left and high on the right) result in changes in current delivery. The system dynamically senses clothing pressure and modulates output voltage in real time to maintain consistent tactile stimulation across different body regions.

Fig. 5 | User study on the full-body electrical stimulation suit

a, Photograph of a participant wearing the textile-based electrical stimulation suit designed for the chest, abdomen, sides, shoulders, and arms. The suit provides targeted compression and high spatial resolution for full-body stimulation. **b**, Schematic illustrating the electrical stimulation mechanism, showing how current levels vary in response to changes in applied pressure (p), contact area (a), and interelectrode distance (d) under low-pressure (top) and high-pressure (bottom) conditions. **c**, Photograph of the integrated pressure sensor embedded in PU-based fabric with a conductive hydrogel for real-time pressure sensing (left). Detection range of the sensor shown on the right. Inset: image of a 4×1 unencapsulated pressure sensor array and a 4×3 tactile stimulator array. Scale bar, 10 mm. **d**, Normalized pressure changes measured by an interdigitated pressure sensor while participants wore a sleeve. Measurement locations included: wrist (1), mid-forearm (2), below the elbow (3), and above the elbow (4), using the elbow as the central reference point ($n = 9$ repetitions of donning and doffing; bar height: mean; error bars, s.d.). **e**, Measured clothing pressure across different body locations (as shown in panel a) for five participants (three female and two male) with varying body types and ages. **f**, Flowchart illustrating the real-time electrical stimulation feedback system. A mobile device sets

the default resistance of the pressure sensor. Real-time clothing pressure at each arm location is captured via four ADC inputs, and normalized pressure values are used to calculate the required stimulation voltage. The selected stimulation channel determines frequency, and the output voltage is computed using a pre-calibrated formula. **g**, Three distinct tactile sensations were simulated through real-time tactile feedback. Wireless pressure sensors positioned across the body calculate garment pressure, triggering spatiotemporal activation of stimulation channels. Before feedback, the system considers both the threshold current and additional current variation with respect to the frequency. After feedback integrates all key parameters—including pressure-induced current adjustments—and enables real-time feedback and dynamic stimulation to accommodate individual users. **h**, Heatmap of sleeve compression levels across arm positions 1–4 in a within-subject experiment. The sleeve on the left arm was operated without feedback compensation, whereas the sleeve on the right arm was operated with feedback compensation. **i**, Heatmap of perceived intensity ratings across arm positions 1–4 in the same within-subject experiment (0–10 scale; 0: no sensation, 5: comfortable/moderate, 10: intolerable). The after-feedback condition shows more stable and comfortable perceived intensity compared to the before-feedback condition.

(Page 7 in Revised Supplementary Information)

Supplementary Fig. 2 | EDS maps of PU-based textile printed with Ag-PU conductor
EDS maps of the cross-section area in about the distribution Ag-PU conductor (top) and PU-based textile (bottom). **In the EDS mapping, the yellow signal corresponds to the Ag L-series, and the green signal represents the O K-series.** The Ag-PU conductor infiltrates the PU-based textile, resulting in a relatively dense and uniformly distributed top layer. Scale bar, 200 μm .

Response to Referees Letter

Referee #1:

Summary Comments: *The authors have addressed all my concerns. It's a very interesting work by using full body ES. I recommend the acceptance of the manuscript as is.*

Our response:

We sincerely thank the reviewer for the positive evaluation and for highlighting the key contributions of our manuscript. The reviewer's constructive feedback has significantly improved the quality, clarity, and overall completeness of the revised manuscript. We are also grateful for the encouraging assessment and the recommendation to accept the manuscript.

Referee #2:

Summary Comments: *The authors have addressed my previous comments and improved the manuscript. However, there are still some writing and formatting issues in the revised manuscript that should be addressed to ensure a high-quality presentation, meeting the high standards of Nat. Commun., Please see the issues as follows.*

Our response:

We sincerely thank the reviewer for the careful evaluation of our revised manuscript and for the constructive comments on remaining writing and formatting issues, which help further improve the quality and presentation to meet the high standards of Nature Communications. We have carefully addressed all of the points raised by the reviewer and revised the manuscript accordingly, as detailed below.

Comment #1: *Too many references are cited to support the revision, which highly reduces the continuity of the manuscript. It is suggested to concisely integrate the revisions in the text, and relocate the current detailed optimization-related discussion and the corresponding references to Supporting Information, such as the points as follows:*

- (1) The discussion of printing accuracy (pages 6-7);*
- (2) The conductivity comparison of Ag-PU conductors with other Ag-based inks for e-textiles (page 8);*
- (3) The optimization of the DMCH electrode thickness (page 11);*

Our response (Comment #1):

We appreciate the reviewer's suggestion to improve the continuity of the manuscript. We agree that the revised version included overly detailed discussion and a number of references in the main text. Accordingly, we have relocated the relevant Results content and corresponding references to the Supplementary Information, as suggested.

Specifically, (1) the discussion of printing accuracy (previously on pages 6–7) has been moved to Supplementary Note 1. (2) The conductivity benchmarking of the Ag-PU conductors against previously reported silver-based e-textile inks (previously on page 8) has been moved from the main text to Supplementary Note 2. (3) The detailed optimization of the DMCH electrode thickness (previously on page 11) has been moved to Supplementary Note 3, and the associated references have all been relocated to the Supplementary Information (page 57).

Authors' modification to the manuscript:

(Page 3 in Revised Supplementary Information; add Supplementary Note 1)

“Supplementary Note 1. Printing accuracy on textile of Ag-PU conductor

To assess printing accuracy on the textile substrate, Ag-PU conductors (30 mm trace length)

were screen-printed using stencil masks with nominal line widths of 100, 250, and 500 μm (Supplementary Fig. 3a,b). All traces remained electrically conductive, and resistance decreased with increasing line width, confirming reliable interconnection and defining the achievable printing fidelity for textile-based conductive arrays. While prior studies report improved pattern fidelity using approaches such as rheology-engineered inks, in-textile photolithography, or thermal transfer printing¹⁻³, these methods typically require additional materials, specialized equipment, and/or extra processing steps. Screen printing was therefore selected as a practical compromise between scalability, throughput, and cost for large-area textile fabrication, while remaining compatible with commercial fabrics.”

(Page 4 in Revised Supplementary Information; add Supplementary Note 2)

“Supplementary Note 2. Electrical properties of printed Ag-PU conductors

Supplementary Figure 5 shows the absolute resistance of the printed Ag-PU conductors as a function of pattern geometry. For the 2 mm trace width used throughout this study, a 30 mm-long conductor exhibits an average resistance of $\sim 1.9 \Omega$. We also benchmarked the conductivity of our Ag-PU conductors against previously reported Ag-based inks for textile electronics (Supplementary Table 1) confirming that our conductivity is comparable to representative textile-based Ag inks reported in prior work. Beyond conductivity, our ink enables hydrogen-bond-mediated adhesion to hydrophilic hydrogels without the additional surface treatments often required for hydrophobic Ag composites⁴⁻⁶, and it can be fabricated without high-temperature post-processing, supporting scalable integration in textile-based wearable systems. To evaluate durability, 13 Ag-PU conductor patterns (30 mm \times 2 mm) were fabricated on the textile and subjected to a washing test consisting of four 10-min cycles under magnetic stirring in water at 650 rpm (Supplementary Fig. 6).”

(Page 5 in Revised Supplementary Information; add Supplementary Note 3)

“Supplementary Note 3. The optimization of the DMCH electrode thickness

The DMCH electrode thickness was optimized by balancing mechanical integrity and electrical stability under compression. As shown in Supplementary Figure 11, the 1 mm DMCH could not sustain stable measurements above 1.3 N, whereas the 2 mm DMCH remained mechanically intact and electrically stable up to 5 N, exhibiting the smallest resistance change. Consistently, Supplementary Figure 12 shows that the 1 mm DMCH has lower conductivity ($\sigma \approx 0.025 \text{ mS}\cdot\text{cm}^{-1}$) than the thickness-insensitive regime observed for thicker DMCH ($\sigma \approx 0.12\text{--}0.14 \text{ mS}\cdot\text{cm}^{-1}$), consistent with prior reports on PEDOT:PSS-based electrodes⁷⁻¹². Together, these results indicate that 2 mm is the minimum thickness required to ensure a robust conductive path and reliable electrical performance under compressive loading up to 5 N.”

(Page 57 in Revised Supplementary Information; add Reference)

Reference

- 1 Zhu, L. *et al.* Self-Adhesive Elastic Conductive Ink with High Permeability and Low Diffusivity for Direct Printing of Universal Textile Electronics. *ACS nano* **18**, 34750-34762 (2024).
- 2 Wang, P. *et al.* Well-defined in-textile photolithography towards permeable textile electronics. *Nature communications* **15**, 887 (2024).
- 3 Ding, C. *et al.* Durability study of thermal transfer printed textile electrodes for wearable electronic applications. *ACS Applied Materials & Interfaces* **14**, 29144-29155 (2022).

- 4 Kim, S. H. *et al.* Ultrastretchable conductor fabricated on skin-like hydrogel–elastomer hybrid substrates for skin electronics. *Advanced Materials* **30**, 1800109 (2018).
- 5 Jin, H. *et al.* Highly durable nanofiber-reinforced elastic conductors for skin-tight electronic textiles. *Acs Nano* **13**, 7905-7912 (2019).
- 6 Matsuhisa, N. *et al.* Printable elastic conductors with a high conductivity for electronic textile applications. *Nature communications* **6**, 7461 (2015).
- 7 Maeda, R. *et al.* The conducting fibrillar networks of a PEDOT: PSS hydrogel and an organogel prepared by the gel-film formation process. *Nanotechnology* **32**, 135403 (2021).
- 8 Greczynski, G., Kugler, T. & Salaneck, W. Characterization of the PEDOT-PSS system by means of X-ray and ultraviolet photoelectron spectroscopy. *Thin Solid Films* **354**, 129-135 (1999).
- 9 Andrei, V. *et al.* Size Dependence of Electrical Conductivity and Thermoelectric Enhancements in Spin-Coated PEDOT: PSS Single and Multiple Layers. *Advanced Electronic Materials* **3**, 1600473 (2017).
- 10 Nardes, A., Kemerink, M. & Janssen, R. Anisotropic hopping conduction in spin-coated PEDOT: PSS thin films. *Physical Review B—Condensed Matter and Materials Physics* **76**, 085208 (2007).
- 11 Carter, J. L., Kelly, C. A., Marshall, J. E. & Jenkins, M. J. Effect of thickness on the electrical properties of PEDOT: PSS/Tween 80 films. *Polymer Journal* **56**, 107-114 (2024).
- 12 Dijk, G., Ruigrok, H. J. & O'Connor, R. P. Influence of PEDOT: PSS coating thickness on the performance of stimulation electrodes. *Advanced Materials Interfaces* **7**, 2000675 (2020).
- 13 Wicaksono, I. *et al.* A tailored, electronic textile conformable suit for large-scale spatiotemporal physiological sensing in vivo. *npj Flexible Electronics* **4**, 1-13 (2020).
- 14 Zhao, L., Li, X., Yu, J., Li, C. & Li, G. Compression sleeves design based on Laplace laws. *Journal of Textile Engineering & Fashion Technology* **2**, 314-320 (2017).
- 15 Accot, J. & Zhai, S. in *Proceedings of the ACM SIGCHI Conference on Human factors in computing systems*. 295-302.
- 16 Zhu, C.-H., Li, L.-M., Wang, J.-H., Wu, Y.-P. & Liu, Y. Three-dimensional highly conductive silver nanowires sponges based on cotton-templated porous structures for stretchable conductors. *RSC Advances* **7**, 51-57 (2017).
- 17 Lv, J. *et al.* Printed sustainable elastomeric conductor for soft electronics. *Nature Communications* **14**, 7132 (2023).
- 18 Zhao, H. *et al.* Ultrastretchable and washable conductive microtextiles by coassembly of silver nanowires and elastomeric microfibers for epidermal human–machine interfaces. *ACS Materials Letters* **3**, 912-920 (2021).
- 19 Zhu, H.-W. *et al.* Printable elastic silver nanowire-based conductor for washable

- electronic textiles. *Nano Research* **13**, 2879-2884 (2020).
- 20 Cheng, Y., Wang, R., Sun, J. & Gao, L. Highly conductive and ultrastretchable electric circuits from covered yarns and silver nanowires. *ACS nano* **9**, 3887-3895 (2015).
- 21 Niu, B., Hua, T. & Xu, B. Robust deposition of silver nanoparticles on paper assisted by polydopamine for green and flexible electrodes. *ACS Sustainable Chemistry & Engineering* **8**, 12842-12851 (2020).
- 22 Lu, Y. *et al.* High-performance stretchable conductive composite fibers from surface-modified silver nanowires and thermoplastic polyurethane by wet spinning. *ACS applied materials & interfaces* **10**, 2093-2104 (2018).
- 23 Park, M. *et al.* Highly stretchable electric circuits from a composite material of silver nanoparticles and elastomeric fibres. *Nature nanotechnology* **7**, 803-809 (2012).
- 24 La, T. G. *et al.* Two-layered and stretchable e-textile patches for wearable healthcare electronics. *Advanced healthcare materials* **7**, 1801033 (2018).
- 25 Kim, S., Lee, S. & Jeong, W. EMG measurement with textile-based electrodes in different electrode sizes and clothing pressures for smart clothing design optimization. *Polymers* **12**, 2406 (2020).
- 26 Kim, H., Rho, S., Han, S., Lim, D. & Jeong, W. Fabrication of textile-based dry electrode and analysis of its surface EMG signal for applying smart wear. *Polymers* **14**, 3641 (2022).
- 27 Nigusse, A. B., Malengier, B., Mengistie, D. A., Tseghai, G. B. & Van Langenhove, L. Development of washable silver printed textile electrodes for long-term ECG monitoring. *Sensors* **20**, 6233 (2020).
- 28 Shi, Y. *et al.* Self-powered electro-tactile system for virtual tactile experiences. *Science Advances* **7**, eabe2943 (2021).
- 29 Liu, M. *et al.* Tactile sensing and rendering patch with dynamic and static sensing and haptic feedback for immersive communication. *ACS Applied Materials & Interfaces* **16**, 53207-53219 (2024).
- 30 Lin, W. *et al.* Super-resolution wearable electrotactile rendering system. *Science advances* **8**, eabp8738 (2022).
- 31 Yao, K. *et al.* Encoding of tactile information in hand via skin-integrated wireless haptic interface. *Nature Machine Intelligence* **4**, 893-903 (2022).
- 32 Xu, G. *et al.* Self-powered electrotactile textile haptic glove for enhanced human-machine interface. *Science advances* **11**, eadt0318 (2025).
- 33 Xu, B. *et al.* An epidermal stimulation and sensing platform for sensorimotor prosthetic control, management of lower back exertion, and electrical muscle activation. *Advanced Materials (Deerfield Beach, Fla.)* **28**, 4462 (2015).
- 34 Chen, S., Chen, Y., Yang, J., Han, T. & Yao, S. Skin-integrated stretchable actuators toward skin-compatible haptic feedback and closed-loop human-machine interactions. *npj Flexible Electronics* **7**, 1 (2023).
- 35 Leroy, E. & Shea, H. Hydraulically amplified electrostatic taxels (HAXELs) for full

body haptics. *Advanced Materials Technologies* **8**, 2300242 (2023).

- 36 Lim, K. *et al.* Interference haptic stimulation and consistent quantitative tactility in transparent electroactile screen with pressure-sensitive transistors. *Nature communications* **15**, 7147 (2024).

Comment #2: *The subtitles in Results can be refined to improve the narrative logic of the entire article, for instance, the section title of “Material characterization” on page 6 is more suitable to occur in the Experimental Section. More accurate titles closely related to the Results and Discussion should be used.*

Our response (Comment #2):

We thank the reviewer for this helpful suggestion. We revised the subtitles to better reflect the Results and Discussion. We changed “Material characterization” (page 6) to “Design and performance of the stimulation electrodes”.

Authors’ modification to the manuscript:

(Page 7; line 120 in Revised Manuscript)

“Design and performance of the Ag-PU conductor and DMCH”

Comment #3: *The Conclusion is not concise enough. Please improve it.*

Our response (Comment #3):

We thank the reviewer for this valuable suggestion. In response, we have revised the Conclusion to be more concise by summarizing the key findings and implications.

Authors’ modification to the manuscript:

(Page 30; line 692 in Revised Manuscript)

“This holistic approach not only enables personalized tactile feedback across the entire body but also points to promising applications for full-body haptic interfaces.

As a result, the TESS platform opens new possibilities for immersive sensory experiences beyond conventional fingertip or palm-based stimulation. Its ability to deliver precise, adaptive stimulation over large skin areas with feedback control supports applications in entertainment, remote collaboration, and military simulation. These capabilities provide a pathway toward integrating electrical tactile feedback into virtual, augmented, and mixed reality systems, extending conventional haptic perception toward an artificial ‘sixth sense.’ The system’s robust control architecture and its capacity to accommodate interindividual variability in tactile perception represent significant advances with broad implications. Beyond haptic feedback, the TESS device also holds potential for therapeutic electrical stimulation in tissue regeneration, wound healing, pain management, and cosmetic treatments. In integrated VR or AR settings, TESS could deliver realistic tactile feedback alongside therapeutic stimulation, highlighting its multifunctionality. As a next-generation wearable platform, TESS offers a durable, scalable, user-centric solution for consistent electrical stimulation—enhancing comfort and interaction. This approach enhances the usability of electrical stimulation in healthcare and rehabilitation and supports its adoption in immersive digital environments.”

Comment #4: *The Methods Section contains too much content, making it hard to get the important experimental details. It is suggested to retain experiments related to the e-textile*

assembly and its crucial performance evaluation. All other details should be moved to the Supporting Information.

Our response (Comment #4):

We thank the reviewer for this helpful suggestion regarding the organization of the Methods section. In response, we have revised the Methods section to retain only the experiments directly related to the e-textile assembly and its crucial performance evaluations, and moved the remaining methodological details to the Supporting Information.

Authors' modification to the manuscript:

(Page 6 in Revised Supplementary Information; add Supplementary Note 4)

“Supplementary Note 4. Water vapor transmission rate test

Water vapor transmission was measured using a 10 ml beaker (radius = 43 mm) filled with deionized (DI) water. Tests were conducted under controlled environmental conditions—37 °C and 19 % relative humidity—for 24 h using a temperature- and humidity-regulated oven (OF-02PW, JEIO TECH). Each sample, standardized to 2 cm × 2 cm in size, was affixed over the beaker opening and sealed with rubber bands. The cumulative weight loss was recorded using a high-precision analytical balance (WBA-220, DAIHAN Scientific). WVTR was calculated using the following equation:

$$\text{WVTR} = \frac{\Delta W(\text{g})}{At(\text{m}^2 \text{ h})}, \quad (1)$$

where ΔW is the weight loss (g), A is the sample area (m²), and t is the duration (24 h).”

(Page 7 in Revised Supplementary Information; add Supplementary Note 5)

“Supplementary Note 5. Sweat user test

Four participants (both male and female) in their 20s, with a mean age of 24.75 years (two females; Supplementary Fig. 36d), took part in a user test designed to evaluate the effects of perspiration on electrical stimulation. The objective was to assess the amplification of current signals caused by sweat and the accompanying spread of stinging sensations across the skin. Prior to the exercise, participants washed their forearms and attached electrodes in a dry state. A standardized stimulation protocol of 10 V at 5 Hz—common across all participants—was applied both before and after physical activity to measure current changes. Participants then engaged in a 10-min run under controlled environmental conditions (22 °C, 72 % relative humidity). To assess breathability, participants rested in a seated position for 10 min at 22.3 °C and 55 % humidity, followed by another current measurement. The total stimulation duration did not exceed 10 min. Immediately afterward, participants evaluated the level of itchiness induced by sweat accumulation. This assessment was subjective and based on relative perception. Stimulation was applied sequentially using the textile-based haptic electrode, TENS electrode-1, and TENS electrode-2, and all evaluations were conducted collectively after the test.”

(Page 8 in Revised Supplementary Information; add Supplementary Note 6)

“Supplementary Note 6. Peeling force measurement

To assess adhesion strength, Ag-PU conductor samples were fabricated by partially overlapping the DMCH hydrogel onto textiles fully printed with Ag-PU ink. For the Ecoflex-Ag comparison samples, Ecoflex (Ecoflex 00-30, Smooth-On) was first mixed with an organic solvent (MIBK, 2-methyl-4-pentanone, Sigma-Aldrich) for 30 min. Ag flakes (~1.3 μm, DSF-500 MWZ-S, Daejoo Electronic Materials Co., Ltd.) were then added and mixed via mechanical stirring for 2 h. Once the conductive inks were printed onto the textiles, DMCH

was partially laminated on top. All samples were cured in an oven at 60 °C for 10 min. The adhesion strength between the textile and DMCH was evaluated using a universal testing machine (Instron-5543, INSTRON) by stretching the samples until delamination occurred.”

(Page 9 in Revised Supplementary Information; add Supplementary Note 7)

“Supplementary Note 7. Components of the control circuit

The entire control circuit was assembled on a custom-designed, double-sided flexible printed circuit board (FPCB) manufactured by PCBway. The board featured an 18- μm thick metal pattern with an immersion gold surface finish. Component soldering was carried out using a soldering iron (WE1010NA, Weller) and a hot air rework station (EX-930, Exso). Solder wire (91-6040-9013, Kester) and solder paste (TS391LT, Chip Quik) were used during the bonding process. The microcontroller unit (MCU) (ISP1807, Insight SIP), equipped with an integrated BLE antenna, provided compact wireless communication capabilities. A quad precision operational amplifier (OPA4191, Texas Instruments), a digital potentiometer (MAX5419, Analog Devices), and various resistors and capacitors were firmly mounted on the board to support electrical stimulation signal generation.

Power for the amplification block was supplied by a DC-DC switching converter (MCP1663, Microchip Technology) in combination with an inductor (LPS6235-123MRC, Coilcraft), a diode (UPS5819, Microchip Technology), and other passive components. The power management circuit included a low-dropout (LDO) linear regulator (NCV8537MN180R2G, Onsemi), a linear charger for a lithium-polymer battery (LTC4065LEDC, Analog Devices), and a buck converter (LM5166XDRCR, Texas Instruments) for wireless DC-to-DC conversion. Additional components included a rectifier diode (BAS4002ARPPE6) and an inductor (LPS5030-562MRB, Coilcraft), along with a silicone-coated wire-wound loop coil for inductive energy harvesting.”

(Page 10 in Revised Supplementary Information; add Supplementary Note 8)

“Supplementary Note 8. Control circuit operation

The haptic system features wireless communication and is operated through a graphical user interface (GUI), allowing users to control and deliver electrical stimulation feedback seamlessly. The custom-designed electrical stimulation circuit supports independent pulse waveform generation of up to 30 V across a 16-channel electrical stimulation array, with four analog-to-digital converter (ADC) ports assigned for pressure sensing (Supplementary Fig. 30). The system is powered by a rechargeable lithium-polymer (Li-Po) battery, regulated to a stable 3.3 V using a LDO linear regulator. A MCU with an integrated 2.4 GHz antenna enables BLE communication, eliminating the need for external antennas and conserving space. Each general-purpose input/output (GPIO) pin of the MCU produces modulated square waves, which are fed into an amplification circuit. The circuit is based on a non-inverting amplifier topology, equipped with a digitally programmable potentiometer connected via an I²C interface. This configuration allows real-time tuning of the amplifier gain per channel, enabling precise output control up to 30 V. The digital potentiometer adjusts the resistance at the negative input of the operational amplifier by communicating with the MCU, achieving the desired gain dynamically. The amplifier, featuring a rail-to-rail output, operates with a boosted supply voltage from a DC converter, which elevates the LDO-regulated 3.3 V to the required 30 V stimulation voltage, ensuring consistent output in response to control signals (Supplementary Fig. 31). To support real-time impedance compensation, the MCU’s integrated 14-bit ADC is coupled with a pressure sensor, enabling feedback control. The system’s flexible printed circuit board (FPCB), made of polyimide, is divided into four segments interconnected by stretchable serpentine traces, enhancing mechanical flexibility and wearability (Supplementary Fig. 32a–

c). Further segmentation was avoided to maintain an optimal balance between board size and mechanical stretchability. Power management is handled by an integrated circuit coupled with a receiver-loop antenna, enabling continuous operation. Energy harvesting is achieved via an inductively coupled coil adjacent to the board. When placed near a near-field communication (NFC) power transmission antenna, the coil captures energy to recharge the Li-Po battery, supporting wireless recharging capabilities (Supplementary Fig. 32d).”

(Page 12 in Revised Supplementary Information; add Supplementary Note 9)

“Supplementary Note 9. Tremor treatment

Hand tremors were accurately captured using the BNO055 IMU. The controllers embedded with this sensor acted as BLE peripherals, transmitting 3-axis accelerometer data, 3-axis orientation data, and switch status to an NRF52832 receiver. These data enabled quantitative and objective evaluation of tremor severity. The receiver relayed the wireless data to a Unity platform using the UART protocol. To assess tremor characteristics, the 3-axis accelerometer data were processed by computing the root mean square (RMS) of the combined signals. A 4th-order Butterworth filter with a bandpass range of 4–12 Hz was applied to isolate frequency components relevant to ET. Data were logged as time-series sequences with a sampling rate of 50 Hz (timestamps recorded every 20 ms), and included three key components: accelerometer and orientation data, movement trajectory within the VR environment, and task identification. Filtered time-domain signals were transformed into the frequency domain using a fast Fourier transform (FFT). After discarding negative frequency components, the PSD was calculated by doubling the positive frequency values to preserve total signal energy. The peak frequency in the PSD was identified as the dominant tremor frequency, while overall tremor power was quantified by integrating the PSD values within the 4–12 Hz range and normalizing the results to each participant's baseline, providing a standardized metric for pre- and post-exercise tremor evaluation.”

Comment #5: The scale bar unit should be μm instead of um , for example, Figure S8

Our response (**Comment #5**):

We thank the reviewer for pointing out this formatting issue. We have corrected the scale bar unit from “um” to “ μm ” in Figure S8.

Authors’ modification to the manuscript:

(Page 23 in Revised Supplementary Information)

Supplementary Fig. 8 | Electrical performance of Ag-PU conductors on different textile substrates

a, Optical microscopy images of the textile substrate. The substrates were: Textile 1 (87% polyester, 13% spandex), Textile 2 (90% nylon, 10% polyurethane), and Textile 3 (88%

polyester, 12% polyurethane). **b**, Absolute resistance of Ag-PU patterns (length 30 mm, pattern width 2 mm) printed on three textile substrates with varying compositions and weave/knit structures. The resulting mean resistances (1.90 Ω , 1.96 Ω , and 2.08 Ω , respectively) indicate that the conductor maintains consistent electrical performance across different substrate structures.

Comment #6: *The numbering in Figure S29 and Figure S33 is inconsistent with that in the figure caption. Please carefully check.*

Our response (Comment #6):

We thank the reviewer for pointing out this formatting issue. We have carefully checked Figures S29 and S33 and corrected the numbering to ensure it is fully consistent with the corresponding figure captions.

Authors' modification to the manuscript:

(Page 44 in Revised Supplementary Information)

Supplementary Fig. 29 | Participant pre-electrical characteristics for haptic systems

a, Additional pressure induced current ($n = 8$). **b**, Electrical stimulation control system channel matching images for each arm position, along with the corresponding threshold current ($n = 8$; bar height, mean; error bars, s.d.). **c**, Linear current flowing through the skin as a function of input voltage.

(Page 48 in Revised Supplementary Information)

Supplementary Fig. 33 | Photograph of a participant wearing a forearm-based TESS system featuring an array of four electrotactile stimulator and pressure sensor pairs printed on a sleeve-type pressure garment

a, A photograph of the garment worn on a participant with a thinner arm, resulting in relatively low contact pressure measured by the sensors. **b**, A photograph of the same garment worn on a participant with a thicker arm, resulting in relatively high contact pressure. Scale bar, 20 mm.